# Dynamic Regret via Discounted-to-Dynamic Reduction with Applications to Curved Losses and Adam Optimizer

**Yan-Feng Xie** [1 2]  **Yu-Jie Zhang** [3]  **Peng Zhao** [1 2]  **Zhi-Hua Zhou** [1 2]

## Abstract

We study dynamic regret minimization in non-stationary online learning, with a primary focus on follow-the-regularized-leader (FTRL) methods. FTRL is important for curved losses and for understanding adaptive optimizers such as Adam, yet existing dynamic regret analyses are less explored for FTRL. To address this, we build on the discounted-to-dynamic reduction and present a modular way to obtain dynamic regret bounds of FTRL-related problems. Specifically, we focus on two representative curved losses: linear regression and logistic regression. Our method not only simplifies existing proofs for the optimal dynamic regret of online linear regression, but also yields new dynamic regret guarantees for online logistic regression. Beyond online convex optimization, we apply the reduction to analyze the Adam optimizers, obtaining optimal convergence rates in stochastic, non-convex, and non-smooth settings. The reduction also enables a more detailed treatment of Adam with two discount parameters $(\beta_1, \beta_2)$, leading to new results for both clipped and clip-free variants of Adam optimizers.

## 1. Introduction

Online learning provides a general framework for modeling machine learning and sequential decision-making problems, and has been studied extensively (Cesa-Bianchi & Lugosi, 2006; Shalev-Shwartz, 2012; Lattimore & Szepesvári, 2020). In this paper, we study the non-stationary online learning under the online convex optimization (OCO) (Zinkevich, 2003; Hazan, 2016; Orabona, 2019), where at each round $t \in [T]$, the learner selects a

decision $\mathbf{x}_t \in \mathcal{X}$ from a compact convex set $\mathcal{X} \subseteq \mathbb{R}^d$, and then suffers a loss $f_t(\mathbf{x}_t)$, where $f_t : \mathcal{X} \to \mathbb{R}$ is convex. A widely-used performance metric for non-stationary online learning is *dynamic regret* (Zhang et al., 2018), which compares the learner against a time-varying sequence of comparators $\mathbf{u}_1, \ldots, \mathbf{u}_T \in \mathcal{X}$:

$$\text{D-REG}_T(\mathbf{u}_{1:T}) = \sum_{t=1}^{T} f_t(\mathbf{x}_t) - \sum_{t=1}^{T} f_t(\mathbf{u}_t), \quad (1)$$

which reduces to standard static regret when the comparator is fixed, i.e., $\mathbf{u}_t = \mathbf{u}$ for all $t$. Dynamic regret can better reflect the performance in non-stationary environments.

One classical algorithmic framework in OCO is online mirror descent (OMD) (Beck & Teboulle, 2003). OMD makes *local* and greedy updates by taking a gradient step from the current iterate $\mathbf{x}_t$, so intuitively it can be less affected by past information and reacts quickly to environmental changes. Accordingly, a common approach to derive dynamic regret guarantees is to use OMD-type methods and then leverage a two-layer online ensemble (Zhao et al., 2024) to adapt to unknown non-stationarity.

Another fundamental framework is follow-the-regularized-leader (FTRL) (Abernethy et al., 2008). In contrast to OMD, FTRL selects $\mathbf{x}_t$ by minimizing a *global* objective, typically a cumulative loss plus a regularizer. This update rule makes it a convenient tool for optimizing the static regret of many online learning problems, especially those where one must carefully exploit loss structure to obtain advanced guarantees. A prominent example is online learning with curved losses, such as online linear regression, online logistic regression (Jézéquel et al., 2020; Agarwal et al., 2022; Mayo et al., 2022; Jacobsen & Cutkosky, 2024), and settings where regret optimality and computational efficiency are both central (Zimmert et al., 2022; Mhammedi & Rakhlin, 2022; Mhammedi & Gatmiry, 2023; Jézéquel et al., 2025).

However, this strength of FTRL comes with a trade-off: using all the past information can slow down adaptation in non-stationary environments. Indeed, there are negative results showing that FTRL can suffer linear dynamic regret even when the environments change mildly (Jacobsen & Cutkosky, 2022, Theorem 2). Recent work aims to retain

---

[1]State Key Laboratory of Novel Software Technology, Nanjing University, China [2]School of Artificial Intelligence, Nanjing University, China [3]University of Washington, United States. Correspondence to: Peng Zhao <zhaop@lamda.nju.edu.cn>.

*Proceedings of the 43rd International Conference on Machine Learning*, Seoul, South Korea. PMLR 306, 2026. Copyright 2026 by the author(s).

the core framework of FTRL while introducing modifications to help it better track changing comparators. A notable result is by Mhaisen & Iosifidis (2025), who propose injecting correction terms to effectively prune the history, achieving an optimal dynamic regret guarantee for convex losses. However, existing results fail to support several important curved losses, where FTRL exhibits advantages over OMD.

A recent insightful result by Ahn et al. (2024) demonstrates a natural yet intriguing connection between the dynamic regret of FTRL and the Adam optimizer (Kingma & Ba, 2015), particularly in understanding the role of its momentum components. Specifically, Ahn et al. (2024) show that Adam can be viewed as an instance of FTRL that optimizes a *discounted regret*, defined as:

$$\text{REG}_{t;\beta}(\mathbf{u}) = \sum_{s=1}^{t} \beta^{t-s}\big(f_s(\mathbf{x}_s) - f_s(\mathbf{u})\big). \tag{2}$$

Ahn et al. (2024) establish a reduction linking discounted regret minimization to dynamic regret. This connection suggests that Adam can be viewed as implicitly minimizing dynamic regret, which helps handle non-stationarity in non-convex and non-smooth settings. Subsequent work (Ahn & Cutkosky, 2024) leverages this perspective to analyze the discounted regret of a slightly modified, clipped variant of Adam and derive convergence guarantees. Although successful, their analysis focuses on the coupled choice $\beta_2 = \beta_1^2$, which is narrower than many parameter settings used in practice. Analyzing the convergence guarantees with more general parameter choices (in particular, $\beta_2 \neq \beta_1^2$) calls for a fine-grained way to carry out the discounted-to-dynamic (D2D) reduction.

Building on the D2D reduction (Ahn et al., 2024), we develop a *modular discounted-to-dynamic analysis* that is suitable for analyzing dynamic regret of FTRL: rather than applying the D2D reduction after committing to a tuned closed-form discounted regret bound, we cast it at the level of *untuned upper-bound templates* on rescaled losses. This template-level view keeps the key terms that contribute to the guarantees explicit throughout the reduction and enables more *flexible* tuning, while remaining reusable across different algorithmic instantiations and optimization settings.

**Our Result I: Dynamic Regret of Curved Losses.** We instantiate the modular reduction on two representative regression tasks. For online linear regression (Vovk, 2001; Azoury & Warmuth, 2001), it *streamlines* the analysis of (Jacobsen & Cutkosky, 2024) by avoiding algorithm-specific Bregman-divergence arguments, while recovering a matching optimal bound. For online logistic regression, it yields a *new* dynamic regret guarantee for a discounted variant of AIOLI (Jézéquel et al., 2020), avoiding the exponential dependence that typically arises in logistic regression and improving robustness in non-stationary environments.

**Our Result II: Adam Optimizers.** Beyond online optimization, the same D2D analysis can be applied to analyze the convergence rates of Adam optimizer in stochastic, non-convex and non-smooth optimization via the online-to-non-convex (O2NC) conversion (Cutkosky et al., 2023; Zhang & Cutkosky, 2024). Using this approach, we obtain convergence guarantees under more flexible parameter choices rather than the restricted choice $\beta_2 = \beta_1^2$ that is required in prior analysis (Ahn & Cutkosky, 2024), while maintaining optimal convergence to a $(c, \epsilon)$-stationary point (Zhang & Cutkosky, 2024) under bounded stochastic gradients and mild conditions on the numerical stability term $\nu$.

In particular, for clipped Adam we provide *two sufficient convergence conditions*. The *first* requires $\beta_2 \geq \max\{1 - \frac{\nu}{G+\sigma}, \beta_1^4\}$, where $\nu$ is the numerical stability term in Adam's denominator and $G$ and $\sigma$ are the Lipschitz constant and gradient variance. This helps clarify the role of clipping: it allows a weaker coupling between the momentum parameters, down to $\beta_2 \geq \beta_1^4$. The *second* gives a margin-style condition with $\beta_2 > \beta_1^2$ and quantifies how the gap $\beta_2 - \beta_1^2$ affects the convergence rate. We also study the *clip-free Adam variant* in non-smooth and non-convex settings and obtain optimal convergence rates using a similar analysis. These flexible $(\beta_1, \beta_2)$ conditions come at the cost of additionally assuming bounded stochastic gradients, a condition also adopted in related analyses; the only prior result without this assumption (Ahn & Cutkosky, 2024) is restricted to $\beta_2 = \beta_1^2$. Finally, our results can imply optimal rates in the non-convex smooth and second-order smooth settings.

**Techniques.** We cast the D2D reduction at the level of untuned upper bounds on rescaled losses, presenting explicit dynamic regret bounds. *(i)* First, this view makes the analysis of dynamic regret more reusable: it leads to a short proof of the discounted linear regression forecaster in the unconstrained setting, and provides a direct way to obtain guarantees for another representative curved loss, the logistic loss. More broadly, the same framework applies beyond the examples in this paper as a general tool for non-stationary environments. *(ii)* Second, when applying the O2NC conversion to analyze Adam in non-convex and non-smooth settings, a main difficulty arises when $\beta_2 \neq \beta_1^2$. In this case, the induced learning rate in the online problem is no longer monotone, which breaks standard arguments. Our reduction helps isolate the terms that determine the final convergence rate, allowing us to control them separately. This is particularly useful for obtaining fine-grained guarantees for Adam. In particular, we introduce a self-confident tuning lemma to handle learning rates whose denominators are exponential-weighted sums of past gradients. We also refine the analysis of the induced online algorithm to remove $\nu$ from the condition on $\beta_2$. A key step is to bound how much the learning rate can change across rounds, fol-

lowing a scale-free style argument (Orabona & Pál, 2018). This leads to a new convergence condition that makes the dependence on the margin $\beta_2 - \beta_1^2$ explicit.

**Organization.** Section 2 presents the discounted-to-dynamic modular analysis. Section 3 studies the dynamic regret of curved losses using the modular reduction. Section 4 studies convergence conditions of both clipped and clip-free variants of Adam via the O2NC conversion. Section 5 concludes the paper. We defer discussion of related work, proofs, and additional details to the appendices.

# 2. Modular Discounted-to-Dynamic Reduction

In this section, we first review the connection between discounted regret and dynamic regret optimization, and then present our discounted-to-dynamic modular analysis with further discussions. All proofs are deferred to Appendix B.

## 2.1. Connection of Dynamic and Discounted Regret

Ahn et al. (2024) reveal the connection between dynamic regret and discounted regret to interpret the role of momentum terms in Adam. We summarize their result below.

**Lemma 1** (Adapted from Theorem B.3 in Ahn et al. (2024)). *The following statement is true for any $T > 0$ and any comparator sequence $\mathbf{u}_1, \ldots, \mathbf{u}_T \in \mathcal{X} \subseteq \mathbb{R}^d$:*

$$\text{D-REG}_T(\mathbf{u}_{1:T}) = \beta \sum_{t=1}^{T-1} \left( \text{REG}_{t;\beta}(\mathbf{u}_t) - \text{REG}_{t;\beta}(\mathbf{u}_{t+1}) \right)$$
$$+ (1 - \beta) \sum_{t=1}^{T} \text{REG}_{t;\beta}(\mathbf{u}_t) + \beta \text{REG}_{T;\beta}(\mathbf{u}_T),$$

*where $\text{REG}_{t;\beta}(\mathbf{u})$ is defined in Eq. (2).*

To interpret this result, note that the left-hand side is dynamic regret, which benchmarks against a *changing* comparator sequence. The right-hand side is expressed through discounted regrets: it depends on the discount factor $\beta$, but at each time it compares only against a *fixed* comparator. This shift is helpful in settings where the available analysis tools are primarily designed for fixed comparators, as is often the case for FTRL methods (Orabona, 2019).

A direct way to leverage Lemma 1 is to first establish discounted regret bounds for $\text{REG}_{t;\beta}$ and then plug them into the identity to obtain a dynamic regret bound. This approach is general and has been successfully used in analyzing clipped Adam (Ahn & Cutkosky, 2024). However, the resulting expression typically involves multiple sums and terms, and simplifying it may obscure how the final bound depends on the key quantities of interest (e.g., non-stationarity and stability). Motivated by this, we present a complementary, modular way to carry out the reduction:

we cast the D2D reduction at the level of untuned upper bounds, which keeps the main terms explicit throughout the derivation and facilitates tuning and trade-offs needed for optimal and more informative guarantees.

## 2.2. Modular D2D Reduction

Building on the D2D reduction, we present the following theorem, which provides a modular analysis for FTRL across a range of settings. Our analysis is stated at the level of untuned upper-bound templates, before committing to a specific tuning. This template-level form preserves flexibility in learning-rate choices and makes the conversion reusable across different algorithmic families.

**Theorem 1** (Modular D2D Reduction). *Assume an online learning algorithm $\mathcal{A}$ outputs decisions $\mathbf{x}_t \in \mathcal{X}$ and satisfies the following rescaled regret bound: for any $t \in [T]$ and any comparator $\mathbf{u} \in \mathcal{X}$,*

$$\sum_{s=1}^{t} \beta^{-s} f_s(\mathbf{x}_s) - \sum_{s=1}^{t} \beta^{-s} f_s(\mathbf{u}) \leq \varphi_t(\mathbf{u}) + \sum_{s=1}^{t} \Lambda_s, \quad (3)$$

*where $\beta \in (0, 1]$, $\Lambda_s \geq 0$ are stability terms, and $\varphi_t(\cdot) \geq 0$ is a comparator-dependent quantity that appears in the analysis. Then, for any $\mathbf{u}_1, \ldots, \mathbf{u}_T \in \mathcal{X}$, $\mathcal{A}$ satisfies*

$$\sum_{t=1}^{T} f_t(\mathbf{x}_t) - \sum_{t=1}^{T} f_t(\mathbf{u}_t) \leq \beta \, \varphi_1(\mathbf{u}_1) + \sum_{t=1}^{T} \beta^t \Lambda_t$$
$$+ \beta \sum_{t=1}^{T-1} \left( F_t^{\beta,\varphi}(\mathbf{u}_{t+1}) - F_t^{\beta,\varphi}(\mathbf{u}_t) \right)$$
$$+ \beta \sum_{t=1}^{T-1} \beta^t \left( \varphi_{t+1}(\mathbf{u}_{t+1}) - \varphi_t(\mathbf{u}_{t+1}) \right),$$

*where $F_t^{\beta,\varphi}(\mathbf{u}) = \beta^t \varphi_t(\mathbf{u}) + \sum_{s=1}^{t} \beta^{t-s} f_s(\mathbf{u})$.*

Theorem 1 is inspired by the work of Ahn et al. (2024); Jacobsen & Cutkosky (2024). The term $\sum_{t=1}^{T} \beta^t \Lambda_t$ aggregates stability contributions and captures how much the iterates change over time (e.g., between $\mathbf{x}_{t-1}$ and $\mathbf{x}_t$). In many analyses, $\Lambda_t$ takes the form $\Lambda_t \approx \|\beta^{-t} \nabla f_t(\mathbf{x}_t)\|_{t,*}^2$, where $\|\cdot\|_t$ is a time-varying norm and $\|\cdot\|_{t,*}$ is its dual.

The quantity $\varphi_t(\cdot)$ is analysis-specific and need not coincide with the regularizer used by the algorithm. For online regression, one can take $\varphi_t(\mathbf{u}) = \frac{\lambda}{2}\|\mathbf{u}\|_2^2$, which is time-independent; hence the term $\varphi_{t+1}(\mathbf{u}_{t+1}) - \varphi_t(\mathbf{u}_{t+1})$ vanishes. For general cases, however, $\varphi_t$ typically varies with $t$, and we need to control the resulting difference term to ensure it does not affect the optimal rate.

To understand the term $\beta \sum_{t=1}^{T-1} \left( F_t^{\beta,\varphi}(\mathbf{u}_{t+1}) - F_t^{\beta,\varphi}(\mathbf{u}_t) \right)$, for simplicity, we assume $\varphi_t(\mathbf{u}) = \varphi(\mathbf{u})$ is time-independent and define $f_0(\mathbf{u}) = \varphi(\mathbf{u})$. Then this term

can be bounded by $\frac{\beta}{1-\beta}P_T^\beta$, where

$$P_T^\beta = \sum_{t=1}^{T-1}\sum_{s=0}^{t} p_{t,s}^\beta \big[f_s(\mathbf{u}_{t+1}) - f_s(\mathbf{u}_t)\big]_+, \qquad (4)$$

where $[a]_+ = \max\{a,0\}$ and $p_{t,s}^\beta = \frac{\beta^{t-s}}{\sum_{\tau=0}^{t}\beta^{t-\tau}}$ is a normalized geometric weight. The quantity $P_T^\beta$ measures comparator variation through loss differences, while the standard path length $P_T = \sum_{t=1}^{T-1}\|\mathbf{u}_{t+1} - \mathbf{u}_t\|_2$ (Zinkevich, 2003) measures variation in decision space. Moreover, if each $f_s$ is $G$-Lipschitz with respect to a norm $\|\cdot\|$ (i.e., $\|\nabla f_s(\mathbf{u})\|_* \leq G$), then $P_T^\beta \leq G\sum_{t=1}^{T-1}\|\mathbf{u}_{t+1} - \mathbf{u}_t\|$, where $\|\cdot\|$ can be chosen to match the problem geometry. As shown by Jacobsen & Cutkosky (2024), dynamic regret bounds in terms of $P_T^\beta$ can be optimal for unconstrained online linear regression.

To summarize, Theorem 1 provides a modular template for deriving explicit dynamic regret guarantees. The next two sections present self-contained applications to further demonstrate the effectiveness of this analysis.

## 3. Dynamic Regret of Curved Losses

We consider the following online regression protocol:

1: **for** $t = 1,\ldots,T$ **do**
2:     the learner receives a feature vector $\mathbf{z}_t \in \mathcal{Z} \subseteq \mathbb{R}^d$;
3:     the learner submits a decision $\mathbf{x}_t \in \mathcal{X} \subseteq \mathbb{R}^d$;
4:     the environment reveals a label $y_t \in \mathcal{Y} \subseteq \mathbb{R}$;
5:     the learner suffers a loss $f_t(\mathbf{x}_t) = \ell(\mathbf{x}_t^\top \mathbf{z}_t, y_t)$.
6: **end for**

Here $\ell(\cdot,\cdot) : \mathbb{R}\times\mathbb{R} \to \mathbb{R}$ is the loss function. We call the protocol improper since the learner chooses $\mathbf{x}_t$ after seeing $\mathbf{z}_t$, using predictions that may not be linear in the instances $\mathbf{z}_t$ (Shalev-Shwartz & Ben-David, 2014). We note that this is a mild assumption in many machine learning problems (Orabona, 2019). In the following, we instantiate this protocol to study the dynamic regret for two representative curved losses: squared loss (Section 3.1) and logistic loss (Section 3.2). Proofs are deferred to Appendix C.

### 3.1. Online Linear Regression

We first study unconstrained online linear regression with squared loss $\ell(\widehat{y}, y) = \frac{1}{2}(\widehat{y} - y)^2$, where $\mathcal{X} = \mathbb{R}^d$, $\mathcal{Z} = \mathbb{R}^d$, and $\mathcal{Y} = \mathbb{R}$ in the protocol. At round $t$, the learner predicts $\widehat{y}_t = \mathbf{x}_t^\top \mathbf{z}_t$ and incurs $f_t(\mathbf{x}_t) = \frac{1}{2}(\mathbf{x}_t^\top \mathbf{z}_t - y_t)^2$.

On bounded domains, the squared loss is exp-concave and second-order methods such as ONS can achieve logarithmic static regret (Hazan, 2016; Hazan et al., 2007). In our setting $\mathcal{X} = \mathbb{R}^d$, such uniform boundedness (and hence global exp-concavity) is not available in general. However, once we can leverage the side information of the feature $\mathbf{z}_t$, the

VAW forecaster (Vovk, 2001; Azoury & Warmuth, 2001) achieves logarithmic regret under this unconstrained setting. Its update $\mathbf{x}_t$ can be written as the FTRL solution:

$$\arg\min_{\mathbf{x}\in\mathbb{R}^d}\Big\{\frac{\lambda}{2}\|\mathbf{x}\|_2^2 + \frac{1}{2}(\mathbf{x}^\top\mathbf{z}_t)^2 + \frac{1}{2}\sum_{s=1}^{t-1}(\mathbf{x}^\top\mathbf{z}_s - y_s)^2\Big\}.$$

The VAW forecaster ensures the following guarantee.

**Lemma 2.** *In unconstrained online linear regression, the VAW forecaster ensures the following bound for any $t \in [T]$:*

$$\sum_{s=1}^{t} f_s(\mathbf{x}_s) - f_s(\mathbf{u}) \leq \frac{\lambda}{2}\|\mathbf{u}\|_2^2 + \frac{1}{2}\sum_{s=1}^{t-1} y_s^2 \mathbf{z}_s^\top A_s^{-1} \mathbf{z}_s,$$

*for any $\mathbf{u} \in \mathbb{R}^d$, where $A_t = \lambda I + \sum_{s=1}^{t}\mathbf{z}_s\mathbf{z}_s^\top$ and $I \in \mathbb{R}^{d\times d}$ is the identity matrix.*

Lemma 2 is stated for a fixed comparator. In non-stationary settings, however, the best predictor may drift over time, so we evaluate performance using dynamic regret (1) and analyze it through the modular D2D reduction in Section 2.

Following Jacobsen & Cutkosky (2024), we consider a discounted variant of VAW with factor $\beta \in (0,1]$, which emphasizes recent data. At round $t$, it chooses $\mathbf{x}_t$ by solving

$$\min_{\mathbf{x}\in\mathbb{R}^d}\Big\{\frac{\lambda\beta^t}{2}\|\mathbf{x}\|_2^2 + \frac{1}{2}(\mathbf{x}^\top\mathbf{z}_t)^2 + \frac{1}{2}\sum_{s=1}^{t-1}\beta^{t-s}(\mathbf{x}^\top\mathbf{z}_s - y_s)^2\Big\}.$$

Using the rescaling trick (Zhang et al., 2024), define $\widetilde{\mathbf{z}}_t = \beta^{-t/2}\mathbf{z}_t$ and $\widetilde{y}_t = \beta^{-t/2}y_t$, so that $\ell(\mathbf{x}^\top\widetilde{\mathbf{z}}_t, \widetilde{y}_t) = \beta^{-t}\ell(\mathbf{x}^\top\mathbf{z}_t, y_t)$. Applying Lemma 2 to the rescaled sequence yields a regret bound of the form for any $t \in [T]$:

$$\sum_{s=1}^{t}\beta^{-s}\Big(\ell(\mathbf{x}_s^\top\mathbf{z}_s, y_s) - \ell(\mathbf{u}^\top\mathbf{z}_s, y_s)\Big) \leq \frac{\lambda}{2}\|\mathbf{u}\|_2^2 + \sum_{s=1}^{t}\Lambda_s,$$

where $\Lambda_s$ is a second-order stability term, given explicitly in Appendix C.1. Thus the above expression matches the condition of Theorem 1 with $\varphi_t(\mathbf{u}) = \frac{\lambda}{2}\|\mathbf{u}\|_2^2$, which yields the following dynamic regret guarantee.

**Theorem 2** (Informal). *In unconstrained online linear regression, the discounted VAW forecaster satisfies the following dynamic regret bound: for any $\mathbf{u}_1,\ldots,\mathbf{u}_T \in \mathbb{R}^d$,*

$$\frac{\beta\lambda}{2}\|\mathbf{u}_1\|_2^2 + \frac{d}{2}\big(\max_{t\in[T]} y_t^2\big)\cdot\ln\left(1 + \frac{\sum_{t=1}^{T}\beta^{T-t}\|\mathbf{z}_t\|_2^2}{\lambda d}\right)$$

$$+ \frac{\beta}{1-\beta}\cdot P_T^\beta + \frac{1-\beta}{\beta}\cdot\frac{d}{2}\sum_{t=1}^{T} y_t^2,$$

*where $\beta \in (0,1)$, $\lambda > 0$, and $P_T^\beta$ is defined in Eq. (4). Moreover, there exists a two-layer ensemble algorithm that ensures an $\mathcal{O}\big(d\log T + \sqrt{dTP_T^\beta}\big)$ dynamic regret.*

The formal statement is deferred to Appendix C.1. Theorem 2 matches the optimal rate of Jacobsen & Cutkosky (2024). Our derivation is more direct: it applies the modular D2D reduction to a rescaled regret template, avoiding the extra step of rewriting the update as OMD and the accompanying Bregman-divergence arguments (Jacobsen & Cutkosky, 2024). Compared to discounted online newton step (ONS) analyses (Hazan et al., 2007; Yuan & Lamperski, 2020), our result does not rely on exp-concavity and allows unconstrained decisions. By leveraging the two-layer algorithm proposed in Jacobsen & Cutkosky (2024), we can obtain an $\mathcal{O}(d \log T + \sqrt{dTP_T^\beta})$ bound, matching the lower bound for this unconstrained setting.

## 3.2. Online Logistic Regression

We next study online logistic regression with $\ell(\widehat{y}, y) = \ln(1 + \exp(-y\widehat{y}))$. At round $t$, the learner predicts $\widehat{y}_t = \mathbf{x}_t^\top \mathbf{z}_t$ and incurs $f_t(\mathbf{x}_t) = \ln(1 + \exp(-y_t \mathbf{x}_t^\top \mathbf{z}_t))$. We assume binary labels $\mathcal{Y} = \{+1, -1\}$ and bounded features $\mathcal{Z} = \{\mathbf{z} \in \mathbb{R}^d : \|\mathbf{z}\|_2 \leq R\}$. The learner may choose $\mathbf{x}_t \in \mathbb{R}^d$, while the comparator sequence satisfies $\|\mathbf{u}_t\|_2 \leq B$ for all $t$. We follow the conventions to make the boundedness assumptions in online logistic regression (Hazan et al., 2007; Foster et al., 2018; Jézéquel et al., 2020).

On the bounded comparator set $\|\mathbf{u}\|_2 \leq B$, the logistic loss is exp-concave, so applying ONS gives a static regret bound $\mathcal{O}(de^B \log T)$: logarithmic in $T$ but exponential in $B$. Such an exponential dependence on $B$ is generally unavoidable for proper OCO algorithms (Hazan et al., 2014). However, Foster et al. (2018) show the importance of impropriety, and one can reduce the dependence on $B$ from $e^B$ to $B$ with an efficient method (Jézéquel et al., 2020).

To handle the potential drift of the regression target, we further aim to efficiently obtain dynamic regret guarantees while still avoiding the exponential dependence on $B$. To this end, we propose a discounted variant of the AIOLI algorithm (Jézéquel et al., 2020), formalized as:

$$\mathbf{x}_t = \arg\min_{\mathbf{x} \in \mathbb{R}^d} \left\{ \frac{\beta^t \lambda}{2} \|\mathbf{x}\|_2^2 + h_t(\mathbf{x}) + \sum_{s=1}^{t-1} \beta^{t-s} \widehat{f}_s(\mathbf{x}) \right\},$$

where $h_t(\mathbf{x}) = \ell(\mathbf{x}^\top \mathbf{z}_t, +1) + \ell(\mathbf{x}^\top \mathbf{z}_t, -1)$ uses the observed feature $\mathbf{z}_t$ and can be viewed as an optimism term (Chiang et al., 2012; Rakhlin & Sridharan, 2013), realized as a guess of the incoming loss function. Here $\widehat{f}_t(\mathbf{x})$ is a surrogate of $f_t(\mathbf{x})$ given by $\widehat{f}_t(\mathbf{x}) = f_t(\mathbf{x}_t) + \langle \nabla f_t(\mathbf{x}_t), \mathbf{x} - \mathbf{x}_t \rangle + \frac{\eta_t}{2} \langle \nabla f_t(\mathbf{x}_t), \mathbf{x} - \mathbf{x}_t \rangle^2$, with $\eta_t = \frac{\exp(y_t \widehat{y}_t)}{1 + BR}$.

In Lemma 3, we establish the anytime guarantees of rescaled regret from the view of optimistic FTRL (Orabona, 2019), satisfying the condition of Theorem 1. By applying the proposed modular analysis, we conclude the following result.

*Table 1.* Comparison of dynamic regret and per-round computational cost for online logistic regression. Here $B$ is the comparator norm and $P_T$ is the path length.

| Work | Dynamic Regret | Per-round Cost |
|---|---|---|
| Baby & Wang (2021) | $\widetilde{\mathcal{O}}(d^{3.5} e^B T^{1/3} P_T^{2/3})$ | $\widetilde{\mathcal{O}}(d^3)$ |
| Zhang et al. (2025) | $\widetilde{\mathcal{O}}(d T^{1/3} P_T^{2/3})$ | $\widetilde{\mathcal{O}}(d^{12} T^{11} + T^{23})$ |
| Ours (Thm. 3) | $\mathcal{O}(dB \log(BT) + \sqrt{dBTP_T})$ | $\widetilde{\mathcal{O}}(d^2)$ |

**Theorem 3** (Informal). *Assume* $\|\mathbf{z}_t\|_2 \leq R$ *for all* $t \in [T]$. *For any comparator sequence* $\mathbf{u}_1, \ldots, \mathbf{u}_T$ *satisfying* $\|\mathbf{u}_t\|_2 \leq B$, *the dynamic regret bound of the discounted AIOLI algorithm is:*

$$\beta\lambda\|\mathbf{u}_1\|_2^2 + d(1 + BR) \log\left(1 + \frac{R^2 \sum_{t=1}^T \beta^{T-t}}{d\lambda(1 + BR)}\right)$$
$$+ \frac{\beta}{1-\beta} \cdot P_T^\beta + \frac{1-\beta}{\beta} \cdot d(1 + BR)T,$$

*where* $\beta \in (0, 1)$, $\lambda > 0$, *and* $P_T^\beta$ *is defined in Eq.* (4). *Moreover, there exists a two-layer ensemble algorithm that ensures an* $\mathcal{O}(dB \log(BT) + \sqrt{dBTP_T^\beta})$ *dynamic regret.*

The formal statement and proof are deferred to Appendix C.2. In Appendix C.3, we leverage the mixability of the logistic losses to propose a two-layer online ensemble algorithm to tune the discount factor $\beta$ (Mhammedi & Rakhlin, 2022; Zhao et al., 2024). We remark that discounted AIOLI update lies $\mathbb{R}^d$ and do not require projection. The main computational cost comes from an implicit update involving the term $h_t(\mathbf{x})$, which can be implemented efficiently with $\mathcal{O}(\log T)$ iterations (Jézéquel et al., 2020).

Since the comparators are bounded in online logistic regression, the lower bound for unconstrained linear regression does not directly apply (Jacobsen & Cutkosky, 2024). There are results that achieve the optimal dependence on $T$ and the standard path length $P_T$, with $\widetilde{\mathcal{O}}(T^{1/3} P_T^{2/3})$ dynamic regret bounds for exp-concave losses under different algorithmic designs (Baby & Wang, 2021; Zhang et al., 2025), where we use $\widetilde{\mathcal{O}}(\cdot)$ to hide logarithmic factors in $T$. These approaches either rely on more involved KKT-condition based analyses (Baby & Wang, 2021; 2022), or in some cases incur higher computational overhead (Zhang et al., 2025). Our approach provides a complementary trade-off, summarized in Table 1: we obtain an $\mathcal{O}(dB \log T + \sqrt{dBTP_T})$ bound via an efficient algorithm with a modular analysis, and our framework also supports unbounded decision sets.

## 4. Convergence Conditions for Adam

In this section, we combine the exponentiated O2NC framework (Cutkosky et al., 2023; Zhang & Cutkosky, 2024) with the modular D2D reduction (Theorem 1) to analyze

Adam optimizers in stochastic, non-convex, and non-smooth settings. We first introduce the setup and the O2NC conversion, study clipped Adam following the template of Ahn & Cutkosky (2024), and then study a clip-free variant that replaces clipping with an additional damping term in the step size. All the proofs are deferred to Appendix D.

## 4.1. Setup

We study the following stochastic, non-convex and non-smooth optimization problem, where $F : \mathbb{R}^d \to \mathbb{R}$ is the objective function defined as:

$$\min_{\mathbf{x} \in \mathbb{R}^d} F(\mathbf{x}) \triangleq \mathbb{E}_\xi \left[ f(\mathbf{x}; \xi) \right].$$

We make the following assumptions.

**Assumption 1.** $F$ is differentiable and $G$-Lipschitz, i.e., $\|\nabla F(\mathbf{x})\|_2 \leq G < \infty$ for all $\mathbf{x} \in \mathbb{R}^d$.

**Assumption 2.** Given an initial point $\mathbf{x}_0$, $F(\mathbf{x}_0) - \inf_{\mathbf{x} \in \mathbb{R}^d} F(\mathbf{x}) \leq F^*$ for some known $F^*$.

**Assumption 3.** Assume $F$ satisfies that for any $\mathbf{x}, \mathbf{y}$, $F(\mathbf{y}) - F(\mathbf{x}) = \int_0^1 \langle \nabla F(\mathbf{x} + t(\mathbf{y} - \mathbf{x})), \mathbf{y} - \mathbf{x} \rangle \mathrm{d}t$.

**Assumption 4.** The stochastic gradient is unbiased, $\mathbb{E}_\xi[\nabla f(\mathbf{x}; \xi)] = \nabla F(\mathbf{x})$, has bounded variance, $\mathbb{E}_\xi \|\nabla f(\mathbf{x}; \xi) - \nabla F(\mathbf{x})\|_2^2 \leq \sigma^2$, and is almost surely bounded, i.e., $\|\nabla f(\mathbf{x}; \xi)\|_2 \leq G$.

Assumptions 1 – 3 are standard in the O2NC framework (Cutkosky et al., 2023; Zhang & Cutkosky, 2024). To obtain more fine-grained guarantees under flexible $(\beta_1, \beta_2)$, we additionally assume bounded stochastic gradients as a trade-off, a common technical condition in related analyses (Cutkosky & Orabona, 2019; Guo et al., 2021; Défossez et al., 2022). This assumption can be relaxed by imposing suitable tail conditions on the noise (van Handel, 2016), at the price of extra logarithmic factors in the rates.

**Definition 1.** Suppose $F : \mathbb{R}^d \to \mathbb{R}$ is a differentiable function. We say that $\mathbf{x}$ is a $(c, \epsilon)$-stationary point if $\|\nabla F(\mathbf{x})\|_c \leq \epsilon$, where

$$\|\nabla F(\mathbf{x})\|_c = \inf_{\substack{\mathbf{y} \sim P \in \mathcal{P}(\mathbb{R}^d) \\ \mathbb{E}[\mathbf{y}] = \mathbf{x}}} \|\mathbb{E}[\nabla F(\mathbf{y})]\|_2 + c \cdot \mathbb{E}\|\mathbf{y} - \mathbf{x}\|_2^2,$$

where $\mathcal{P}(\mathbb{R}^d)$ denotes the set of all distributions over $\mathbb{R}^d$.

This notion is quite general. With a suitable choice of $c$, it recovers several standard notions (Zhang & Cutkosky, 2024), including first-order stationary points if the objective is smooth or second-order smooth. If $F$ is $G$-Lipschitz, a $(c, \epsilon)$-stationary point also implies a $(\delta, (1 + \frac{2G}{c\delta^2})\epsilon)$-Goldstein stationary point, which is widely used in non-convex and non-smooth optimization (Goldstein, 1977; Zhang et al., 2020; Tian et al., 2022; Davis et al., 2022).

---

**Algorithm 1** Exp-O2NC (Zhang & Cutkosky, 2024)

**Input:** OCO algorithm $\mathcal{A}$, initial point $\mathbf{x}_0$, parameters $T \in \mathbb{N}$, $\beta \in (0, 1)$, loss function $\ell_t(\Delta)$.
1: **Initialize:** $\bar{\mathbf{x}}_0 = \mathbf{x}_0$
2: **for** $t = 1, \dots, T$ **do**
3:     Receive $\Delta_t$ from $\mathcal{A}$;
4:     Update $\mathbf{x}_t = \mathbf{x}_{t-1} + s_t \cdot \Delta_t$, $s_t \overset{\text{i.i.d.}}{\sim} \mathrm{Exp}(1)$;
5:     Compute $\mathbf{g}_t = \nabla f(\mathbf{x}_t, \xi_t)$;
6:     Send loss $\ell_t(\Delta)$ to $\mathcal{A}$;
7:     $\bar{\mathbf{x}}_t = \frac{\beta - \beta^t}{1 - \beta^t} \bar{\mathbf{x}}_{t-1} + \frac{1 - \beta}{1 - \beta^t} \mathbf{x}_t$;
8: **end for**
9: **Return:** $\bar{\mathbf{x}} \sim \mathrm{Unif}(\{\bar{\mathbf{x}}_t : t \in [T]\})$.

---

The seminal work of Cutkosky et al. (2023) introduces the O2NC conversion, which reduces stochastic non-convex and non-smooth optimization to an online learning problem and the convergence rate can be bounded by the dynamic regret of the online learner. We focus on an exponentiated variant of O2NC (Exp-O2NC) (Zhang & Cutkosky, 2024), which is more compatible with the D2D reduction.

Algorithm 1 summarizes the exponentiated O2NC conversion (Zhang & Cutkosky, 2024). In Line 4, $\mathrm{Exp}(a)$ denotes the exponential distribution with parameter $a$. Different surrogate losses $\ell_t(\Delta)$ in Line 6 lead to the clipped or the clip-free variants of Adam, which will be specified later.

To understand O2NC, by Lemma 6 and the update rule, we have $\mathbb{E}_{s_t}[F(\mathbf{x}_t) - F(\mathbf{x}_{t-1})] = \mathbb{E}_{s_t}[\langle \nabla F(\mathbf{x}_t), \Delta_t \rangle] = \mathbb{E}_{s_t, \xi_t}[\langle \mathbf{g}_t, \Delta_t \rangle]$. This suggests using an online learner to choose the update direction $\Delta_t$ based on losses built from $\langle \mathbf{g}_t, \Delta \rangle$ (and its variants), which turns the optimization problem into an online learning one.

## 4.2. Understanding Adam via FTRL

Ahn et al. (2024) show that Adam can be understood as a discounted FTRL, which links Adam to the dynamic regret, highlighting the importance of introducing the discount factors. Building on this connection, we can use the modular reduction to analyze the dynamic regret of discounted FTRL, and then transfer the results to Adam. We first formalize the discounted FTRL below.

$$\Delta_{t+1} = \arg\min_{\Delta \in \mathcal{D}} \left\{ \frac{\|\Delta\|_2^2}{2\eta_t} + \sum_{s=1}^t \beta^{-s} \ell_s(\Delta) \right\}, \quad (5)$$

where $\eta_t > 0$ and $\mathcal{D} \subseteq \mathbb{R}^d$ (a closed convex set) will be specified later. Equivalently, by the first-order optimality condition of (5), $\Delta_{t+1}$ satisfies the following form

$$\Delta_{t+1} = \Pi_{\mathcal{D}} \left[ -\eta_t \sum_{s=1}^t \beta^{-s} \nabla \ell_s(\Delta_{t+1}) \right]. \quad (6)$$

In the rest of this section, we will choose $\ell_t(\cdot)$, $\eta_t$, and $\mathcal{D}$ so that the resulting discounted FTRL update matches the (clipped or clip-free) Adam update under the O2NC framework, and provide the guarantees and analysis by reduction.

### 4.3. Results for the Clipped Adam Variant

We first follow the algorithmic template of Ahn & Cutkosky (2024) to study the clipped Adam, and set $\ell_t(\Delta) = \langle \mathbf{g}_t, \Delta \rangle$ and $\mathcal{D} = \{\Delta : \|\Delta\|_2 \leq D\}$. We choose

$$\eta_t = \gamma \cdot \frac{(1-\beta_1)\beta_1^t}{\nu + \sqrt{(1-\beta_2)\sum_{s=1}^t \beta_2^{t-s}\|\mathbf{g}_s\|_2^2}}, \quad (7)$$

and we set $\beta = \beta_1$ in Eq. (5). With these choices, the update rule in Eq. (6) results in:

$$\text{Clip}_D\left[-\gamma \cdot \frac{(1-\beta_1)\sum_{s=1}^t \beta_1^{t-s}\mathbf{g}_s}{\nu + \sqrt{(1-\beta_2)\sum_{s=1}^t \beta_2^{t-s}\|\mathbf{g}_s\|_2^2}}\right], \quad (8)$$

where $\text{Clip}_D[\mathbf{a}] = \min\{\|\mathbf{a}\|_2, D\} \cdot \mathbf{a}/\|\mathbf{a}\|_2$ is the clipping operator. Compared with Ahn & Cutkosky (2024), our algorithm and analysis do not require $\beta_2 = \beta_1^2$. The numerator and denominator take the standard momentum forms used in Adam. Compared to the original Adam (Kingma & Ba, 2015), we propose to drop the corrective terms to keep the analysis focused on the main effects of momentum and adaptive scaling (Défossez et al., 2022; Wang et al., 2023; Ahn & Cutkosky, 2024). These correction factors mainly affect the early iterations and decay exponentially over time.

#### 4.3.1. A RELAXED CONDITION ON $\beta_2$

For clipped Adam, we first present the following theorem with the proof deferred to Appendix D.6.

**Theorem 4.** *Under Assumptions 1 – 4, run Algorithm 1 with $\beta = \beta_1$ and the discounted FTRL update in Eq. (5) on $\mathcal{D} = \{\Delta : \|\Delta\|_2 \leq D\}$, using $\ell_t(\Delta) = \langle \mathbf{g}_t, \Delta \rangle$ and $\eta_t$ in Eq. (7). This implies the clipped Adam update rule in Eq. (8). Set $1 - (\frac{\epsilon}{16(G+\sigma)})^2 \leq \beta_1 < 1$, $D = \frac{(1-\beta_1)\sqrt{\epsilon}}{\sqrt{48c}}$, $\gamma = \frac{\beta_1 D}{\sqrt{1-\beta_1}}$, $0 < \nu \leq G + \sigma$ and $\max\{1 - \frac{\nu}{G+\sigma}, \beta_1^4\} \leq \beta_2 < 1$. If $T$ is at least*

$$\max\left\{\frac{1}{1-\beta_1}\max\left\{\frac{16F^*\sqrt{48c}}{\epsilon^{3/2}}, \frac{16(G+\sigma)}{\epsilon}\right\}, \frac{\ln 2}{1-\beta_2}\right\},$$

*then the output $\bar{\mathbf{x}}$ of Algorithm 1 satisfies $\mathbb{E}[\|\nabla F(\bar{\mathbf{x}})\|_c] \leq \epsilon$.*

This theorem provides a convergence guarantee for the exponential moving average (EMA) iterate $\bar{\mathbf{x}}$ (Ruppert, 1988; Polyak & Juditsky, 1992; Ahn & Cutkosky, 2024) returned by exponentiated O2NC when the update directions are generated by clipped Adam. The theorem guarantees an iteration complexity of order

$$\mathcal{O}(\max\{(G+\sigma)^2 F^* c^{1/2}\epsilon^{-7/2}, (G+\sigma)^3\epsilon^{-3}, (G+\sigma)\nu^{-1}\}).$$

The last term dominates only when $\nu$ is sufficiently small, when $\nu \leq \Theta(\min\{\epsilon^{7/2}/(GF^*c^{1/2}), \epsilon^3/G^2\})$. In most regimes, the leading term matches the lower bound $\Omega(F^*G^2c^{1/2}\epsilon^{-7/2})$ for finding a $(c, \epsilon)$-stationary point (Zhang & Cutkosky, 2024).

Existing convergence analyses typically impose $\beta_2 \geq \beta_1^2$ (Luo et al., 2019; Zhang et al., 2022b; Défossez et al., 2022). We view this as the essential condition, especially for clip-free Adam. Under clipping, the condition $\beta_2 \geq 1 - \frac{\nu}{G+\sigma}$ in Theorem 4 is needed to achieve the optimal rate; if one tolerates an additional $(G+\sigma)\nu^{-2}$ factor in the iteration complexity (by choosing $\beta_1 = \Theta(1 - \frac{(\epsilon\nu)^2}{(G+\sigma)^3})$), the condition on $\beta_2$ reduces to $\beta_2 \geq \beta_1^4$, removing the explicit dependence on $\nu$. We remark this relaxation is better understood as a theoretical signal that clipping can weaken the coupling between $\beta_1$ and $\beta_2$ by bounding $\|\Delta_t\|_2 \leq D$.

Lemma 8 (the O2NC conversion) reduces the convergence analysis to controlling a dynamic regret. Applying the modular D2D conversion (Theorem 1) together with the discounted-FTRL regret bounds, the key dynamic regret terms can be decomposed into three explicit components:

$$\frac{1}{DT}\mathbb{E}\left[\frac{1-\beta_1}{\beta_1}\gamma\sum_{t=1}^T \alpha_{t-1}\|\mathbf{g}_t\|_2^2 \right. \quad (9)$$

$$+ \frac{1}{\gamma(1-\beta_1)}\sum_{t=1}^T\left(\frac{\beta_1}{2\alpha_{t-1}}\|\Delta_{t+1}\|_2^2 - \frac{1}{2\alpha_t}\|\Delta_{t+1}\|_2^2\right)$$

$$\left. + \frac{\beta_1}{\gamma(1-\beta_1)}\sum_{t=1}^{T-1}\left(\frac{1}{2\alpha_t}\|\mathbf{u}_{t+1}\|_2^2 - \frac{1}{2\alpha_{t-1}}\|\mathbf{u}_t\|_2^2\right)\right],$$

where $\alpha_t = 1/(\nu + \sqrt{(1-\beta_2)\sum_{s=1}^t \beta_2^{t-s}\|\mathbf{g}_s\|_2^2})$, and $\{\mathbf{u}_t\}_{t=1}^T \subset \mathbb{R}^d$ is the comparator sequence with $\|\mathbf{u}_t\|_2 = D$ introduced in the analysis within the O2NC conversion.

The first term is known as the stability term in online learning. We analyze it using a new self-confident tuning lemma (Lemma 9), which gives the following upper bound $\mathcal{O}(\sqrt{1-\beta_1}(1-\beta_2)\frac{(G+\sigma)^2}{\nu})$. To obtain the desirable order $\mathcal{O}(\sqrt{1-\beta_1}(G+\sigma))$, it is required to tune $\beta_2 \geq 1 - \frac{\nu}{G+\sigma}$. Hence the condition on $\beta_2$ involves $\nu$.

For the second term in (9), the key observation is that clipping enforces $\|\Delta_t\|_2 \leq D$, and in the tuning process $D$ is small, which allows a weaker requirement on $\beta_2$ rather than tuning $\beta_2 \geq \beta_1^2$ and discarding all the terms. By Lemma 11, this term can be bounded by $\frac{[\beta_1 - \sqrt{\beta_2}]_+}{\beta_1\sqrt{1-\beta_1}}(G+\sigma)$. To match the target order, it suffices to require $[\beta_1 - \sqrt{\beta_2}]_+ \leq \beta_1(1-\beta_1)$, which implies that $\beta_2 \geq \beta_1^4$. For the last term in (9), using $\|\mathbf{u}_t\|_2 = D$ for all $t$ makes the sum telescoping, and it can be bounded using Lemma 12.

### 4.3.2. A MARGINAL CONDITION ON $\beta_2$

Theorem 4 involves the parameter $\nu$ in the choice of $\beta_2$. Intuitively, this comes from a worst-case analysis on the first term in Eq. (9). That term is in the form of $\alpha_{t-1}\|\mathbf{g}_t\|_2^2$. Since the learning rate $\alpha_{t-1}$ is one step behind, the analysis cannot directly pair $\alpha_{t-1}$ with the current gradient $\mathbf{g}_t$, and in the worst case we can only upper bound $\alpha_{t-1}\|\mathbf{g}_t\|_2^2 \leq \|\mathbf{g}_t\|_2^2/\nu$, requiring tuning $\beta_2$ to absorb the additional factor $G/\nu$ to ensure the optimal convergence rates.

Using ideas from scale-free online learning (Orabona & Pál, 2018), we can refine this step by rewriting the bound in terms of $\alpha_t\|\mathbf{g}_t\|_2^2$ plus an additional correction term that depends on the change of the step size, $\mathcal{O}(|\eta_t - \eta_{t-1}| \cdot \|\sum_{s=1}^{t-1}\beta_1^{t-1-s}\mathbf{g}_s\|_2)$. The benefit is that $\alpha_t$ is involved with $\mathbf{g}_t$. Intuitively, since $\alpha_t \leq 1/(\nu + \sqrt{(1-\beta_2)\|\mathbf{g}_t\|_2^2})$, we can have $\alpha_t\|\mathbf{g}_t\|_2^2 \leq \|\mathbf{g}_t\|_2/\sqrt{1-\beta_2}$, which avoids the dominance of $\nu$. This refinement is not for free: the extra correction term must be analyzed carefully, and it leads to a margin-style condition in the optimal convergence rate.

**Theorem 5.** *Under Assumptions 1 – 4, run Algorithm 1 with $\beta = \beta_1$ and the discounted FTRL update in Eq. (5) on $\mathcal{D} = \{\Delta : \|\Delta\|_2 \leq D\}$, using $\ell_t(\Delta) = \langle\mathbf{g}_t, \Delta\rangle$ and $\eta_t$ in Eq. (7). This implies the clipped Adam update rule in Eq. (8). Choose any $\rho \in [0,1)$ and any $\nu \in (0, G+\sigma]$. Tune $1 - \left(\frac{\epsilon\sqrt{1-\rho^2}}{64(G+\sigma)}\right)^2 \leq \beta_1 < 1$. Define the margin $m = \frac{1-\rho}{2}(1-\beta_1^2)$, and choose $\beta_2$ such that $\beta_2 \in [\beta_1^2+m,\ 1-m]$. Set $D = \frac{(1-\beta_1)\sqrt{\epsilon}}{\sqrt{48c}}, \gamma = \frac{\beta_1 D}{\sqrt{1-\beta_1}}$. If $T$ satisfies*

$$T \geq \max\left\{\frac{1}{1-\beta_1}\cdot\max\left\{\frac{32F^*\sqrt{c}}{\epsilon^{3/2}},\ \frac{16(G+\sigma)}{\epsilon}\right\},\right.$$

$$\left.\frac{32G}{\epsilon\sqrt{1-\beta_1}\sqrt{1-\rho^2}}\cdot\ln\left(1+\frac{G}{\nu}\right),\ \frac{\ln 2}{1-\beta_2}\right\},$$

*then the output $\bar{\mathbf{x}}$ of Algorithm 1 satisfies $\mathbb{E}[\|\nabla F(\bar{\mathbf{x}})\|_c] \leq \epsilon$.*

The proof is deferred to Appendix D.7. Theorem 5 conveys that we require $\beta_2$ to stay away from both $\beta_1^2$ and 1 by a margin $m$. Concretely, the condition $\beta_2 \in [\beta_1^2 + m, 1-m]$ implies $\beta_2 > \beta_1^2$, and it is centered at $(1+\beta_1^2)/2$. The parameter $\rho$ controls the width of the valid range: smaller $\rho$ forces $\beta_2$ closer to $(1+\beta_1^2)/2$, and $\rho = 0$ gives the tightest but most restrictive choice $\beta_2 = (1+\beta_1^2)/2$. For other choices of $\beta_2$, the theorem quantifies the effect through $\rho$, giving a trade-off between a wider admissible range and a looser bound. Appendix D.2 lists common choices of $(\beta_1, \beta_2)$ to illustrate how different settings affect $\rho$.

The convergence rate of Theorem 5 is

$$\mathcal{O}\left(\max\left\{\frac{(G+\sigma)^2F^*c^{1/2}}{(1-\rho)\epsilon^{7/2}},\ \frac{(G+\sigma)^3}{(1-\rho)\epsilon^3},\ \frac{G^2\ln\left(\frac{G}{\nu}\right)}{(1-\rho)\epsilon^2}\right\}\right),$$

where the gap $\beta_2 - \beta_1^2$ can be quantified by the choice of $\rho$ in the final rate. Treating $1 - \rho$ as a constant, Theorem 5 matches previous optimal convergence rates as long as $G/\nu \leq \mathcal{O}(\exp(G/\epsilon))$, which is a mild condition. This type of result is obtained by conducting a refined analysis following the spirit of scale-free online learning (Orabona & Pál, 2018), where we summarize this technique for FTRL with general, time-varying learning rates in Lemma 21. The crux is to carefully control the rate of the deviation $|\eta_t - \eta_{t-1}| \cdot \|\sum_{s=1}^{t-1}\beta_1^{t-1-s}\mathbf{g}_s\|_2$, resulting in the non-dominated rate in the final result.

### 4.4. Results for the Clip-Free Adam Variant

The preceding results show that clipping could relax the coupling between $\beta_1$ and $\beta_2$ in the convergence conditions. Since clipping may not always be preferred in practice, we next study a clip-free variant that replaces clipping with an additional damping term in the step size.

Within the exponentiated O2NC framework, the clip-free case can be handled by letting the online learner minimize a composite loss (Zhang et al., 2024). The main idea is that, the O2NC analysis yields $\mathbb{E}[\|\nabla F(\bar{\mathbf{x}})\|_c] \lesssim \mathbb{E}[\sum_{t=1}^T \langle\mathbf{g}_t, \Delta_t - \mathbf{u}_t\rangle] + c\mu\mathbb{E}[\sum_{t=1}^T \|\Delta_t\|_2^2]$. When clipping is present, then $\|\Delta_t\|_2 \leq D$ and the second term is immediately bounded. Without clipping, Zhang et al. (2024) propose to absorb this quadratic term into the regret via a standard change-of-measure technique (Foster et al., 2020; Chen et al., 2021): $\mathbb{E}[\sum_{t=1}^T \langle\mathbf{g}_t, \Delta_t - \mathbf{u}_t\rangle] + c\mu\mathbb{E}[\sum_{t=1}^T \|\Delta_t\|_2^2] = \mathbb{E}[\text{D-REG}'_T(\mathbf{u}_{1:T})] + c\mu\mathbb{E}[\sum_{t=1}^T \|\mathbf{u}_t\|_2^2]$, where D-REG$'_T$ is defined using the composite loss $\langle\mathbf{g}_t, \Delta\rangle + \mu\|\Delta\|_2^2$. Since the comparator in the analysis is chosen such that $\|\mathbf{u}_t\|_2 = D$, the term $c\mu\mathbb{E}[\sum_{t=1}^T \|\mathbf{u}_t\|_2^2]$ can be bounded by $\mathcal{O}\left(c\mu TD^2\right)$.

Motivated by this, we set the surrogate loss in Algorithm 1 as $\ell_t(\Delta) = \langle\mathbf{g}_t, \Delta\rangle + \frac{\mu}{2}\|\Delta\|_2^2$ with $\mu > 0$ to be tuned, and take $\mathcal{D} = \mathbb{R}^d$. Substituting this choice into Eq. (5) with $\eta_t$ in Eq. (7), Lemma 4 gives the closed-form update of $\Delta_{t+1}$:

$$-\frac{\gamma(1-\beta_1)\sum_{s=1}^t \beta_1^{t-s}\mathbf{g}_s}{\nu + \gamma\mu(1-\beta_1^t) + \sqrt{(1-\beta_2)\sum_{s=1}^t \beta_2^{t-s}\|\mathbf{g}_s\|_2^2}}. \quad (10)$$

The resulting algorithm is an approximation of the original Adam, with an extra damping term $\gamma\mu(1-\beta_1^t)$ in the denominator. As $t$ grows, $\gamma\mu(1-\beta_1^t)$ quickly reaches $\mathcal{O}(\gamma\mu)$. Under our tuning, $\gamma\mu = \mathcal{O}(G+\sigma)$, so in the worst case this term is of the same order as the second-moment term.

The clip-free Adam satisfies the following theorem, and the proof can be found in Appendix D.8.

**Theorem 6.** *Under Assumptions 1 – 4, run Algorithm 1 with $\beta = \beta_1$ and the clip-free Adam update in Eq. (10), using $\eta_t$ in Eq. (7). Set $1 - \left(\frac{\epsilon}{16(G+\sigma)}\right)^2 \leq \beta_1 < 1, \gamma = \frac{\beta_1\sqrt{(1-\beta_1)\epsilon}}{\sqrt{96c}},$ $\mu = \frac{\sqrt{6c\epsilon}}{1-\beta_1}, 0 < \nu \leq G+\sigma$, and $\max\{1-\frac{\nu}{G+\sigma}, \beta_1^2\} \leq$*

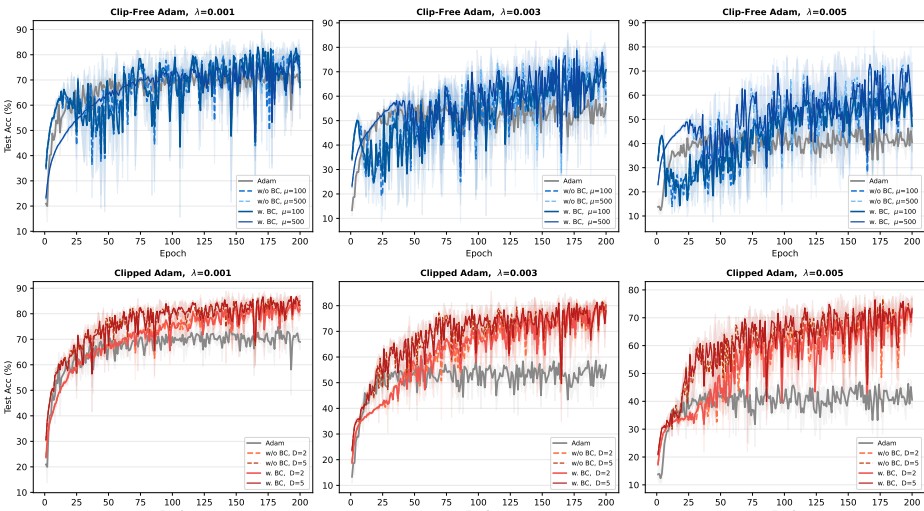

*Figure 1.* Clipped Adam and clip-free Adam with random scaling on ResNet-18 and CIFAR-10 using cross-entropy loss with $\lambda\ell_1$ regularization over three random seeds. Dashed and solid lines correspond to variants without and with bias correction, respectively.

$\beta_2 < 1$. *If $T$ is at least*

$$\max\left\{\frac{1}{1-\beta_1}\max\left\{\frac{16F^*\sqrt{96c}}{\epsilon^{3/2}}, \frac{48(G+\sigma)}{\epsilon}\right\}, \frac{\ln 2}{1-\beta_2}\right\},$$

*then the output $\bar{\mathbf{x}}$ of Algorithm 1 satisfies $\mathbb{E}[\|\nabla F(\bar{\mathbf{x}})\|_c] \leq \epsilon$.*

The proof of Theorem 6 follows the same main steps as Theorem 4. The key difference is the second term in Eq. (9): without clipping, $\|\Delta_t\|_2^2$ cannot be bounded in the desired order, so we require $\frac{\beta_1}{2\alpha_{t-1}} - \frac{1}{2\alpha_t} \leq 0$ to keep this term non-positive, which leads to the condition $\beta_2 \geq \beta_1^2$.

We also provide a margin-style convergence condition for clip-free Adam, thanks to the flexibility of the modular D2D analysis. The proof is deferred to Appendix D.9.

**Theorem 7.** *Under Assumptions 1 − 4, run Algorithm 1 with $\beta = \beta_1$ and the clip-free Adam update in Eq. (10), using $\eta_t$ in Eq. (7). Fix any $\rho \in [0,1)$ and any $\nu \in (0, G+\sigma)$. Choose $\beta_1$ such that $1 - \left(\frac{\epsilon\sqrt{1-\rho^2}}{64(G+\sigma)}\right)^2 \leq \beta_1 < 1$. Define the margin $m = \frac{1-\rho}{2}(1-\beta_1^2)$ and choose $\beta_2 \in [\beta_1^2+m, 1-m]$. Set $\gamma = \frac{\beta_1\sqrt{(1-\beta_1)\epsilon}}{\sqrt{96c}}, \mu = \frac{\sqrt{6c\epsilon}}{1-\beta_1}$. If $T$ satisfies*

$$T \geq \max\left\{\frac{1}{1-\beta_1}\max\left\{\frac{32F^*\sqrt{96c}}{\epsilon^{3/2}}, \frac{48(G+\sigma)}{\epsilon}\right\},\right.$$

$$\left.\frac{\ln 2}{1-\beta_2}, \frac{32G}{\epsilon\sqrt{1-\beta_1}\sqrt{1-\rho^2}}\cdot\ln\left(1+\frac{\gamma\mu+G}{\nu}\right)\right\},$$

*then the output $\bar{\mathbf{x}}$ of Algorithm 1 satisfies $\mathbb{E}[\|\nabla F(\bar{\mathbf{x}})\|_c] \leq \epsilon$.*

### 4.5. More Implications

By the generality of $(c,\epsilon)$-stationarity (Zhang & Cutkosky, 2024), in Appendix D.3, we demonstrate how our theorems

specialize to stochastic, non-convex and smooth settings. By setting $c$ with smoothness constants properly, our results can imply the optimal convergence rates to the first-order stationary points for non-convex smooth optimization.

Our results can also be extended to a coordinate-wise analysis, yielding convergence rates in $\ell_1$-norm via the conversion in Appendix G of Cutkosky et al. (2023).

### 4.6. Experiments

We evaluate clipped Adam and clip-free Adam on ResNet-18 (He et al., 2016) trained on CIFAR-10 (Krizhevsky & Hinton, 2009) with a $\lambda\ell_1$ regularizer, where larger $\lambda$ introduces stronger non-smoothness. All variants use the same learning rate lr = 0.001 and $(\beta_1, \beta_2) = (0.9, 0.999)$ as default Adam. As shown in Figure 1, both clipped and clip-free variants consistently outperform default Adam, especially when non-smoothness increases, validating the algorithmic modifications motivated by our theory. Moreover, bias correction only affects early-stage behavior without altering final convergence, justifying its omission in our analysis.

## 5. Conclusion

We study dynamic regret via a modular D2D reduction. We streamline the analyses for linear regression, obtaining new guarantees for logistic regression. Beyond OCO, combining this reduction with the exponentiated O2NC framework leads to optimal convergence rates for both clipped and clip-free Adam in stochastic non-convex and non-smooth settings, and admits more flexible parameter choices $(\beta_1, \beta_2)$. Future directions include extending the approach to more sophisticated FTRL variants (Jézéquel et al., 2025) and obtaining the guarantees of Adam under relaxed assumptions.

## Acknowledgements

This work was supported by National Science and Technology Major Project (2022ZD0114800), NSFC (62361146852), the Fundamental and Interdisciplinary Disciplines Breakthrough Plan of the Ministry of Education of China (No. JYB2025XDXM118), and the "111 Center" (No. B26023). Yu-Jie Zhang was supported in part by the Singapore National Research Foundation (NRF) under its AI Visiting Professorship programme and NSF Award TRIPODS 202323.

## Impact Statement

This paper presents work whose goal is to advance the field of machine learning. There are many potential societal consequences of our work, none of which we feel must be specifically highlighted here.

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

# A. Related Work

**Non-stationary Online Learning.** In online learning, data arrives sequentially and the underlying environment may change over time, making non-stationarity a natural concern. A common way to model such drift is to strengthen the benchmark beyond static regret, leading to notions such as interval regret (Hazan & Seshadhri, 2007; Daniely et al., 2015; Jun et al., 2017), switching regret (Herbster & Warmuth, 1998; György & Szepesvári, 2016), and dynamic regret (Zinkevich, 2003; Cesa-Bianchi et al., 2012; Hall & Willett, 2013). In this paper, we focus on dynamic regret, which benchmarks the learner against a time-varying comparator sequence and has received extensive study in recent years (Zhang et al., 2018; Zhao et al., 2020; Cutkosky, 2020; Zhao & Zhang, 2021; Zhang et al., 2022a; Baby & Wang, 2021; Bai et al., 2022; Baby & Wang, 2022; Jacobsen & Cutkosky, 2022; Hu et al., 2023; Qian et al., 2024; Li et al., 2024; Zhao et al., 2024; 2025).

Obtaining dynamic regret guarantees becomes more delicate for curved losses: even when a drift measure such as the path length is available, tuning learning rates efficiently while keeping sharp dependence on $(T, P_T)$ can be challenging (Zhang et al., 2025). A general approach is to convert interval regret guarantees into dynamic regret bounds (Baby & Wang, 2021; 2022). These analyses typically rely on the Karush–Kuhn–Tucker conditions of comparators and can be technically involved, which may limit flexibility when extended to broader settings. Zhang et al. (2025) leverage mixability (Vovk, 2001; van Erven et al., 2012) and exponential-weights methods to obtain sharp trade-offs, at the expense of higher computational costs. Yuan & Lamperski (2020) analyze and present dynamic regret bounds using discounted algorithms, while their methods are restricted to a bounded domain. Closest to our linear regression application, Jacobsen & Cutkosky (2024) analyze discounted VAW and provide matching lower bounds in the unconstrained setting with Bregman-divergence-based arguments. Relative to these works, our contribution is a modular discounted-to-dynamic reduction that presents a more direct analysis and can recover optimal guarantees, while enabling new guarantees for curved losses on unbounded domains.

**Discounted Online Learning and D2D Reduction.** Discounted regret has been used as one of the important ways to emphasize recent losses and adapt to drifting environments (Freund & Hsu, 2008; Chernov & Zhdanov, 2010; Cesa-Bianchi et al., 2012; Krichene et al., 2014). In OCO, Zhang et al. (2024) develop gradient-adaptive guarantees for given discount factors, and subsequent work studies adaptation when the discount factor is unknown (Yang et al., 2026). Recently, Nguyen (2026) investigate the choice of $(\beta_1, \beta_2)$ through discounted-regret analysis of an FTRL method derived from a simplified Adam variant in OCO. Compared to this work, our focus is on stochastic, non-convex, and non-smooth optimization via O2NC, and we analyze Adam with the standard EMA momentum form, deriving convergence guarantees under flexible $(\beta_1, \beta_2)$ regimes. Another related direction to make FTRL more responsive to recent changes is to prune the history. In particular, Mhaisen & Iosifidis (2025) retain the core framework of FTRL while adding a correction term to the input gradient, which can be viewed as pruning the influence of past history, and achieve optimal dynamic regret guarantees. Compared to their work, our analysis applies to curved losses, even in unconstrained settings. As a limitation, it remains unclear how to use our approach to recover optimal dynamic regret bounds for general convex losses.

Bridging discounted regret and dynamic regret, Ahn et al. (2024) establish an explicit connection between these two measures. In different approaches, reductions that relate dynamic and static regret by working in extended spaces have also been explored (Jacobsen & Orabona, 2024; Jacobsen et al., 2025). Building on the discounted-to-dynamic reduction, our work develops a modular analysis that keeps key terms explicit and supports reusable analyses across multiple settings, including both convex problems and non-convex optimization through the O2NC conversion.

**Convergence of Adam.** Adam is introduced by Kingma & Ba (2015) with guarantees in the convex setting, and its convergence rates have since been studied extensively for convex and smooth non-convex objectives (Reddi et al., 2018; Zhou et al., 2019; Zou et al., 2019; Alacaoglu et al., 2020; Guo et al., 2021; Zhang et al., 2022b; Défossez et al., 2022; Li et al., 2023; Wang et al., 2023). In parallel, another line of work seeks to understand the stability and dynamics of Adam through the lens of loss landscape geometry (Li et al., 2025). Our work follows the former direction and focuses on the convergence rates of Adam to stationary points in the non-smooth setting.

We focus on Adam through the O2NC framework to derive guarantees in non-smooth settings. The non-smooth regime presents challenges for the analysis of Adam: Xiao et al. (2024) establish convergence of Adam-family methods to stationary points in non-smooth neural network training, which requires diminishing stepsizes and time-varying momentum coefficients $\beta_1, \beta_2 \to 1$. Ahn & Cutkosky (2024) analyze clipped Adam for exponential moving average (EMA) iterates in stochastic, non-convex, and non-smooth settings. Their update uses a slightly different momentum normalization (e.g., $m_t = \beta m_{t-1} + \mathbf{g}_t$ rather than the standard EMA form $m_t = \beta m_{t-1} + (1 - \beta)\mathbf{g}_t$), and their theory focuses on the restricted parameter choice $\beta_2 = \beta_1^2$. These results highlight the value of carefully designed variants of Adam in the non-smooth

regime, which motivates our study of clipped Adam and a clip-free variant with an additional damping term (Section 4). Our guarantees apply to the EMA iterate $\bar{\mathbf{x}}$ returned by the O2NC framework, rather than the per-round iterate $\mathbf{x}_t$. To obtain more flexible conditions on $(\beta_1, \beta_2)$, we additionally assume bounded stochastic gradients (Assumption 4), a condition also adopted in related analyses (Cutkosky & Orabona, 2019; Guo et al., 2021; Défossez et al., 2022); the only prior result without this assumption (Ahn & Cutkosky, 2024) is restricted to $\beta_2 = \beta_1^2$. Under this trade-off, our results provide optimal convergence rates to $(c, \epsilon)$-stationary points for both clipped and clip-free Adam variants under flexible $(\beta_1, \beta_2)$, and by appropriate choices of $c$, recover optimal rates for smooth and second-order smooth settings.

**O2NC Conversion.** Motivated by tractable stationary notions for non-smooth objectives (Zhang et al., 2020), there has been growing interest in non-smooth non-convex optimization (Davis et al., 2022; Tian et al., 2022; Jordan et al., 2023). The seminal work by Cutkosky et al. (2023) proposes the online-to-non-convex (O2NC) framework, reducing stochastic non-convex and non-smooth optimization to an online learning problem and achieving optimal rates under standard assumptions. Zhang & Cutkosky (2024) extend O2NC by introducing $(c, \epsilon)$-stationarity and allowing unbounded directions by minimizing composite losses. We build on this exponentiated O2NC framework, and through our modular D2D analysis, derive convergence guarantees for the EMA iterates of clipped and clip-free Adam variants.

## B. Omitted Details for Section 2

In this part, we provide the proofs of the presented theoretical results in Section 2.

### B.1. Proof of Lemma 1

For completeness, we include a proof of Lemma 1, adapted from Theorem B.3 in Ahn et al. (2024).

*Proof.* Recall the discounted regret up to time $t$ against a fixed comparator $\mathbf{u} \in \mathcal{X}$:

$$\mathrm{REG}_{t;\beta}(\mathbf{u}) = \sum_{s=1}^{t} \beta^{t-s}\big(f_s(\mathbf{x}_s) - f_s(\mathbf{u})\big), \qquad \text{with the convention } \mathrm{REG}_{0;\beta}(\mathbf{u}) = 0.$$

A key identity is that, for every $t \geq 1$ and any $\mathbf{u} \in \mathcal{X}$,

$$\begin{aligned} f_t(\mathbf{x}_t) - f_t(\mathbf{u}) &= \mathrm{REG}_{t;\beta}(\mathbf{u}) - \beta \mathrm{REG}_{t-1;\beta}(\mathbf{u}) \\ &= (1-\beta)\mathrm{REG}_{t;\beta}(\mathbf{u}) + \beta\big(\mathrm{REG}_{t;\beta}(\mathbf{u}) - \mathrm{REG}_{t-1;\beta}(\mathbf{u})\big). \end{aligned} \tag{11}$$

Indeed, expanding the definition gives

$$\mathrm{REG}_{t;\beta}(\mathbf{u}) - \beta \mathrm{REG}_{t-1;\beta}(\mathbf{u}) = \sum_{s=1}^{t} \beta^{t-s}(f_s(\mathbf{x}_s) - f_s(\mathbf{u})) - \sum_{s=1}^{t-1} \beta^{t-s}(f_s(\mathbf{x}_s) - f_s(\mathbf{u})) = f_t(\mathbf{x}_t) - f_t(\mathbf{u}).$$

Now sum Eq. (11) over $t = 1, \ldots, T$ with the time-varying comparator $\mathbf{u} = \mathbf{u}_t$:

$$\sum_{t=1}^{T} \big(f_t(\mathbf{x}_t) - f_t(\mathbf{u}_t)\big) = (1-\beta)\sum_{t=1}^{T} \mathrm{REG}_{t;\beta}(\mathbf{u}_t) + \beta \sum_{t=1}^{T} \big(\mathrm{REG}_{t;\beta}(\mathbf{u}_t) - \mathrm{REG}_{t-1;\beta}(\mathbf{u}_t)\big).$$

It remains to rewrite the second summation via telescoping. Observe that

$$\sum_{t=1}^{T} \big(\mathrm{REG}_{t;\beta}(\mathbf{u}_t) - \mathrm{REG}_{t-1;\beta}(\mathbf{u}_t)\big) = \mathrm{REG}_{T;\beta}(\mathbf{u}_T) + \sum_{t=1}^{T-1} \big(\mathrm{REG}_{t;\beta}(\mathbf{u}_t) - \mathrm{REG}_{t;\beta}(\mathbf{u}_{t+1})\big),$$

where we use $\mathrm{REG}_{0;\beta}(\mathbf{u}_1) = 0$. Plugging this into the previous equation finishes the proof. $\square$

## B.2. Proof of Theorem 1

*Proof.* By Lemma 1, we have

$$\sum_{t=1}^{T} f_t(\mathbf{x}_t) - \sum_{t=1}^{T} f_t(\mathbf{u}_t) = (1-\beta)\sum_{t=1}^{T} \text{REG}_{t;\beta}(\mathbf{u}_t) + \beta\sum_{t=1}^{T-1}\Big(\text{REG}_{t;\beta}(\mathbf{u}_t) - \text{REG}_{t;\beta}(\mathbf{u}_{t+1})\Big) + \beta\text{REG}_{T;\beta}(\mathbf{u}_T).$$

For any $t \in [T]$ and any fixed $\mathbf{u} \in \mathcal{X}$, multiplying the assumed rescaled bound by $\beta^t$ gives

$$\text{REG}_{t;\beta}(\mathbf{u}) = \sum_{s=1}^{t}\beta^{t-s}\big(f_s(\mathbf{x}_s) - f_s(\mathbf{u})\big) \le \beta^t\varphi_t(\mathbf{u}) + \sum_{s=1}^{t}\beta^t\Lambda_s.$$

Applying it with $\mathbf{u} = \mathbf{u}_t$ and summing provides

$$(1-\beta)\sum_{t=1}^{T}\text{REG}_{t;\beta}(\mathbf{u}_t) \le (1-\beta)\sum_{t=1}^{T}\Big(\beta^t\varphi_t(\mathbf{u}_t) + \sum_{s=1}^{t}\beta^t\Lambda_s\Big)$$

$$= (1-\beta)\sum_{t=1}^{T}\beta^t\varphi_t(\mathbf{u}_t) + (1-\beta)\sum_{t=1}^{T}\sum_{s=1}^{t}\beta^t\Lambda_s = (1-\beta)\sum_{t=1}^{T}\beta^t\varphi_t(\mathbf{u}_t) + (1-\beta)\sum_{s=1}^{T}\Lambda_s\sum_{t=s}^{T}\beta^t$$

$$= (1-\beta)\sum_{t=1}^{T}\beta^t\varphi_t(\mathbf{u}_t) + (1-\beta)\sum_{s=1}^{T}\Lambda_s \cdot \frac{\beta^s(1-\beta^{T-s+1})}{1-\beta} = (1-\beta)\sum_{t=1}^{T}\beta^t\varphi_t(\mathbf{u}_t) + \sum_{s=1}^{T}\beta^s\Lambda_s - \sum_{s=1}^{T}\beta^{T+1}\Lambda_s,$$

where the negative term $-\beta^{T+1}\sum_{s=1}^{T}\Lambda_s$ is saved for later use. Next, by the definition of $\text{REG}_{t;\beta}(\cdot)$,

$$\beta\sum_{t=1}^{T-1}\Big(\text{REG}_{t;\beta}(\mathbf{u}_t) - \text{REG}_{t;\beta}(\mathbf{u}_{t+1})\Big) = \beta\sum_{t=1}^{T-1}\sum_{s=1}^{t}\beta^{t-s}\Big(f_s(\mathbf{u}_{t+1}) - f_s(\mathbf{u}_t)\Big).$$

We now combine the above equation with the term $(1-\beta)\sum_{t=1}^{T}\beta^t\varphi_t(\mathbf{u}_t)$. Using $(1-\beta)\beta^t = \beta^t - \beta^{t+1}$, we write

$$(1-\beta)\sum_{t=1}^{T}\beta^t\varphi_t(\mathbf{u}_t) = \beta\varphi_1(\mathbf{u}_1) - \beta^{T+1}\varphi_T(\mathbf{u}_T) + \sum_{t=1}^{T-1}\Big(\beta^{t+1}\varphi_{t+1}(\mathbf{u}_{t+1}) - \beta^{t+1}\varphi_t(\mathbf{u}_t)\Big)$$

$$= \beta\varphi_1(\mathbf{u}_1) - \beta^{T+1}\varphi_T(\mathbf{u}_T) + \beta\sum_{t=1}^{T-1}\Big(\beta^t\varphi_t(\mathbf{u}_{t+1}) - \beta^t\varphi_t(\mathbf{u}_t)\Big) + \beta\sum_{t=1}^{T-1}\beta^t\Big(\varphi_{t+1}(\mathbf{u}_{t+1}) - \varphi_t(\mathbf{u}_{t+1})\Big),$$

where the negative term $-\beta^{T+1}\varphi_T(\mathbf{u}_T)$ is useful and we apply the identity

$$\beta^{t+1}\varphi_{t+1}(\mathbf{u}_{t+1}) = \beta \cdot \beta^t\varphi_t(\mathbf{u}_{t+1}) + \beta \cdot \beta^t\Big(\varphi_{t+1}(\mathbf{u}_{t+1}) - \varphi_t(\mathbf{u}_{t+1})\Big).$$

Define $F_t^{\beta,\varphi}(\mathbf{u}) = \beta^t\varphi_t(\mathbf{u}) + \sum_{s=1}^{t}\beta^{t-s}f_s(\mathbf{u})$. Then

$$\beta\sum_{t=1}^{T-1}\Big(\beta^t\varphi_t(\mathbf{u}_{t+1}) - \beta^t\varphi_t(\mathbf{u}_t)\Big) + \beta\sum_{t=1}^{T-1}\sum_{s=1}^{t}\beta^{t-s}\Big(f_s(\mathbf{u}_{t+1}) - f_s(\mathbf{u}_t)\Big) = \beta\sum_{t=1}^{T-1}\Big(F_t^{\beta,\varphi}(\mathbf{u}_{t+1}) - F_t^{\beta,\varphi}(\mathbf{u}_t)\Big).$$

Finally, applying the bound on $\text{REG}_{T;\beta}(\mathbf{u}_T)$ gives

$$\beta\text{REG}_{T;\beta}(\mathbf{u}_T) \le \beta^{T+1}\varphi_T(\mathbf{u}_T) + \beta^{T+1}\sum_{s=1}^{T}\Lambda_s,$$

which can be cancelled by the negative terms $-\beta^{T+1}\varphi_T(\mathbf{u}_T)$ and $-\beta^{T+1}\sum_{s=1}^{T}\Lambda_s$ above. Putting everything together finishes the proof:

$$\sum_{t=1}^{T} f_t(\mathbf{x}_t) - \sum_{t=1}^{T} f_t(\mathbf{u}_t) \le \beta\varphi_1(\mathbf{u}_1) + \sum_{t=1}^{T}\beta^t\Lambda_t + \beta\sum_{t=1}^{T-1}\Big(F_t^{\beta,\varphi}(\mathbf{u}_{t+1}) - F_t^{\beta,\varphi}(\mathbf{u}_t)\Big)$$

$$+ \beta \sum_{t=1}^{T-1} \beta^t \Big( \varphi_{t+1}(\mathbf{u}_{t+1}) - \varphi_t(\mathbf{u}_{t+1}) \Big).$$

$\square$

## C. Omitted Details for Section 3

We present the omitted details in Section 3, including several omitted key lemmas and discussions.

### C.1. Proof of Online Linear Regression

We first give a formal statement of Theorem 2.

**Theorem 2** (Formal Version). *In unconstrained online linear regression, the discounted VAW forecaster satisfies the following dynamic regret bound: for any $\mathbf{u}_1, \ldots, \mathbf{u}_T \in \mathbb{R}^d$,*

$$\sum_{t=1}^{T} f_t(\mathbf{x}_t) - \sum_{t=1}^{T} f_t(\mathbf{u}_t)$$

$$\leq \frac{\beta \lambda}{2} \|\mathbf{u}_1\|_2^2 + \frac{d}{2} \Big( \max_{t \in [T]} y_t^2 \Big) \cdot \ln \left( 1 + \frac{\sum_{t=1}^{T} \beta^{T-t} \|\mathbf{z}_t\|_2^2}{\lambda d} \right) + \beta \sum_{t=1}^{T-1} \Big( F_t^{\beta, \varphi}(\mathbf{u}_{t+1}) - F_t^{\beta, \varphi}(\mathbf{u}_t) \Big) + \frac{1-\beta}{\beta} \cdot \frac{d}{2} \sum_{t=1}^{T} y_t^2$$

$$\leq \frac{\beta \lambda}{2} \|\mathbf{u}_1\|_2^2 + \frac{d}{2} \Big( \max_{t \in [T]} y_t^2 \Big) \cdot \ln \left( 1 + \frac{\sum_{t=1}^{T} \beta^{T-t} \|\mathbf{z}_t\|_2^2}{\lambda d} \right) + \frac{\gamma}{1-\gamma} P_T^\gamma + \frac{1-\beta}{\beta} \cdot \frac{d}{2} \sum_{t=1}^{T} y_t^2,$$

*where we assume $0 < \beta \leq \gamma < 1$, and $P_T^\gamma$ is defined in Eq. (4). Moreover, there exists a two-layer ensemble algorithm that ensures an $\mathcal{O}\big( d \log T + \sqrt{dT P_T^{\beta_\star}} \big)$ dynamic regret bound, where*

$$\beta_\star = \frac{\sqrt{\frac{d}{2} \sum_{t=1}^{T} y_t^2}}{\sqrt{\frac{d}{2} \sum_{t=1}^{T} y_t^2} + \sqrt{P_T^{\beta_\star}}}.$$

*Proof of Theorem 2.* By reparameterizing as $\widetilde{\mathbf{z}}_t = \beta^{-t/2} \mathbf{z}_t$ and $\widetilde{y}_t = \beta^{-t/2} y_t$, and feeding the VAW forecaster with samples $(\widetilde{\mathbf{z}}_t, \widetilde{y}_t)$, Lemma 2 ensures that for any $\mathbf{u} \in \mathbb{R}^d$,

$$\sum_{s=1}^{t} \beta^{-s} \big( f_s(\mathbf{x}_s) - f_s(\mathbf{u}) \big) = \sum_{s=1}^{t} \big( \ell(\mathbf{x}_s^\top \widetilde{\mathbf{z}}_s, \widetilde{y}_s) - \ell(\mathbf{u}^\top \widetilde{\mathbf{z}}_s, \widetilde{y}_s) \big)$$

$$\leq \frac{\lambda}{2} \|\mathbf{u}\|_2^2 + \sum_{s=1}^{t} \beta^{-2s} y_s^2 \mathbf{z}_s^\top \Big( \lambda I + \sum_{\tau=1}^{s} \beta^{-\tau} \mathbf{z}_\tau \mathbf{z}_\tau^\top \Big)^{-1} \mathbf{z}_s.$$

Choosing $\varphi_t(\mathbf{u}) = \frac{\lambda}{2} \|\mathbf{u}\|_2^2$ and $\Lambda_t = \beta^{-2t} y_t^2 \mathbf{z}_t^\top \Big( \lambda I + \sum_{s=1}^{t} \beta^{-s} \mathbf{z}_s \mathbf{z}_s^\top \Big)^{-1} \mathbf{z}_t$, Theorem 1 gives that,

$$\sum_{t=1}^{T} f_t(\mathbf{x}_t) - \sum_{t=1}^{T} f_t(\mathbf{u}_t)$$

$$\leq \frac{\beta \lambda}{2} \|\mathbf{u}_1\|_2^2 + \sum_{t=1}^{T} \beta^{-t} y_t^2 \mathbf{z}_t^\top \Big( \lambda I + \sum_{s=1}^{t} \beta^{-s} \mathbf{z}_s \mathbf{z}_s^\top \Big)^{-1} \mathbf{z}_t + \beta \sum_{t=1}^{T-1} \Big( F_t^{\beta, \varphi}(\mathbf{u}_{t+1}) - F_t^{\beta, \varphi}(\mathbf{u}_t) \Big)$$

$$= \frac{\beta \lambda}{2} \|\mathbf{u}_1\|_2^2 + \sum_{t=1}^{T} y_t^2 \mathbf{z}_t^\top \Big( \lambda \beta^t I + \sum_{s=1}^{t} \beta^{t-s} \mathbf{z}_s \mathbf{z}_s^\top \Big)^{-1} \mathbf{z}_t + \beta \sum_{t=1}^{T-1} \Big( F_t^{\beta, \varphi}(\mathbf{u}_{t+1}) - F_t^{\beta, \varphi}(\mathbf{u}_t) \Big), \quad (12)$$

where $F_t^{\beta, \varphi}(\mathbf{u}) = \beta^t \varphi_t(\mathbf{u}) + \sum_{s=1}^{t} \beta^{t-s} f_s(\mathbf{u})$.

For the second term in (12), Lemma 25 implies

$$\sum_{t=1}^{T} y_t^2 \mathbf{z}_t^{\top} \Big(\lambda\beta^t I + \sum_{s=1}^{t} \beta^{t-s}\mathbf{z}_s\mathbf{z}_s^{\top}\Big)^{-1}\mathbf{z}_t \leq d\ln\Big(\tfrac{1}{\beta}\Big)\sum_{t=1}^{T} y_t^2 + \big(\max_t y_t^2\big)d\ln\Big(1 + \tfrac{\sum_{t=1}^{T}\beta^{T-t}\|\mathbf{z}_t\|_2^2}{\lambda d}\Big)$$

$$\leq d\frac{1-\beta}{\beta}\sum_{t=1}^{T} y_t^2 + \big(\max_t y_t^2\big)d\ln\Big(1 + \tfrac{\sum_{t=1}^{T}\beta^{T-t}\|\mathbf{z}_t\|_2^2}{\lambda d}\Big),$$

where we use $\ln(1/\beta) \leq (1-\beta)/\beta$.

For the third term in (12), letting $f_0(\mathbf{u}) = \frac{\lambda}{2}\|\mathbf{u}\|_2^2$, by Lemma 18, we have

$$\beta\sum_{t=1}^{T-1}\Big(F_t^{\beta,\varphi}(\mathbf{u}_{t+1}) - F_t^{\beta,\varphi}(\mathbf{u}_t)\Big) \leq \frac{\gamma}{1-\gamma}P_T^{\gamma},$$

Combining the above bounds completes the proof of the first claim. As for the final dynamic regret guarantee, notice that the untuned dynamic regret bound exactly matches Theorem 3.1 in Jacobsen & Cutkosky (2024). Applying Theorem 4.2 in Jacobsen & Cutkosky (2024) to the untuned guarantees can finish the proof. $\square$

## C.2. Proofs of Online Logistic Regression

In this part, we prove Theorem 3. Our main proof steps follow Jézéquel et al. (2020), but we interpret the argument from an optimistic FTRL viewpoint and utilize the modular D2D framework to derive the dynamic regret guarantee.

### C.2.1. PROOF OF THEOREM 3

We prove the following full version of Theorem 3 in this section.

**Theorem 3** (Formal Version). *Assume $\|\mathbf{z}_t\|_2 \leq R$ for all $t \in [T]$. For any comparator sequence $\mathbf{u}_1, \ldots, \mathbf{u}_T$ satisfying $\|\mathbf{u}_t\|_2 \leq B$, the dynamic regret bound of the discounted AIOLI algorithm with parameters $\lambda > 0$ and $\beta \in (0,1)$ is:*

$$\sum_{t=1}^{T} f_t(\mathbf{x}_t) - \sum_{t=1}^{T} f_t(\mathbf{u}_t) \leq \frac{\beta\lambda}{2}\|\mathbf{u}_1\|_2^2 + d(1 + BR)\log\left(1 + \frac{R^2\sum_{t=1}^{T}\beta^{T-t}}{d\lambda(1 + BR)}\right)$$

$$+ \beta\sum_{t=1}^{T-1}\Big(F_t^{\beta,\varphi}(\mathbf{u}_{t+1}) - F_t^{\beta,\varphi}(\mathbf{u}_t)\Big) + \frac{1-\beta}{\beta}\cdot d(1+BR)T$$

$$\leq \frac{\beta\lambda}{2}\|\mathbf{u}_1\|_2^2 + d(1+BR)\log\left(1 + \frac{R^2\sum_{t=1}^{T}\beta^{T-t}}{d\lambda(1+BR)}\right) + \frac{\gamma}{1-\gamma}P_T^{\gamma} + \frac{1-\beta}{\beta}\cdot d(1+BR)T$$

*where we assume $0 < \beta \leq \gamma < 1$, and $P_T^{\gamma}$ is defined in Eq. (4).*

*In Theorem 8, we prove that there exists an online ensemble algorithm that learns the suitable discount factor on the fly and guarantees an $\mathcal{O}(dB\log(BT) + \sqrt{dBTP_T^{\beta_\star}})$ dynamic regret bound where $\beta_\star$ is defined in Eq. (22).*

*Proof.* We apply Lemma 3 and the D2D reduction. Let $\mathbf{u}_1, \ldots, \mathbf{u}_T \in \mathbb{R}^d$ with $\|\mathbf{u}_t\|_2 \leq B$. Lemma 3 gives, for every $t \in [T]$ and any fixed $\mathbf{u}$,

$$\sum_{s=1}^{t} \beta^{-s}\big(f_s(\mathbf{x}_s) - f_s(\mathbf{u})\big) \leq \frac{\lambda}{2}\|\mathbf{u}\|_2^2 + \sum_{s=1}^{t}(1 + BR)\beta^{-2s}\eta_s\mathbf{g}_s^{\top}A_s^{-1}\mathbf{g}_s,$$

where $A_s$ is defined in (14). Thus the assumption of Theorem 1 holds with $\varphi_t(\mathbf{u}) = \frac{\lambda}{2}\|\mathbf{u}\|_2^2$ and $\Lambda_s = (1 + BR)\beta^{-2s}\eta_s\mathbf{g}_s^{\top}A_s^{-1}\mathbf{g}_s$. Since $\varphi_t$ is time-independent, the last term in Theorem 1 vanishes, and we obtain

$$\sum_{t=1}^{T} f_t(\mathbf{x}_t) - \sum_{t=1}^{T} f_t(\mathbf{u}_t) \leq \frac{\beta\lambda}{2}\|\mathbf{u}_1\|_2^2 + (1 + BR)\sum_{t=1}^{T}\beta^t\cdot\beta^{-2t}\eta_t\mathbf{g}_t^{\top}A_t^{-1}\mathbf{g}_t + \beta\sum_{t=1}^{T-1}\Big(F_t^{\beta,\varphi}(\mathbf{u}_{t+1}) - F_t^{\beta,\varphi}(\mathbf{u}_t)\Big)$$

$$= \frac{\beta\lambda}{2}\|\mathbf{u}_1\|_2^2 + (1 + BR)\sum_{t=1}^{T}\eta_t\mathbf{g}_t^\top\widetilde{A}_t^{-1}\mathbf{g}_t + \beta\sum_{t=1}^{T-1}\left(F_t^{\beta,\varphi}(\mathbf{u}_{t+1}) - F_t^{\beta,\varphi}(\mathbf{u}_t)\right),$$

where we used $\widetilde{A}_t = \beta^t A_t = \lambda\beta^t I + \sum_{s=1}^{t}\beta^{t-s}\eta_s\mathbf{g}_s\mathbf{g}_s^\top$, so that $A_t^{-1} = \beta^t\widetilde{A}_t^{-1}$. We bound the stability sum $\sum_{t=1}^{T}\eta_t\mathbf{g}_t^\top\widetilde{A}_t^{-1}\mathbf{g}_t$ via the discounted potential lemma (Lemma 25) by applying it to $\widetilde{\mathbf{g}}_t = \sqrt{\eta_t}\mathbf{g}_t$:

$$\sum_{t=1}^{T}\eta_t\mathbf{g}_t^\top\widetilde{A}_t^{-1}\mathbf{g}_t \leq dT\ln\frac{1}{\beta} + d\ln\left(1 + \frac{\sum_{t=1}^{T}\beta^{T-t}\eta_t\|\mathbf{g}_t\|_2^2}{d\lambda}\right) \leq dT\frac{1-\beta}{\beta} + d\ln\left(1 + \frac{R^2\sum_{t=1}^{T}\beta^{T-t}}{d\lambda(1+BR)}\right),$$

where in the second line we use $\ln(1/\beta) \leq (1-\beta)/\beta$, and $\eta_t\|\mathbf{g}_t\|_2^2 = \|\eta_t\mathbf{g}_t\|_2\|\mathbf{g}_t\|_2 = \frac{1}{1+BR}\|\mathbf{g}_t^{-y_t}\|_2\|\mathbf{g}_t\|_2 \leq \frac{R^2}{1+BR}$. For the path length term, letting $f_0(\mathbf{u}) = \frac{\lambda}{2}\|\mathbf{u}\|_2^2$, following Lemma 18, we conclude,

$$\beta\sum_{t=1}^{T-1}\left(F_t^{\beta,\varphi}(\mathbf{u}_{t+1}) - F_t^{\beta,\varphi}(\mathbf{u}_t)\right) \leq \frac{\gamma}{1-\gamma}P_T^\gamma.$$

The online ensemble algorithm is presented in Algorithm 2, and Theorem 8 presents the corresponding dynamic regret bound of $\mathcal{O}(dB\log(BT) + \sqrt{dBTP_T^{\beta_\star}})$ with $\beta_\star$ defined in Eq. (22). $\qquad\square$

### C.2.2. KEY LEMMA

**Lemma 3.** *Consider online logistic regression with $f_t(\mathbf{x}) = \ell(\mathbf{x}^\top\mathbf{z}_t, y_t) = \ln(1 + \exp(-y_t\mathbf{x}^\top\mathbf{z}_t))$, where $y_t \in \{+1, -1\}$ and $\|\mathbf{z}_t\|_2 \leq R$ for all $t \in [T]$. Assume the comparator satisfies $\|\mathbf{u}\|_2 \leq B$. Let $\beta \in (0,1)$ and $\lambda > 0$. Define the optimistic function $h_t(\mathbf{x}) = \ell(\mathbf{x}^\top\mathbf{z}_t, +1) + \ell(\mathbf{x}^\top\mathbf{z}_t, -1)$, the gradient $\mathbf{g}_t = \nabla f_t(\mathbf{x}_t)$, and the learning rate*

$$\eta_t = \frac{\exp(y_t\mathbf{x}_t^\top\mathbf{z}_t)}{1 + BR}. \tag{13}$$

*Define the surrogate loss $\widehat{f}_t(\mathbf{x}) = f_t(\mathbf{x}_t) + \langle\mathbf{g}_t, \mathbf{x} - \mathbf{x}_t\rangle + \frac{\eta_t}{2}\langle\mathbf{g}_t, \mathbf{x} - \mathbf{x}_t\rangle^2$, and run the discounted AIOLI update (as in Section 3.2):*

$$\mathbf{x}_t \in \arg\min_{\mathbf{x}\in\mathbb{R}^d}\left\{\frac{\lambda\beta^t}{2}\|\mathbf{x}\|_2^2 + h_t(\mathbf{x}) + \sum_{s=1}^{t-1}\beta^{t-s}\widehat{f}_s(\mathbf{x})\right\}.$$

*Let*

$$A_t = \lambda I + \sum_{s=1}^{t}\beta^{-s}\eta_s\mathbf{g}_s\mathbf{g}_s^\top, \qquad t \in [T]. \tag{14}$$

*Then for any $t \in [T]$ and any $\mathbf{u} \in \mathbb{R}^d$ with $\|\mathbf{u}\|_2 \leq B$, discounted AIOLI satisfies the following bound:*

$$\sum_{s=1}^{t}\beta^{-s}\left(f_s(\mathbf{x}_s) - f_s(\mathbf{u})\right) \leq \frac{\lambda}{2}\|\mathbf{u}\|_2^2 + (1 + BR)\sum_{s=1}^{t}\beta^{-2s}\eta_s\mathbf{g}_s^\top A_s^{-1}\mathbf{g}_s.$$

*Proof.* By Lemma 17, for any $\mathbf{u}$ with $\|\mathbf{u}\|_2 \leq B$, the surrogate loss satisfies

$$f_t(\mathbf{x}_t) - f_t(\mathbf{u}) \leq \widehat{f}_t(\mathbf{x}_t) - \widehat{f}_t(\mathbf{u}) = \langle\mathbf{g}_t, \mathbf{x}_t - \mathbf{u}\rangle - \frac{\eta_t}{2}\langle\mathbf{g}_t, \mathbf{x}_t - \mathbf{u}\rangle^2.$$

Multiplying the above equation by $\beta^{-t}$ and summing over $s \in [t]$ gives

$$\sum_{s=1}^{t}\beta^{-s}\left(f_s(\mathbf{x}_s) - f_s(\mathbf{u})\right) \leq \sum_{s=1}^{t}\beta^{-s}\langle\mathbf{g}_s, \mathbf{x}_s - \mathbf{u}\rangle - \sum_{s=1}^{t}\frac{\beta^{-s}\eta_s}{2}\langle\mathbf{g}_s, \mathbf{x}_s - \mathbf{u}\rangle^2. \tag{15}$$

Next, rewrite the discounted AIOLI in the rescaled form by multiplying the objective by $\beta^{-t}$:

$$\mathbf{x}_t \in \arg\min_{\mathbf{x} \in \mathbb{R}^d} \left\{ \sum_{s=1}^{t-1} \langle \beta^{-s} \mathbf{g}_s, \mathbf{x} \rangle + \beta^{-t} h_t(\mathbf{x}) + \psi_t(\mathbf{x}) \right\}, \tag{16}$$

where the regularizer is

$$\psi_t(\mathbf{x}) = \frac{\lambda}{2} \|\mathbf{x}\|_2^2 + \sum_{s=1}^{t-1} \frac{\beta^{-s} \eta_s}{2} \langle \mathbf{g}_s, \mathbf{x} - \mathbf{x}_s \rangle^2.$$

Thus Eq. (16) is exactly the optimistic FTRL (Lemma 22) with linear losses $\langle \beta^{-t} \mathbf{g}_t, \mathbf{x} \rangle$ and optimistic function $\beta^{-t} h_t(\mathbf{x})$. Let $\widehat{\mathbf{x}}_t$ denote the decision without optimism:

$$\widehat{\mathbf{x}}_t \in \arg\min_{\mathbf{x} \in \mathbb{R}^d} \left\{ F_t(\mathbf{x}) \right\}, \qquad F_t(\mathbf{x}) = \psi_t(\mathbf{x}) + \sum_{s=1}^{t-1} \langle \beta^{-s} \mathbf{g}_s, \mathbf{x} \rangle.$$

Applying Lemma 22 to $\{ \langle \beta^{-t} \mathbf{g}_t, \mathbf{x} \rangle \}_{t=1}^s$ gives, for any $\mathbf{u}$ such that $\|\mathbf{u}\|_2 \leq B$,

$$\sum_{s=1}^{t} \langle \beta^{-s} \mathbf{g}_s, \mathbf{x}_s - \mathbf{u} \rangle \leq \psi_{t+1}(\mathbf{u}) - \min_{\mathbf{x}} \psi_1(\mathbf{x}) + \sum_{s=1}^{t} \left( \langle \beta^{-s} \mathbf{g}_s, \mathbf{x}_s \rangle + F_s(\widehat{\mathbf{x}}_s) - F_{s+1}(\widehat{\mathbf{x}}_{s+1}) \right), \tag{17}$$

where we use $F_{t+1}(\widehat{\mathbf{x}}_{t+1}) \leq F_{t+1}(\mathbf{u})$ to drop the last term in Lemma 22. Since $\psi_1(\mathbf{x}) = \frac{\lambda}{2} \|\mathbf{x}\|_2^2$, we have $\min_{\mathbf{x}} \psi_1(\mathbf{x}) = 0$. Subtracting $\sum_{s=1}^{t} \frac{\beta^{-s} \eta_s}{2} \langle \mathbf{g}_s, \mathbf{x}_s - \mathbf{u} \rangle^2$ from both sides of Eq. (17) and combining with Eq. (15) provides

$$\sum_{s=1}^{t} \beta^{-s} \left( f_s(\mathbf{x}_s) - f_s(\mathbf{u}) \right) \leq \frac{\lambda}{2} \|\mathbf{u}\|_2^2 + \sum_{s=1}^{t} \left( \langle \beta^{-s} \mathbf{g}_s, \mathbf{x}_s \rangle + F_s(\widehat{\mathbf{x}}_s) - F_{s+1}(\widehat{\mathbf{x}}_{s+1}) \right). \tag{18}$$

It remains to bound the stability terms on the right-hand side. By definition, $\psi_{s+1}(\cdot)$ is the proximal regularizer (Orabona, 2019) and satisfies $\psi_{s+1}(\mathbf{x}) - \psi_s(\mathbf{x}) = \frac{\beta^{-s} \eta_s}{2} \langle \mathbf{g}_s, \mathbf{x} - \mathbf{x}_s \rangle^2$. In particular,

$$\psi_{s+1}(\mathbf{x}_s) - \psi_s(\mathbf{x}_s) = 0, \quad \nabla \psi_{s+1}(\mathbf{x}_s) - \nabla \psi_s(\mathbf{x}_s) = \mathbf{0}. \tag{19}$$

First, Eq. (19) implies

$$\begin{aligned} \langle \beta^{-s} \mathbf{g}_s, \mathbf{x}_s \rangle + F_s(\widehat{\mathbf{x}}_s) - F_{s+1}(\widehat{\mathbf{x}}_{s+1}) &= F_{s+1}(\mathbf{x}_s) - F_s(\mathbf{x}_s) + \psi_s(\mathbf{x}_s) - \psi_{s+1}(\mathbf{x}_s) + F_s(\widehat{\mathbf{x}}_s) - F_{s+1}(\widehat{\mathbf{x}}_{s+1}) \\ &= F_{s+1}(\mathbf{x}_s) - F_s(\mathbf{x}_s) + F_s(\widehat{\mathbf{x}}_s) - F_{s+1}(\widehat{\mathbf{x}}_{s+1}) \\ &= (F_{s+1}(\mathbf{x}_s) - F_{s+1}(\widehat{\mathbf{x}}_{s+1})) - (F_s(\mathbf{x}_s) - F_s(\widehat{\mathbf{x}}_s)). \end{aligned}$$

Second, by the first-order optimal condition,

$$\begin{aligned} \nabla F_s(\mathbf{x}_s) + \beta^{-s} \nabla h_s(\mathbf{x}_s) &= \mathbf{0}, \\ \nabla F_{s+1}(\mathbf{x}_s) = \nabla F_s(\mathbf{x}_s) + \beta^{-s} \mathbf{g}_s + \nabla \psi_{s+1}(\mathbf{x}_s) - \nabla \psi_s(\mathbf{x}_s) &= \beta^{-s} \mathbf{g}_s - \beta^{-s} \nabla h_s(\mathbf{x}_s), \end{aligned}$$

where the last equality uses Eq. (19) and $\nabla F_s(\mathbf{x}_s) = -\beta^{-s} \nabla h_s(\mathbf{x}_s)$. Since $F_{s+1}(\cdot)$ is a quadratic function with Hessian $A_s$ defined in Eq. (14), we have the standard identity

$$F_{s+1}(\mathbf{x}) - F_{s+1}(\widehat{\mathbf{x}}_{s+1}) = \frac{1}{2} \|\nabla F_{s+1}(\mathbf{x})\|_{A_s^{-1}}^2 \qquad \text{for all } \mathbf{x} \in \mathbb{R}^d.$$

Using this identity and $A_s \succeq A_{s-1}$, we obtain

$$\begin{aligned} &\langle \beta^{-s} \mathbf{g}_s, \mathbf{x}_s \rangle + F_s(\widehat{\mathbf{x}}_s) - F_{s+1}(\widehat{\mathbf{x}}_{s+1}) \\ &= (F_{s+1}(\mathbf{x}_s) - F_{s+1}(\widehat{\mathbf{x}}_{s+1})) - (F_s(\mathbf{x}_s) - F_s(\widehat{\mathbf{x}}_s)) = \frac{1}{2} \|\nabla F_{s+1}(\mathbf{x}_s)\|_{A_s^{-1}}^2 - \frac{1}{2} \|\nabla F_s(\mathbf{x}_s)\|_{A_{s-1}^{-1}}^2 \end{aligned}$$

$$\leq \frac{\beta^{-2s}}{2}\Big(\big\|\mathbf{g}_s - \nabla h_s(\mathbf{x}_s)\big\|_{A_s^{-1}}^2 - \big\|\nabla h_s(\mathbf{x}_s)\big\|_{A_s^{-1}}^2\Big) = \frac{\beta^{-2s}}{2}\langle \mathbf{g}_s, \mathbf{g}_s - 2\nabla h_s(\mathbf{x}_s)\rangle_{A_s^{-1}}. \tag{20}$$

For the logistic loss, let $\mathbf{g}_s^{-y_s} = \nabla\ell(\mathbf{x}_s^\top \mathbf{z}_s, -y_s)$ denote the gradient w.r.t. the opposite label. Then $\nabla h_s(\mathbf{x}_s) = \mathbf{g}_s + \mathbf{g}_s^{-y_s}$, and

$$\frac{1}{2}\langle \mathbf{g}_s, \mathbf{g}_s - 2\nabla h_s(\mathbf{x}_s)\rangle_{A_s^{-1}} = \frac{1}{2}\langle \mathbf{g}_s, -\mathbf{g}_s - 2\mathbf{g}_s^{-y_s}\rangle_{A_s^{-1}} \leq -\langle \mathbf{g}_s, \mathbf{g}_s^{-y_s}\rangle_{A_s^{-1}}.$$

Moreover, by the explicit form of the logistic gradient, $\mathbf{g}_s^{-y_s} = -\exp(y_s\mathbf{x}_s^\top \mathbf{z}_s)\mathbf{g}_s = -(1 + BR)\eta_s\mathbf{g}_s$ by (13). Hence

$$-\langle \mathbf{g}_s, \mathbf{g}_s^{-y_s}\rangle_{A_s^{-1}} = (1 + BR)\eta_s\mathbf{g}_s^\top A_s^{-1}\mathbf{g}_s.$$

Combining this with (20) shows

$$\langle \beta^{-s}\mathbf{g}_s, \mathbf{x}_s\rangle + F_s(\widehat{\mathbf{x}}_s) - F_{s+1}(\widehat{\mathbf{x}}_{s+1}) \leq (1 + BR)\beta^{-2s}\eta_s\mathbf{g}_s^\top A_s^{-1}\mathbf{g}_s.$$

Plugging the above inequality into (18) finishes the proof. $\qquad\square$

### C.3. Learning the Discount Factor for Online Logistic Regression

In this subsection, we demonstrate how to learn a suitable discount factor for online logistic regression. Similar to Mhammedi & Rakhlin (2022); Jacobsen & Cutkosky (2024), we employ the *online ensemble* framework (Zhao et al., 2024) to handle the uncertainty in choosing $\beta$ in an online manner.

In specific, we maintain $N$ base learners. Base learner $i$ runs discounted AIOLI with an assigned discount factor $\beta_i \in (0, 1]$, and outputs a decision $\mathbf{x}_{t,i} \in \mathbb{R}^d$ at round $t$. Let $\widehat{y}_{t,i} = \mathbf{x}_{t,i}^\top \mathbf{z}_t$ be its prediction. A meta learner combines the base learners and produces a final prediction $\widehat{y}_t \in \mathbb{R}$, after which the learner suffers logistic loss $\ell(\widehat{y}_t, y_t)$.

For any fixed index $i \in [N]$, the (interval) dynamic regret can be decomposed as

$$\sum_{t=1}^T \ell(\widehat{y}_t, y_t) - \sum_{t=1}^T \ell(\mathbf{u}_t^\top \mathbf{z}_t, y_t) = \underbrace{\sum_{t=1}^T \Big(\ell(\widehat{y}_t, y_t) - \ell(\widehat{y}_{t,i}, y_t)\Big)}_{\text{META REGRET}} + \underbrace{\sum_{t=1}^T \Big(\ell(\widehat{y}_{t,i}, y_t) - \ell(\mathbf{u}_t^\top \mathbf{z}_t, y_t)\Big)}_{\text{BASE REGRET}}. \tag{21}$$

If there exists a $\beta_i$ close to an ideal choice, then the base regret has the desired order. The main remaining issue is to control the meta regret without introducing an exponential dependence on $B$. For this purpose, we use the mixability of the logistic loss (Vovk, 2001; Cesa-Bianchi & Lugosi, 2006; van Erven et al., 2012; Foster et al., 2018; Mhammedi & Rakhlin, 2022; Zhang et al., 2025). We first give the definition of a mixable loss.

**Definition 2** (mixable loss). A loss $\ell(\cdot, \cdot) : \widehat{\mathcal{Y}} \times \mathcal{Y} \to \mathbb{R}$ is $\eta$-*mixable* if for any predictions $\widehat{y}_1, \ldots, \widehat{y}_N \in \mathbb{R}$, any weights $p_1, \ldots, p_N \geq 0$ with $\sum_{i=1}^N p_i = 1$, there exists a mixed prediction $\widehat{y}_{\mathrm{mix}} \in \mathbb{R}$ such that for any $y \in \mathcal{Y}$,

$$\ell(\widehat{y}_{\mathrm{mix}}, y) \leq -\frac{1}{\eta}\ln\Big(\sum_{i=1}^N p_i e^{-\eta\ell(\widehat{y}_i, y)}\Big).$$

The logistic loss is 1-mixable, and using this property lets us avoid the exponential factor in the meta regret.

**Proposition 1** (Foster et al. (2018)). *The logistic loss $\ell(\widehat{y}, y) = \ln(1 + \exp(-y\widehat{y}))$ on $\mathbb{R} \times \{-1, +1\}$ is 1-mixable.*

Moreover, the mixed prediction can be written in the closed form (Foster et al., 2018). Let $\sigma(a) = \exp(a)/(1 + \exp(a))$. Given $\widehat{y}_1, \ldots, \widehat{y}_N$ and weights $p_i$, one valid choice is

$$\widehat{y}_{\mathrm{mix}} = \ln\left(\frac{\sum_{i=1}^N p_i\sigma(\widehat{y}_i)}{\sum_{i=1}^N p_i(1 - \sigma(\widehat{y}_i))}\right).$$

Based on this, we develop the following two-layer structure ensemble method, presented in Algorithm 2, to learn the ideal $\beta$. The theoretical guarantees of Algorithm 2 are summarized in the following theorem.

---

**Algorithm 2** Ensemble algorithm for learning $\beta$ in logistic regression

---

**Input:** number of base learners $N$, a discount factor pool $\{\beta_i\}_{i=1}^N$, and base-learner parameters $(B, R, \lambda)$.

1: **Initialize:** $q_{1,i} = 1$ and base learner $\mathcal{B}_i$ (discounted AIOLI) with $\beta_i, B, R, \lambda$, for each $i \in [N]$.

2: **for** $t = 1$ **to** $T$ **do**

3:     Receive feature $\mathbf{z}_t \in \mathbb{R}^d$;

4:     Each base learner $\mathcal{B}_i$ outputs $\mathbf{x}_{t,i} \in \mathbb{R}^d$ and predicts $\widehat{y}_{t,i} = \mathbf{x}_{t,i}^\top \mathbf{z}_t$;

5:     Set $p_{t,i} = q_{t,i} / \sum_{j=1}^N q_{t,j}$;

6:     Predict

$$\widehat{y}_t = \ln \left( \frac{\sum_{i=1}^N p_{t,i} \frac{\exp(\widehat{y}_{t,i})}{1+\exp(\widehat{y}_{t,i})}}{\sum_{i=1}^N p_{t,i} \frac{1}{1+\exp(\widehat{y}_{t,i})}} \right);$$

7:     Observe label $y_t \in \{-1, +1\}$ and update $q_{t+1,i} = q_{t,i} \exp\left( - \ell(\widehat{y}_{t,i}, y_t) \right)$ for all $i \in [N]$.

8: **end for**

---

**Theorem 8.** *Assume $\|\mathbf{z}_t\|_2 \le R$ for all $t \in [T]$ and define the convex loss $f_t(\mathbf{x}) = \ell(\mathbf{x}^\top \mathbf{z}_t, y_t)$, where $\ell(\widehat{y}, y) = \ln(1 + \exp(-y\widehat{y}))$. Set $\lambda = 1/B^2$. Let $C = \max\{1, 2R\}$ and define the parameter range $\eta_{\min} = \sqrt{\frac{d(1+BR)}{CB}}$, $\eta_{\max} = dT$, and the geometric grid $\{\eta_i = 2^{i-1}\eta_{\min} : \beta_i = \frac{\eta_i}{1+\eta_i} \in (0,1), i \in [N]\}$, where $N = \left\lceil \log_2\left(\frac{\eta_{\max}}{\eta_{\min}}\right) \right\rceil + 1$. Run Algorithm 2 with the discount factors pool $\{\beta_i = \frac{\eta_i}{1+\eta_i}\}_{i=1}^N$. Denote*

$$P_T^\beta = \sum_{t=1}^{T-1} \sum_{s=0}^t p_{t,s}^\beta \left[ f_s(\mathbf{u}_{t+1}) - f_s(\mathbf{u}_t) \right]_+,$$

*with $f_0(\mathbf{u}) = \frac{\lambda}{2}\|\mathbf{u}\|_2^2$ and let $\beta_\star \in (0, 1]$ satisfy*

$$\beta_\star = \frac{\sqrt{d(1+BR)T}}{\sqrt{d(1+BR)T} + \sqrt{P_T^{\beta_\star}}}. \tag{22}$$

*Then Algorithm 2 ensures that for any comparator sequence $\mathbf{u}_1, \dots, \mathbf{u}_T$ with $\|\mathbf{u}_t\|_2 \le B$,*

$$\sum_{t=1}^T \ell(\widehat{y}_t, y_t) - \sum_{t=1}^T \ell(\mathbf{u}_t^\top \mathbf{z}_t, y_t) \le \mathcal{O}\left( dB \log(BT) + \sqrt{dBT P_T^{\beta_\star}} \right).$$

*Proof of Theorem 8.* Fix any base learner index $i \in [N]$. Recall the decomposition in (21):

$$\sum_{t=1}^T \ell(\widehat{y}_t, y_t) - \sum_{t=1}^T \ell(\mathbf{u}_t^\top \mathbf{z}_t, y_t) = \underbrace{\sum_{t=1}^T \left( \ell(\widehat{y}_t, y_t) - \ell(\widehat{y}_{t,i}, y_t) \right)}_{\text{META REGRET}} + \underbrace{\sum_{t=1}^T \left( f_t(\mathbf{x}_{t,i}) - f_t(\mathbf{u}_t) \right)}_{\text{BASE REGRET}}.$$

**Meta regret.** By the 1-mixability of the logistic loss (Proposition 1) and the specific mixed prediction in Algorithm 2, for every $t$,

$$\ell(\widehat{y}_t, y_t) \le - \ln \left( \sum_{j=1}^N p_{t,j} \exp\left( - \ell(\widehat{y}_{t,j}, y_t) \right) \right).$$

Using $p_{t,j} = q_{t,j} / \sum_{k=1}^N q_{t,k}$ and the update $q_{t+1,j} = q_{t,j} \exp(-\ell(\widehat{y}_{t,j}, y_t))$, we obtain

$$\ell(\widehat{y}_t, y_t) \le - \ln \left( \sum_{j=1}^N \frac{q_{t,j}}{\sum_{k=1}^N q_{t,k}} \exp\left( - \ell(\widehat{y}_{t,j}, y_t) \right) \right) = - \ln \left( \frac{\sum_{j=1}^N q_{t+1,j}}{\sum_{k=1}^N q_{t,k}} \right).$$

Summing over $t = 1, \ldots, T$ yields the telescoping bound

$$\sum_{t=1}^{T} \ell(\widehat{y}_t, y_t) \leq -\ln\left(\frac{\sum_{j=1}^{N} q_{T+1,j}}{\sum_{j=1}^{N} q_{1,j}}\right) = \ln\left(\sum_{j=1}^{N} q_{1,j}\right) - \ln\left(\sum_{j=1}^{N} q_{T+1,j}\right).$$

Since $q_{1,j} = 1$, we have $\sum_{j=1}^{N} q_{1,j} = N$, and also $\sum_{j=1}^{N} q_{T+1,j} \geq q_{T+1,i}$. Therefore,

$$\sum_{t=1}^{T} \ell(\widehat{y}_t, y_t) \leq \ln N - \ln q_{T+1,i} = \ln N + \sum_{t=1}^{T} \ell(\widehat{y}_{t,i}, y_t),$$

which implies META REGRET $\leq \ln N$. Notice that $N = \mathcal{O}(\log T)$, so $\ln N = \mathcal{O}(\log \log T)$, which we treat as a constant following previous conventions (Gaillard et al., 2014; Luo & Schapire, 2015).

For any $s \in [T]$ the loss $f_s(\mathbf{x}) = \ell(\mathbf{x}^\top \mathbf{z}_s, y_s)$ is $R$-Lipschitz w.r.t. $\|\cdot\|_2$ since $\|\nabla f_s(\mathbf{x})\|_2 \leq \|\mathbf{z}_s\|_2 \leq R$. Hence for any $t \in [T-1]$ and any $s \leq t$,

$$\left[f_s(\mathbf{u}_{t+1}) - f_s(\mathbf{u}_t)\right]_+ \leq |f_s(\mathbf{u}_{t+1}) - f_s(\mathbf{u}_t)| \leq \max\left\{R, \lambda B\right\}\|\mathbf{u}_{t+1} - \mathbf{u}_t\|_2 \leq \max\{2R, 1\}B,$$

where we used $\|\mathbf{u}_t\|_2 \leq B$ and $\lambda = \frac{1}{B^2}$. Since $\sum_{s=0}^{t} p_{t,s}^\beta = 1$, we obtain for all $\beta \in (0, 1]$,

$$P_T^\beta = \sum_{t=1}^{T-1} \sum_{s=0}^{t} p_{t,s}^\beta \left[f_s(\mathbf{u}_{t+1}) - f_s(\mathbf{u}_t)\right]_+ \leq CBT,$$

where $C = \max\{2R, 1\}$. In particular, for the $\beta_\star$ in the theorem statement, $P_T^{\beta_\star} \leq CBT$, and thus

$$\eta_\star = \frac{\beta_\star}{1 - \beta_\star} = \sqrt{\frac{d(1 + BR)T}{P_T^{\beta_\star}}} \geq \sqrt{\frac{d(1 + BR)T}{CBT}} = \sqrt{\frac{d(1 + BR)}{CB}} = \eta_{\min}.$$

**Base regret.** For a base learner with discount $\beta_i \in (0, 1)$ and $\lambda = 1/B^2$, the discounted AIOLI guarantee (Theorem 3, in the form before converting the path-length term into $\frac{\beta}{1-\beta} P_T^\beta$) gives

$$\sum_{t=1}^{T} \left(f_t(\mathbf{x}_{t,i}) - f_t(\mathbf{u}_t)\right) \leq \underbrace{\beta_i \cdot \frac{\lambda}{2}\|\mathbf{u}_1\|_2^2}_{\leq 1/2} + \underbrace{d(1 + BR)\log\left(1 + \frac{R^2 B^2 T}{d(1 + BR)}\right)}_{\text{TERM-A}} + \underbrace{d(1 + BR)T\log\left(\frac{1}{\beta_i}\right)}_{\text{TERM-B}}$$

$$+ \underbrace{\beta_i \sum_{t=1}^{T-1} \left(F_t^{\beta_i, \varphi}(\mathbf{u}_{t+1}) - F_t^{\beta_i, \varphi}(\mathbf{u}_t)\right)}_{\text{TERM-C}}. \tag{23}$$

We now upper bound TERM-B and TERM-C by choosing a suitable $\beta_i$ from the pool. Define $\eta_i = \beta_i/(1 - \beta_i)$. Using $\ln(1/x) \leq (1/x) - 1$, we have $\ln(\frac{1}{\beta_i}) \leq (1 - \beta_i)/\beta_i$, and therefore

$$\text{TERM-B} \leq \frac{d(1 + BR)T}{\eta_i}.$$

Next by Lemma 18, for any $0 < \beta \leq \gamma < 1$,

$$\beta \sum_{t=1}^{T-1} \left(F_t^{\beta, \varphi}(\mathbf{u}_{t+1}) - F_t^{\beta, \varphi}(\mathbf{u}_t)\right) \leq \frac{\gamma}{1 - \gamma} P_T^\gamma. \tag{24}$$

Let $\eta_\star = \beta_\star/(1 - \beta_\star)$. Since $\eta_\star \geq \eta_{\min}$, by construction of the learning rates pool there exists an index $i \in [N]$ such that $\eta_i \leq \min\{\eta_\star, \eta_{\max}\} \leq 2\eta_i$. We consider two cases.

**Case 1**: if $\eta_\star \leq \eta_{\max}$. Then $\eta_i \leq \eta_\star$ and hence $\beta_i \leq \beta_\star$ by monotonicity of $\beta = \eta/(1+\eta)$ in $\eta$. Applying (24) with $(\beta, \gamma) = (\beta_i, \beta_\star)$ yields

$$\text{TERM-C} \leq \frac{\beta_\star}{1 - \beta_\star} P_T^{\beta_\star} = \eta_\star P_T^{\beta_\star}.$$

Moreover, $\eta_\star \leq 2\eta_i$ implies $\eta_i \geq \eta_\star/2$, and thus

$$\text{TERM-B} \leq \frac{d(1 + BR)T}{\eta_i} \leq \frac{2d(1 + BR)T}{\eta_\star}.$$

Plugging these bounds into (23) gives

$$\sum_{t=1}^{T} \left( f_t(\mathbf{x}_{t,i}) - f_t(\mathbf{u}_t) \right) \leq \frac{1}{2} + d(1 + BR) \log \left( 1 + \frac{R^2 B^2 T}{d(1 + BR)} \right) + \eta_\star P_T^{\beta_\star} + \frac{2d(1 + BR)T}{\eta_\star}.$$

Finally, since the optimal value is $\eta_\star = \sqrt{d(1 + BR)T/P_T^{\beta_\star}}$, we have $\eta_\star P_T^{\beta_\star} = \sqrt{d(1 + BR)T P_T^{\beta_\star}}$ and $\frac{d(1+BR)T}{\eta_\star} = \sqrt{d(1 + BR)T P_T^{\beta_\star}}$, so the last two terms are bounded by $3\sqrt{d(1 + BR)T P_T^{\beta_\star}}$.

**Case 2**: if $\eta_\star > \eta_{\max}$. We take $i$, such that $\frac{1}{2}\eta_{\max} \leq \eta_i \leq \eta_{\max}$ and still $\eta_i \leq \eta_\star$, hence $\beta_i \leq \beta_\star$. The same argument gives $\text{TERM-C} \leq \eta_\star P_T^{\beta_\star} = \sqrt{d(1 + BR)T P_T^{\beta_\star}}$, while $\text{TERM-B} \leq 2d(1 + BR)T/\eta_{\max} = 2(1 + BR)$ by $\eta_{\max} = dT$. Thus

$$\sum_{t=1}^{T} \left( f_t(\mathbf{x}_{t,N}) - f_t(\mathbf{u}_t) \right) \leq \frac{1}{2} + d(1 + BR) \log \left( 1 + \frac{R^2 B^2 T}{d(1 + BR)} \right) + \sqrt{d(1 + BR)T P_T^{\beta_\star}} + 2(1 + BR).$$

Combining the meta regret bound and base regret bound together, we finish the proof. $\qquad \square$

# D. Omitted Details for Section 4

In this part, we present the details omitted from Section 4, including several key lemmas and discussions, as well as detailed proofs for the results on Adam optimizers.

## D.1. Closed-Form Update Rule for Clip-Free Adam

In this part, we provide the derivation of the closed-form update rule for clip-free Adam.

**Lemma 4.** *Fix $\beta_1, \beta_2 \in (0, 1)$ and parameters $\gamma > 0$, $\nu > 0$, and $\mu > 0$. Consider the discounted FTRL update (5) with $\beta = \beta_1$ and $\mathcal{D} = \mathbb{R}^d$, where the surrogate loss is*

$$\ell_t(\Delta) = \langle \mathbf{g}_t, \Delta \rangle + \frac{\mu}{2} \|\Delta\|_2^2,$$

*and the base step size $\eta_t$ is chosen as in (7), i.e.,*

$$\eta_t = \gamma \cdot \frac{(1 - \beta_1)\beta_1^t}{\nu + \sqrt{(1 - \beta_2) \sum_{s=1}^{t} \beta_2^{t-s} \|\mathbf{g}_s\|_2^2}}.$$

*Then the update admits the closed form*

$$\Delta_{t+1} = -\frac{\gamma(1 - \beta_1) \sum_{s=1}^{t} \beta_1^{t-s} \mathbf{g}_s}{\nu + \gamma\mu(1 - \beta_1^t) + \sqrt{(1 - \beta_2) \sum_{s=1}^{t} \beta_2^{t-s} \|\mathbf{g}_s\|_2^2}}.$$

*Proof.* By substituting $\ell_s(\Delta) = \langle \mathbf{g}_s, \Delta \rangle + \frac{\mu}{2} \|\Delta\|_2^2$ and $\beta = \beta_1$ into (5), the update is

$$\Delta_{t+1} = \arg\min_{\Delta \in \mathbb{R}^d} \left\{ \frac{1}{2\eta_t} \|\Delta\|_2^2 + \sum_{s=1}^{t} \beta_1^{-s} \langle \mathbf{g}_s, \Delta \rangle + \sum_{s=1}^{t} \beta_1^{-s} \frac{\mu}{2} \|\Delta\|_2^2 \right\}$$

$$= \underset{\Delta \in \mathbb{R}^d}{\arg\min} \Big\{ \Big\langle \sum_{s=1}^{t} \beta_1^{-s} \mathbf{g}_s, \Delta \Big\rangle + \frac{1}{2} \Big( \frac{1}{\eta_t} + \mu \sum_{s=1}^{t} \beta_1^{-s} \Big) \|\Delta\|_2^2 \Big\}.$$

The objective is a strictly convex quadratic in $\Delta$, by first-order optimality, the minimizer is:

$$0 = \sum_{s=1}^{t} \beta_1^{-s} \mathbf{g}_s + \Big( \frac{1}{\eta_t} + \mu \sum_{s=1}^{t} \beta_1^{-s} \Big) \Delta_{t+1},$$

which gives

$$\Delta_{t+1} = -\eta_t' \sum_{s=1}^{t} \beta_1^{-s} \mathbf{g}_s, \qquad \eta_t' = \frac{1}{\frac{1}{\eta_t} + \mu \sum_{s=1}^{t} \beta_1^{-s}}. \tag{25}$$

Next, compute the geometric sum $\sum_{s=1}^{t} \beta_1^{-s} = \frac{\beta_1^{-t} - 1}{1 - \beta_1}$. Using the definition of $\eta_t$ in (7), we have

$$\frac{1}{\eta_t} = \frac{\nu + \sqrt{(1-\beta_2) \sum_{s=1}^{t} \beta_2^{t-s} \|\mathbf{g}_s\|_2^2}}{\gamma(1-\beta_1)\beta_1^t}.$$

Substituting these identities into the definition of $\eta_t'$ in (25) gives

$$\eta_t' = \frac{1}{\frac{\nu + \sqrt{(1-\beta_2) \sum_{s=1}^{t} \beta_2^{t-s} \|\mathbf{g}_s\|_2^2}}{\gamma(1-\beta_1)\beta_1^t} + \mu \cdot \frac{\beta_1^{-t}-1}{1-\beta_1}} = \frac{\gamma(1-\beta_1)\beta_1^t}{\nu + \gamma\mu(1-\beta_1^t) + \sqrt{(1-\beta_2) \sum_{s=1}^{t} \beta_2^{t-s} \|\mathbf{g}_s\|_2^2}}.$$

Multiplying this expression by $\sum_{s=1}^{t} \beta_1^{-s} \mathbf{g}_s$ finishes the proof. $\qquad\square$

### D.2. Effects of $(\beta_1, \beta_2)$ Choices through $\rho$

To give a sense of scale in Theorem 5, we list several representative choices of $(\beta_1, \beta_2)$ below:

- Default setting in PyTorch $(0.9, 0.999)$: $\rho \approx 0.989$ and $1/\sqrt{1-\rho^2} \approx 6.9$;

- Common choices in large language models $(0.9, 0.95)$: $\rho \approx 0.473$ and $1/\sqrt{1-\rho^2} \approx 1.13$;

- Recommended by Orvieto & Gower (2025) $(0.95, 0.95)$: $\rho \approx 0.025$ and $1/\sqrt{1-\rho^2} \approx 1.0$.

For typical choices where $\beta_2$ is close to 1, the resulting constant is moderate. Our theory allows a broad range of $\beta_2$ as long as $\beta_2 > \beta_1^2$, and with a suitable choice of $\beta_1$ and a sufficiently long horizon $T$, it may provide theoretical support for the convergence behavior of Adam in practice, especially in settings where the model involves non-smooth components.

### D.3. Implications for Non-Convex and Smooth Settings

Similar to the $(\epsilon, \delta)$-Goldstein stationary points, $(c, \epsilon)$-stationary points can also be reduced to first-order stationary points via the following lemma.

**Lemma 5** (Lemma 2.3 in Zhang & Cutkosky (2024)). *The $(c, \epsilon)$-stationarity can be related to first-order stationarity for non-convex and smooth functions:*

- *Suppose $F$ is $L$-smooth. If $\|\nabla F(\mathbf{x})\|_c \leq \epsilon$ where $c = L^2\epsilon^{-1}$, then $\|\nabla F(\mathbf{x})\|_2 \leq 2\epsilon$.*

- *Suppose $F$ is $\rho$-second-order-smooth. If $\|\nabla F(\mathbf{x})\|_c \leq \epsilon$ where $c = \rho/2$, then $\|\nabla F(\mathbf{x})\|_2 \leq 2\epsilon$.*

Our results (Theorem $4-7$) attain the $\mathcal{O}(\max\{c^{1/2}\epsilon^{-7/2}, \epsilon^{-3}\}) = \mathcal{O}(c^{1/2}\epsilon^{-7/2})$ rate to a $(c, \epsilon)$-stationary point. By setting $c = \mathcal{O}(\epsilon^{-1})$, we achieve the $\mathcal{O}(\epsilon^{-4})$ optimal rate for non-convex and smooth functions (Arjevani et al., 2023). By setting $c = \mathcal{O}(1)$, we obtain the $\mathcal{O}(\epsilon^{-7/2})$ optimal rate for the second-order-smooth functions (Arjevani et al., 2023).

## D.4. Lemmas for O2NC Conversion

This section collects the lemmas for O2NC conversion.

**Lemma 6** (Lemma 3.1 in Zhang & Cutkosky (2024)). *Under Assumption 3, let $s \sim \mathrm{Exp}(\lambda)$ for some $\lambda > 0$, then*

$$\mathbb{E}_s\left[F(\mathbf{x} + s \cdot \Delta) - F(\mathbf{x})\right] = \mathbb{E}_s\left[\langle \nabla F(\mathbf{x} + s \cdot \Delta), \Delta \rangle\right].$$

**Lemma 7** (Adapted from Theorem C.1 in Zhang & Cutkosky (2024)). *Under Assumptions 1 − 4, set the loss in Algorithm 1 as $\ell_t(\Delta) = \langle \mathbf{g}_t, \Delta \rangle + \frac{\mu}{2}\|\Delta\|_2^2$, define the comparator $\mathbf{u}_t$ and the dynamic regret $\mathrm{D\text{-}REG}_T(\mathbf{u}_{1:T})$ as follows,*

$$\mathbf{u}_t = -D \cdot \frac{\sum_{s=1}^t \beta^{t-s}\nabla F(\mathbf{x}_s)}{\|\sum_{s=1}^t \beta^{t-s}\nabla F(\mathbf{x}_s)\|_2}, \quad \mathrm{D\text{-}REG}_T(\mathbf{u}_{1:T}) = \sum_{t=1}^T \langle \mathbf{g}_t, \Delta_t - \mathbf{u}_t \rangle + \frac{\mu}{2}\|\Delta_t\|_2^2 - \frac{\mu}{2}\|\mathbf{u}_t\|_2^2.$$

*Set $\mu = \frac{24cD}{(1-\beta)^2}$, then the returned decision $\bar{\mathbf{x}}$ in Algorithm 1 ensures:*

$$\mathbb{E}\|\nabla F(\bar{\mathbf{x}})\|_c \leq \frac{F^*}{DT} + \frac{2G + \sigma}{(1-\beta)T} + \sigma\sqrt{1-\beta} + \frac{12cD^2}{(1-\beta)^2}$$
$$+ \frac{1}{DT}\mathbb{E}\left[\mathrm{D\text{-}REG}_T(\mathbf{u}_{1:T}) + \beta\sum_{t=1}^{T-1}\sum_{s=1}^t \beta^{t-s}\left(\ell_s(\mathbf{u}_t) - \ell_s(\mathbf{u}_{t+1})\right)\right].$$

*Proof.* By Theorem C.1 in Zhang & Cutkosky (2024), the same setting provides the guarantee of:

$$\mathbb{E}\|\nabla F(\bar{\mathbf{x}})\|_c \leq \frac{F^*}{DT} + \frac{2G + \sigma}{(1-\beta)T} + \sigma\sqrt{1-\beta} + \frac{12cD^2}{(1-\beta)^2} + \frac{1}{DT}\mathbb{E}\left[\beta\mathrm{REG}_{T;\beta}(\mathbf{u}_T) + (1-\beta)\sum_{t=1}^T \mathrm{REG}_{t;\beta}(\mathbf{u}_t)\right],$$

where $\mathrm{REG}_{t;\beta}(\mathbf{u}) = \sum_{s=1}^t \left(\beta^{t-s}\langle \mathbf{g}_s, \Delta_s - \mathbf{u}\rangle + \frac{\mu\beta^{t-s}}{2}(\|\Delta_s\|_2^2 - \|\mathbf{u}\|_2^2)\right)$. By Lemma 1, and the definition of $\ell_t(\Delta) = \langle \mathbf{g}_t, \Delta \rangle + \frac{\mu}{2}\|\Delta\|_2^2$, we have:

$$\beta\mathrm{REG}_{T;\beta}(\mathbf{u}_T) + (1-\beta)\sum_{t=1}^T \mathrm{REG}_{t;\beta}(\mathbf{u}_t) = \mathrm{D\text{-}REG}_T(\mathbf{u}_{1:T}) + \beta\sum_{t=1}^{T-1}\sum_{s=1}^t \beta^{t-s}\left(\ell_s(\mathbf{u}_t) - \ell_s(\mathbf{u}_{t+1})\right),$$

which finishes the proof. $\square$

**Lemma 8** (Adapted from Lemma 7 and Lemma 10 in Ahn & Cutkosky (2024)). *Under Assumptions 1 − 4, set the loss in Algorithm 1 as $\ell_t(\Delta) = \langle \mathbf{g}_t, \Delta \rangle$, define the comparator $\mathbf{u}_t$ and the dynamic regret $\mathrm{D\text{-}REG}_T(\mathbf{u}_{1:T})$ as follows,*

$$\mathbf{u}_t = -D \cdot \frac{\sum_{s=1}^t \beta^{t-s}\nabla F(\mathbf{x}_s)}{\|\sum_{s=1}^t \beta^{t-s}\nabla F(\mathbf{x}_s)\|_2}, \quad \mathrm{D\text{-}REG}_T(\mathbf{u}_{1:T}) = \sum_{t=1}^T \langle \mathbf{g}_t, \Delta_t - \mathbf{u}_t \rangle.$$

*Assume that $\|\Delta_t\|_2 \leq D$ for any $t$, then the returned decision $\bar{\mathbf{x}}$ in Algorithm 1 ensures:*

$$\mathbb{E}\|\nabla F(\bar{\mathbf{x}})\|_c \leq \frac{F^*}{DT} + \frac{2G + \sigma}{(1-\beta)T} + \sigma\sqrt{1-\beta} + \frac{12cD^2}{(1-\beta)^2}$$
$$+ \frac{1}{DT}\mathbb{E}\left[\mathrm{D\text{-}REG}_T(\mathbf{u}_{1:T}) + \beta\sum_{t=1}^{T-1}\sum_{s=1}^t \beta^{t-s}\left(\ell_s(\mathbf{u}_t) - \ell_s(\mathbf{u}_{t+1})\right)\right].$$

The proof of this lemma is identical to that of Lemma 7, and we omit it.

## D.5. Technical Lemmas for Adam Analyses

In this section, we present and prove several useful technical lemmas when analyzing Adam under the O2NC conversion.

**Lemma 9.** *Let $\beta \in (0,1)$, $\epsilon > 0$, and let $\{a_t\}_{t=1}^{T}$ satisfy $0 \le a_t \le A$ for all $t \in [T]$. Define the exponential moving average $V_0 = 0, V_t = \beta V_{t-1} + (1-\beta)a_t$, for $t \ge 1$. Then,*

$$\sum_{t=1}^{T} \frac{a_t}{\epsilon + \sqrt{V_{t-1}}} \le \left( \frac{T(1-\beta)}{\ln 2} + 1 \right) \frac{A}{\epsilon} + \frac{2}{\sqrt{(1-\beta)2^{-1/\beta}}} \sqrt{\left( \frac{T(1-\beta)}{\ln 2} + 1 \right) \sum_{t=1}^{T} a_t}.$$

*Proof.* Unrolling the recursion gives, for any $t \ge 1$,

$$V_{t-1} = (1-\beta) \sum_{s=1}^{t-1} \beta^{t-1-s} a_s.$$

Fix $\tau \ge 1$ (to be chosen later), and partition $[T]$ into $K = \lceil T/\tau \rceil$ consecutive segments $I_k = [r_k, s_k]$ of length at most $\tau$, where $r_k = (k-1)\tau + 1$ and $s_k = \min\{k\tau, T\}$. For any $t \in [r_k, s_k]$ and any $s \in [r_k, t-1]$, we have $t - 1 - s \le \tau - 1$, hence $\beta^{t-1-s} \ge \beta^{\tau-1}$. Therefore,

$$V_{t-1} = (1-\beta) \sum_{s=1}^{t-1} \beta^{t-1-s} a_s \ge (1-\beta) \sum_{s=r_k}^{t-1} \beta^{t-1-s} a_s \ge (1-\beta)\beta^{\tau-1} \sum_{s=r_k}^{t-1} a_s.$$

Let $c = (1-\beta)\beta^{\tau-1}$. Using the above inequality and summing over segments, we obtain

$$\sum_{t=1}^{T} \frac{a_t}{\epsilon + \sqrt{V_{t-1}}} = \sum_{k=1}^{K} \sum_{t=r_k}^{s_k} \frac{a_t}{\epsilon + \sqrt{V_{t-1}}} \le \sum_{k=1}^{K} \sum_{t=r_k}^{s_k} \frac{a_t}{\epsilon + \sqrt{c \sum_{s=r_k}^{t-1} a_s}}. \tag{26}$$

We now bound each inner sum. Fix a segment $I_k$ and define $S_{t-1} = \sum_{s=r_k}^{t-1} a_s$ (so $S_{r_k - 1} = 0$). For any $x \ge 0$, we have $(\epsilon + \sqrt{cx})^2 = \epsilon^2 + cx + 2\epsilon\sqrt{cx} \ge \epsilon^2 + cx$, hence

$$\frac{1}{\epsilon + \sqrt{cx}} \le \frac{1}{\sqrt{\epsilon^2 + cx}} = \frac{1}{\sqrt{c}} \cdot \frac{1}{\sqrt{\epsilon^2/c + x}}.$$

Let $\delta = \epsilon^2/c > 0$. Applying the above equation with $x = S_{t-1}$ gives

$$\sum_{t=r_k}^{s_k} \frac{a_t}{\epsilon + \sqrt{c S_{t-1}}} \le \frac{1}{\sqrt{c}} \sum_{t=r_k}^{s_k} \frac{a_t}{\sqrt{\delta + S_{t-1}}}.$$

We apply Lemma 23 with $a_0 = \delta$, $B = A$, and $f(u) = u^{-1/2}$, which gives

$$\sum_{t=r_k}^{s_k} \frac{a_t}{\sqrt{\delta + S_{t-1}}} \le A \cdot \frac{1}{\sqrt{\delta}} + \int_{\delta}^{\delta + \sum_{t=r_k}^{s_k} a_t} u^{-1/2} du$$

$$= A \cdot \frac{1}{\sqrt{\delta}} + 2\left( \sqrt{\delta + \sum_{t=r_k}^{s_k} a_t} - \sqrt{\delta} \right) \le A \cdot \frac{1}{\sqrt{\delta}} + 2\sqrt{\sum_{t=r_k}^{s_k} a_t},$$

where we used $\sqrt{\delta + S} - \sqrt{\delta} \le \sqrt{S}$ for $S \ge 0$. Since $\sqrt{\delta} = \epsilon/\sqrt{c}$, we conclude for each $I_k$ that

$$\sum_{t=r_k}^{s_k} \frac{a_t}{\epsilon + \sqrt{c S_{t-1}}} \le \frac{A}{\epsilon} + \frac{2}{\sqrt{c}} \sqrt{\sum_{t=r_k}^{s_k} a_t}. \tag{27}$$

Combining (26) and (27), and then applying Cauchy–Schwarz inequality, we obtain

$$\sum_{t=1}^{T} \frac{a_t}{\epsilon + \sqrt{V_{t-1}}} \le K \cdot \frac{A}{\epsilon} + \frac{2}{\sqrt{c}} \sum_{k=1}^{K} \sqrt{\sum_{t=r_k}^{s_k} a_t} \le K \cdot \frac{A}{\epsilon} + \frac{2}{\sqrt{c}} \sqrt{K} \sqrt{\sum_{t=1}^{T} a_t}. \tag{28}$$

It remains to choose $\tau$. Let $\tau = \left\lceil \frac{\ln 2}{1-\beta} \right\rceil$. Then $\tau - 1 < \frac{\ln 2}{1-\beta}$, so $\beta^{\tau-1} > \beta^{\frac{\ln 2}{1-\beta}}$. Moreover, for $\beta \in (0,1)$, we have $\ln \beta \geq -\frac{1-\beta}{\beta}$, which implies

$$\beta^{\frac{\ln 2}{1-\beta}} = \exp\left( \frac{\ln 2}{1-\beta} \ln \beta \right) \geq \exp\left( \frac{\ln 2}{1-\beta} \cdot \left( -\frac{1-\beta}{\beta} \right) \right) = 2^{-1/\beta}.$$

Hence $\beta^{\tau-1} \geq 2^{-1/\beta}$ and thus $c = (1-\beta)\beta^{\tau-1} \geq (1-\beta)2^{-1/\beta}$. Also, since $K = \lceil T/\tau \rceil \leq T/\tau + 1$ and $\tau \geq \ln 2/(1-\beta)$, we have

$$K \leq \frac{T(1-\beta)}{\ln 2} + 1.$$

Substituting these bounds for $c$ and $K$ into (28) completes the proof. $\qquad\square$

**Lemma 10.** *Let $\beta \in (0,1)$, $\epsilon > 0$, and let $\{a_t\}_{t=1}^T$ satisfy $0 \leq a_t \leq A$ for all $t \in [T]$. Define the exponential moving average $V_0 = 0, V_t = \beta V_{t-1} + (1-\beta)a_t$ for $t \geq 1$. Then,*

$$\sum_{t=1}^T \frac{a_t}{\epsilon + \sqrt{V_t}} \leq \frac{2}{\sqrt{(1-\beta)2^{-1/\beta}}} \sqrt{\left( \frac{T(1-\beta)}{\ln 2} + 1 \right) \sum_{t=1}^T a_t}.$$

*Proof sketch.* The argument follows the proof of Lemma 9, with one change: we apply Lemma 24 to Eq. (26) instead of using Lemma 23. Therefore, we omit the details. $\qquad\square$

**Lemma 11.** *Let $\beta_1, \beta_2 \in (0,1)$. Let $\{\epsilon_t\}_{t\geq 0}$ be a nonnegative nondecreasing sequence, and let $\{\mathbf{g}_t\}_{t\geq 1} \subseteq \mathbb{R}^d$ be arbitrary. Define*

$$V_0 = 0, \qquad V_t = (1-\beta_2)\sum_{s=1}^t \beta_2^{t-s}\|\mathbf{g}_s\|_2^2 \quad (t \geq 1), \qquad \alpha_t = \frac{1}{\epsilon_t + \sqrt{V_t}} \quad (t \geq 0).$$

*Then, for every $t \geq 1$,*

$$\left[ \frac{\beta_1}{\alpha_{t-1}} - \frac{1}{\alpha_t} \right]_+ \leq [\beta_1 - \sqrt{\beta_2}]_+\sqrt{V_{t-1}},$$

*where $[x]_+ = \max\{x, 0\}$.*

*Proof.* By definition of $\alpha_t$, we have

$$\begin{aligned}
\frac{\beta_1}{\alpha_{t-1}} - \frac{1}{\alpha_t} &= \beta_1\left( \epsilon_{t-1} + \sqrt{V_{t-1}} \right) - \left( \epsilon_t + \sqrt{V_t} \right) \\
&= \underbrace{\left( \beta_1\epsilon_{t-1} - \epsilon_t \right)}_{\leq 0} + \left( \beta_1\sqrt{V_{t-1}} - \sqrt{V_t} \right) \leq \beta_1\sqrt{V_{t-1}} - \sqrt{V_t},
\end{aligned}$$

where the inequality uses $\epsilon_{t-1} \leq \epsilon_t$ and $\beta_1 \leq 1$.

Next, $V_t = \beta_2 V_{t-1} + (1-\beta_2)\|\mathbf{g}_t\|_2^2 \geq \beta_2 V_{t-1}$ implies $\sqrt{V_t} \geq \sqrt{\beta_2}\sqrt{V_{t-1}}$. Therefore,

$$\frac{\beta_1}{\alpha_{t-1}} - \frac{1}{\alpha_t} \leq (\beta_1 - \sqrt{\beta_2})\sqrt{V_{t-1}}.$$

Taking $[\cdot]_+$ on both sides proves the lemma. $\qquad\square$

**Lemma 12.** *Let $\beta_1, \beta_2 \in (0,1)$, $\gamma > 0$, $\epsilon > 0$, and $\mu \geq 0$. Let $\{\mathbf{g}_t\}_{t=1}^{T-1} \subseteq \mathbb{R}^d$ be arbitrary, and define*

$$V_0 = 0, \qquad V_t = (1-\beta_2)\sum_{s=1}^t \beta_2^{t-s}\|\mathbf{g}_s\|_2^2 \quad (t \geq 1).$$

*For $t \geq 0$, define*

$$\alpha_t = \frac{1}{\epsilon + \gamma\mu(1 - \beta_1^t) + \sqrt{V_t}}, \qquad \eta_t = \gamma(1 - \beta_1)\beta_1^t\alpha_t.$$

*Then for any $T \geq 2$,*

$$\sum_{t=1}^{T-1} \beta_1^t\left(\frac{1}{\eta_t} - \frac{1}{\eta_{t-1}}\right) \leq \frac{\sqrt{V_{T-1}}}{\gamma(1 - \beta_1)} + \frac{1}{\gamma}\sum_{t=1}^{T-2} \sqrt{V_t} + \frac{(T-1)\epsilon}{\gamma} + \mu(T - 1).$$

*Proof.* Let $b_t = 1/\eta_t$. By a direct rearrangement of the summation,

$$\sum_{t=1}^{T-1} \beta_1^t(b_t - b_{t-1}) = \beta_1^{T-1}b_{T-1} - \beta_1 b_0 + (1 - \beta_1)\sum_{t=1}^{T-2} \beta_1^t b_t, \tag{29}$$

which follows from

$$\sum_{t=1}^{T-1} \beta_1^t(b_t - b_{t-1}) = \sum_{t=1}^{T-1} \beta_1^t b_t - \sum_{t=1}^{T-1} \beta_1^t b_{t-1} = \sum_{t=1}^{T-1} \beta_1^t b_t - \sum_{t=0}^{T-2} \beta_1^{t+1} b_t.$$

By the definition of $\eta_t$ and $\alpha_t$, for every $t \geq 0$,

$$\beta_1^t b_t = \beta_1^t \cdot \frac{1}{\gamma(1 - \beta_1)\beta_1^t\alpha_t} = \frac{1}{\gamma(1 - \beta_1)} \cdot \frac{1}{\alpha_t} = \frac{\epsilon + \gamma\mu(1 - \beta_1^t) + \sqrt{V_t}}{\gamma(1 - \beta_1)}.$$

In particular, $V_0 = 0$ and $\beta_1^0 = 1$ give $b_0 = \epsilon/(\gamma(1 - \beta_1))$. Substituting into (29) gives

$$\sum_{t=1}^{T-1} \beta_1^t(b_t - b_{t-1}) = \frac{\sqrt{V_{T-1}}}{\gamma(1 - \beta_1)} + \frac{1}{\gamma}\sum_{t=1}^{T-2} \sqrt{V_t} + \frac{(T-1)\epsilon}{\gamma} + \mu\left(\frac{1 - \beta_1^{T-1}}{1 - \beta_1} + \sum_{t=1}^{T-2}(1 - \beta_1^t)\right). \tag{30}$$

We simplify the last term by observing

$$\frac{1 - \beta_1^{T-1}}{1 - \beta_1} + \sum_{t=1}^{T-2}(1 - \beta_1^t) = \sum_{t=0}^{T-2} \beta_1^t + \sum_{t=1}^{T-2} 1 - \sum_{t=1}^{T-2} \beta_1^t = 1 + (T - 2) = T - 1.$$

Combining this with (30) proves the lemma. $\qquad\square$

**Lemma 13.** *Let $\beta_1, \beta_2 \in (0, 1)$ satisfy $\beta_2 > \beta_1^2$, and let $\gamma > 0$, $\nu > 0$, and $\mu \geq 0$. Let $\{\mathbf{g}_t\}_{t \geq 1} \subseteq \mathbb{R}^d$ be arbitrary, and define for $t \geq 1$*

$$V_t = (1 - \beta_2)\sum_{s=1}^{t} \beta_2^{t-s}\|\mathbf{g}_s\|_2^2, \qquad A_t = \nu + \gamma\mu(1 - \beta_1^t) + \sqrt{V_t}, \qquad \eta_t = \gamma(1 - \beta_1)\frac{\beta_1^t}{A_t}.$$

*Then for every $t \geq 2$,*

$$\left|\eta_{t-1} - \eta_t\right| \cdot \left\|\sum_{s=1}^{t-1} \beta_1^{t-1-s}\mathbf{g}_s\right\|_2 \leq \gamma(1 - \beta_1)\beta_1^{t-1} \cdot \frac{A_t - \beta_1 A_{t-1}}{A_t} \cdot \sqrt{\frac{\beta_2}{(\beta_2 - \beta_1^2)(1 - \beta_2)}}.$$

*Proof.* We first rewrite the difference of step sizes:

$$\eta_{t-1} - \eta_t = \gamma(1 - \beta_1)\beta_1^{t-1}\left(\frac{1}{A_{t-1}} - \frac{\beta_1}{A_t}\right) = \gamma(1 - \beta_1)\beta_1^{t-1} \cdot \frac{A_t - \beta_1 A_{t-1}}{A_{t-1}A_t}.$$

Under $\beta_2 \geq \beta_1^2$, we have $\sqrt{V_t} \geq \sqrt{\beta_2}\sqrt{V_{t-1}} \geq \beta_1\sqrt{V_{t-1}}$. Hence

$$A_t - \beta_1 A_{t-1} = \left(\nu + \gamma\mu(1 - \beta_1^t) + \sqrt{V_t}\right) - \beta_1\left(\nu + \gamma\mu(1 - \beta_1^{t-1}) + \sqrt{V_{t-1}}\right)$$

$$= (1 - \beta_1)(\nu + \gamma\mu) + \left(\sqrt{V_t} - \beta_1\sqrt{V_{t-1}}\right) \geq 0,$$

which implies $\eta_{t-1} \geq \eta_t$ and thus $|\eta_{t-1} - \eta_t| = \eta_{t-1} - \eta_t$.

Next, by Cauchy–Schwarz inequality, letting $k = t - 1 - s$,

$$\left\| \sum_{s=1}^{t-1} \beta_1^{t-1-s} \mathbf{g}_s \right\|_2 = \left\| \sum_{k=0}^{t-2} \beta_1^k \mathbf{g}_{t-1-k} \right\|_2 = \left\| \sum_{k=0}^{t-2} \left(\frac{\beta_1^2}{\beta_2}\right)^{k/2} \cdot \beta_2^{k/2} \mathbf{g}_{t-1-k} \right\|_2$$

$$\leq \sqrt{\sum_{k=0}^{t-2} \left(\frac{\beta_1^2}{\beta_2}\right)^k} \cdot \sqrt{\sum_{k=0}^{t-2} \beta_2^k \|\mathbf{g}_{t-1-k}\|_2^2} = \sqrt{\sum_{k=0}^{t-2} \left(\frac{\beta_1^2}{\beta_2}\right)^k} \cdot \sqrt{\frac{V_{t-1}}{1 - \beta_2}}.$$

Combining the last display with the expression of $|\eta_{t-1} - \eta_t|$ and using $\sqrt{V_{t-1}} \leq A_{t-1}$ gives

$$|\eta_{t-1} - \eta_t| \cdot \left\| \sum_{s=1}^{t-1} \beta_1^{t-1-s} \mathbf{g}_s \right\|_2 \leq \gamma(1 - \beta_1)\beta_1^{t-1} \cdot \frac{A_t - \beta_1 A_{t-1}}{A_{t-1} A_t} \cdot \sqrt{\frac{V_{t-1}}{1 - \beta_2}} \cdot \sqrt{\sum_{k=0}^{t-2} \left(\frac{\beta_1^2}{\beta_2}\right)^k}$$

$$\leq \gamma(1 - \beta_1)\beta_1^{t-1} \cdot \frac{A_t - \beta_1 A_{t-1}}{A_t} \cdot \sqrt{\frac{1}{1 - \beta_2} \sum_{k=0}^{t-2} \left(\frac{\beta_1^2}{\beta_2}\right)^k}$$

$$\leq \gamma(1 - \beta_1)\beta_1^{t-1} \cdot \frac{A_t - \beta_1 A_{t-1}}{A_t} \cdot \sqrt{\frac{\beta_2}{(\beta_2 - \beta_1^2)(1 - \beta_2)}},$$

which finishes the proof. $\qquad\square$

**Lemma 14.** *Let $\beta \in (0, 1)$ and let $\mathcal{X} \subseteq \mathbb{R}^d$ be a nonempty closed convex set. Consider linear losses $f_t(\mathbf{x}) = \langle \mathbf{g}_t, \mathbf{x} \rangle$ with $\mathbf{g}_t \in \mathbb{R}^d$. Run the* rescaled *discounted FTRL*

$$\mathbf{x}_{t+1} \in \arg\min_{\mathbf{x} \in \mathcal{X}} \left\{ \sum_{s=1}^{t} \langle \beta^{-s} \mathbf{g}_s, \mathbf{x} \rangle + \psi_{t+1}(\mathbf{x}) \right\}, \qquad \psi_{t+1}(\mathbf{x}) = \frac{1}{2\eta_t} \|\mathbf{x}\|_2^2,$$

*where $\{\eta_t\}_{t \geq 0}$ is any positive stepsize sequence. Let $\mathbf{u}_1, \ldots, \mathbf{u}_T \in \mathcal{X}$ be any comparator sequence. Define the diameter $D_{\mathcal{X}} = \max_{\mathbf{x}, \mathbf{y} \in \mathcal{X}} \|\mathbf{x} - \mathbf{y}\|_2$ and the rescaled cumulative gradients*

$$\widetilde{\mathbf{g}}_{1:t} = \sum_{s=1}^{t} \beta^{-s} \mathbf{g}_s.$$

*Then the following dynamic-regret decomposition holds:*

$$\sum_{t=1}^{T} \langle \mathbf{g}_t, \mathbf{x}_t - \mathbf{u}_t \rangle \leq \frac{\beta}{2\eta_0} \|\mathbf{u}_1\|_2^2 + \sum_{t=1}^{T} \beta^t \Lambda_t + \sum_{t=1}^{T} \beta^t \left( \psi_t(\mathbf{x}_{t+1}) - \psi_{t+1}(\mathbf{x}_{t+1}) \right)$$

$$+ \beta \sum_{t=1}^{T-1} \sum_{s=1}^{t} \beta^{t-s} \langle \mathbf{g}_s, \mathbf{u}_{t+1} - \mathbf{u}_t \rangle + \beta \sum_{t=1}^{T-1} \beta^t \left( \psi_{t+1}(\mathbf{u}_{t+1}) - \psi_t(\mathbf{u}_t) \right).$$

*Moreover, the one-step stability term $\Lambda_t$ can be instantiated in* either *of the following two ways:*

**Option I:** *For every $t \in [T]$, $\Lambda_t = \frac{\eta_{t-1}}{2} \|\beta^{-t} \mathbf{g}_t\|_2^2 = \frac{1}{2} \eta_{t-1} \beta^{-2t} \|\mathbf{g}_t\|_2^2$*

**Option II:** *For every $t \in [T]$,*

$$\Lambda_t = \eta_t \|\beta^{-t} \mathbf{g}_t\|_2^2 + \min\left\{ \frac{\eta_{t-1}}{2} \|\beta^{-t} \mathbf{g}_t\|_2^2, \ \min\left\{ D_{\mathcal{X}}, \ |\eta_{t-1} - \eta_t| \|\widetilde{\mathbf{g}}_{1:t-1}\|_2 \right\} \cdot \|\beta^{-t} \mathbf{g}_t\|_2 \right\}.$$

*Proof.* Apply Lemma 19 to the rescaled linear losses

$$\widetilde{f}_t(\mathbf{x}) = \langle \beta^{-t} \mathbf{g}_t, \mathbf{x} \rangle \qquad (t \in [T]),$$

with regularizers $\psi_{t+1}(\mathbf{x}) = \frac{1}{2\eta_t}\|\mathbf{x}\|_2^2$. This provides the standard decomposition in terms of the one-step quantities

$$F_t(\mathbf{x}_t) - F_{t+1}(\mathbf{x}_{t+1}) + \widetilde{f}_t(\mathbf{x}_t) \quad \text{and} \quad \psi_t(\mathbf{x}_{t+1}) - \psi_{t+1}(\mathbf{x}_{t+1}).$$

Next, we upper bound the one-step quantity in two different ways:

**Option I:** By Lemma 20 we have,

$$F_t(\mathbf{x}_t) - F_{t+1}(\mathbf{x}_{t+1}) + \widetilde{f}_t(\mathbf{x}_t) \leq \frac{\eta_{t-1}}{2}\|\beta^{-t}\mathbf{g}_t\|_2^2 + \psi_t(\mathbf{x}_{t+1}) - \psi_{t+1}(\mathbf{x}_{t+1}).$$

**Option II:** Alternatively, applying Lemma 21 to the rescaled gradients $\{\beta^{-t}\mathbf{g}_t\}_{t=1}^T$ gives

$$F_t(\mathbf{x}_t) - F_{t+1}(\mathbf{x}_{t+1}) + \widetilde{f}_t(\mathbf{x}_t) \leq \Lambda_t + \psi_t(\mathbf{x}_{t+1}) - \psi_{t+1}(\mathbf{x}_{t+1}).$$

Combining the above analysis with Theorem 1 finishes the proof. $\qquad\square$

**Lemma 15.** *Fix $\beta \in (0,1)$ and $\mu \geq 0$. Let $\mathcal{X} \subseteq \mathbb{R}^d$ be a nonempty closed convex set and let $\{\mathbf{g}_t\}_{t\geq 1} \subseteq \mathbb{R}^d$ be arbitrary. Define the composite surrogate loss $\ell_t(\mathbf{x}) = \langle \mathbf{g}_t, \mathbf{x}\rangle + \frac{\mu}{2}\|\mathbf{x}\|_2^2$. Consider the discounted FTRL update*

$$\mathbf{x}_{t+1} \in \arg\min_{\mathbf{x}\in\mathcal{X}} \left\{ \sum_{s=1}^t \beta^{-s}\ell_s(\mathbf{x}) + \frac{1}{2\eta_t}\|\mathbf{x}\|_2^2 \right\},$$

*which is equivalent to*

$$\mathbf{x}_{t+1} \in \arg\min_{\mathbf{x}\in\mathcal{X}} \left\{ \sum_{s=1}^t \langle\beta^{-s}\mathbf{g}_s, \mathbf{x}\rangle + \frac{1}{2\eta_t'}\|\mathbf{x}\|_2^2 \right\}.$$

*with $\frac{1}{\eta_t'} = \frac{1}{\eta_t} + \mu\sum_{s=1}^t \beta^{-s}$. For any comparator sequence $\mathbf{u}_1,\ldots,\mathbf{u}_T \in \mathcal{X}$, it ensures:*

$$\sum_{t=1}^T \ell_t(\mathbf{x}_t) - \sum_{t=1}^T \ell_t(\mathbf{u}_t) \leq \frac{\beta}{2\eta_0'}\|\mathbf{u}_1\|_2^2 + \sum_{t=1}^T \beta^t\Lambda_t + \sum_{t=1}^T \beta^t\left(\frac{1}{2\eta_{t-1}}\|\mathbf{x}_{t+1}\|_2^2 - \frac{1}{2\eta_t}\|\mathbf{x}_{t+1}\|_2^2\right)$$

$$+ \beta\sum_{t=1}^{T-1}\sum_{s=1}^t \beta^{t-s}(\ell_s(\mathbf{u}_{t+1}) - \ell_s(\mathbf{u}_t)) + \beta\sum_{t=1}^{T-1} \beta^t\left(\frac{1}{2\eta_t}\|\mathbf{u}_{t+1}\|_2^2 - \frac{1}{2\eta_{t-1}}\|\mathbf{u}_t\|_2^2\right),$$

*where $D_{\mathcal{X}} = \max_{\mathbf{x},\mathbf{y}\in\mathcal{X}} \|\mathbf{x}-\mathbf{y}\|_2$, $\widetilde{\mathbf{g}}_{1:t} = \sum_{s=1}^t \beta^{-s}\mathbf{g}_s$, and $\Lambda_t$ can be instantiated in* either *of the following two ways:*

**Option I:** *For every $t \in [T]$, $\Lambda_t = \frac{\eta_{t-1}'}{2}\|\beta^{-t}\mathbf{g}_t\|_2^2$.*

**Option II:** *For every $t \in [T]$,*

$$\Lambda_t = \eta_t'\|\beta^{-t}\mathbf{g}_t\|_2^2 + \min\left\{ \frac{\eta_{t-1}'}{2}\|\beta^{-t}\mathbf{g}_t\|_2^2,\ \min\left\{ D_{\mathcal{X}},\ |\eta_{t-1}' - \eta_t'|\|\widetilde{\mathbf{g}}_{1:t-1}\|_2 \right\} \cdot \|\beta^{-t}\mathbf{g}_t\|_2 \right\}.$$

*Proof.* The discounted FTRL update on $\ell_s(\cdot)$ with regularizer $\frac{1}{2\eta_t}\|\cdot\|_2^2$ is equivalent to running FTRL on the linearized losses $\langle\mathbf{g}_s, \cdot\rangle$ with the regularizer $\frac{1}{2\eta_t'}\|\cdot\|_2^2$, where $\frac{1}{\eta_t'} = \frac{1}{\eta_t} + \mu\sum_{s=1}^t \beta^{-s}$. Applying Lemma 14 (Option I or II) to the linearized losses with stepsizes $\{\eta_t'\}$ gives

$$\sum_{t=1}^T \langle\mathbf{g}_t, \mathbf{x}_t - \mathbf{u}_t\rangle \leq \frac{\beta}{2\eta_0'}\|\mathbf{u}_1\|_2^2 + \sum_{t=1}^T \beta^t\Lambda_t + \sum_{t=1}^T \beta^t\left(\frac{1}{2\eta_{t-1}'} - \frac{1}{2\eta_t'}\right)\|\mathbf{x}_{t+1}\|_2^2$$

$$+ \beta\sum_{t=1}^{T-1}\sum_{s=1}^t \beta^{t-s}\langle\mathbf{g}_s, \mathbf{u}_{t+1} - \mathbf{u}_t\rangle + \beta\sum_{t=1}^{T-1} \beta^t\left(\frac{1}{2\eta_t'}\|\mathbf{u}_{t+1}\|_2^2 - \frac{1}{2\eta_{t-1}'}\|\mathbf{u}_t\|_2^2\right), \qquad (31)$$

where $\Lambda_t$ is determined by the choice of Option I or II in Lemma 14 with stepsizes $\{\eta_t'\}$. We now convert every $\eta'$ to $\eta$ and recover the composite losses $\ell_s$. Since $\frac{1}{\eta_{t-1}'} - \frac{1}{\eta_t'} = \left(\frac{1}{\eta_{t-1}} - \frac{1}{\eta_t}\right) - \mu\beta^{-t}$, the $\mathbf{x}$-telescoping term satisfies

$$\beta^t\left(\frac{1}{2\eta_{t-1}'} - \frac{1}{2\eta_t'}\right)\|\mathbf{x}_{t+1}\|_2^2 = \beta^t\left(\frac{1}{2\eta_{t-1}} - \frac{1}{2\eta_t}\right)\|\mathbf{x}_{t+1}\|_2^2 - \frac{\mu}{2}\|\mathbf{x}_{t+1}\|_2^2.$$

For the path length term, we use $\frac{\beta^t}{2\eta_t'} = \frac{\beta^t}{2\eta_t} + \frac{\mu}{2}\sum_{s=1}^t \beta^{t-s}$ to split

$$\beta^t\left(\frac{1}{2\eta_t'}\|\mathbf{u}_{t+1}\|_2^2 - \frac{1}{2\eta_{t-1}'}\|\mathbf{u}_t\|_2^2\right) = \beta^t\left(\frac{1}{2\eta_t}\|\mathbf{u}_{t+1}\|_2^2 - \frac{1}{2\eta_{t-1}}\|\mathbf{u}_t\|_2^2\right) + \frac{\mu}{2}\sum_{s=1}^t \beta^{t-s}\left(\|\mathbf{u}_{t+1}\|_2^2 - \|\mathbf{u}_t\|_2^2\right) + \frac{\mu}{2}\|\mathbf{u}_t\|_2^2.$$

The quadratic terms from the path length combine with the linearized terms to form $\ell(\cdot)$:

$$\beta\sum_{t=1}^{T-1}\sum_{s=1}^t \beta^{t-s}\left[\langle\mathbf{g}_s, \mathbf{u}_{t+1} - \mathbf{u}_t\rangle + \frac{\mu}{2}\left(\|\mathbf{u}_{t+1}\|_2^2 - \|\mathbf{u}_t\|_2^2\right)\right] = \beta\sum_{t=1}^{T-1}\sum_{s=1}^t \beta^{t-s}\left(\ell_s(\mathbf{u}_{t+1}) - \ell_s(\mathbf{u}_t)\right).$$

Substituting all splittings into (31) and adding $\frac{\mu}{2}\sum_{t=1}^T(\|\mathbf{x}_t\|_2^2 - \|\mathbf{u}_t\|_2^2)$ to both sides converts the LHS to $\sum_{t=1}^T \ell_t(\mathbf{x}_t) - \sum_{t=1}^T \ell_t(\mathbf{u}_t)$. Two non-positive remainder terms appear and can be dropped: using $\mathbf{x}_1 = \mathbf{0}$ by definition,

$$\frac{\mu}{2}\sum_{t=1}^T\|\mathbf{x}_t\|_2^2 - \frac{\mu}{2}\sum_{t=1}^T\|\mathbf{x}_{t+1}\|_2^2 \leq 0, \quad \frac{\mu\beta}{2}\sum_{t=1}^{T-1}\|\mathbf{u}_t\|_2^2 - \frac{\mu}{2}\sum_{t=1}^T\|\mathbf{u}_t\|_2^2 \leq 0.$$

Combining everything yields the claimed bound. $\qquad\square$

### D.6. Proof of Theorem 4

*Proof.* Throughout the proof, we use $\beta_1$ for the first-moment discount factor in Adam and the discount factor in Algorithm 1, and $\beta_2$ for the second-moment terms.

**O2NC Reduction.** By Lemma 8, we have

$$\mathbb{E}\|\nabla F(\bar{\mathbf{x}})\|_c \leq \frac{F^*}{DT} + \frac{2G + \sigma}{(1 - \beta_1)T} + \sigma\sqrt{1 - \beta_1} + \frac{12cD^2}{(1 - \beta_1)^2}$$

$$+ \frac{1}{DT}\mathbb{E}\left[\text{D-REG}_T(\mathbf{u}_{1:T}) + \beta_1\sum_{t=1}^{T-1}\sum_{s=1}^t \beta_1^{t-s}\left(\ell_s(\mathbf{u}_t) - \ell_s(\mathbf{u}_{t+1})\right)\right], \tag{32}$$

where $\ell_t(\Delta) = \langle\mathbf{g}_t, \Delta\rangle$ and $\text{D-REG}_T(\mathbf{u}_{1:T}) = \sum_{t=1}^T\langle\mathbf{g}_t, \Delta_t - \mathbf{u}_t\rangle$.

**Use D2D Reduction.** We apply Lemma 14 to the discounted FTRL run on $\mathcal{D} = \{\Delta : \|\Delta\|_2 \leq D\}$, with decisions $\Delta_t$, gradients $\mathbf{g}_t$, and comparator sequence $\mathbf{u}_t$ in Lemma 8. Recall the stepsizes (Algorithm 1 with clipped Adam):

$$V_0 = 0, \qquad V_t = (1 - \beta_2)\sum_{s=1}^t \beta_2^{t-s}\|\mathbf{g}_s\|_2^2, \qquad \alpha_t = \frac{1}{\nu + \sqrt{V_t}}, \qquad \eta_t = \gamma(1 - \beta_1)\beta_1^t\alpha_t.$$

Since $\|\mathbf{u}_t\|_2 = D$ for all $t$ by construction, the path length term in (32) cancels $\beta_1\sum_{t=1}^{T-1}\sum_{s=1}^t \beta_1^{t-s}\langle\mathbf{g}_s, \mathbf{u}_{t+1} - \mathbf{u}_t\rangle$ brought by Lemma 14, and we obtain the decomposition

$$\text{D-REG}_T(\mathbf{u}_{1:T}) + \beta_1\sum_{t=1}^{T-1}\sum_{s=1}^t \beta_1^{t-s}\left(\ell_s(\mathbf{u}_t) - \ell_s(\mathbf{u}_{t+1})\right)$$

$$\leq \underbrace{\frac{1 - \beta_1}{2\beta_1}\gamma\sum_{t=1}^T \alpha_{t-1}\|\mathbf{g}_t\|_2^2 + \frac{\beta_1}{2\gamma(1-\beta_1)\alpha_0}\|\mathbf{u}_1\|_2^2}_{\text{TERM-A}} + \underbrace{\beta_1\sum_{t=1}^{T-1}\left(\frac{\beta_1^t}{2\eta_t}\|\mathbf{u}_{t+1}\|_2^2 - \frac{\beta_1^t}{2\eta_{t-1}}\|\mathbf{u}_t\|_2^2\right)}_{\text{TERM-B}}$$

$$+ \sum_{t=1}^{T} \underbrace{\left( \frac{\beta_1^t}{2\eta_{t-1}} \|\Delta_{t+1}\|_2^2 - \frac{\beta_1^t}{2\eta_t} \|\Delta_{t+1}\|_2^2 \right)}_{\text{TERM-C}}. \tag{33}$$

**TERM-A.** We first bound $\sum_{t=1}^{T} \alpha_{t-1} \|\mathbf{g}_t\|_2^2$ via Lemma 9. Apply Lemma 9 with

$$a_t = \|\mathbf{g}_t\|_2^2, \qquad \epsilon = \nu, \qquad \beta = \beta_2, \qquad V_t = \beta_2 V_{t-1} + (1-\beta_2) a_t.$$

Let $K \geq \frac{T(1-\beta_2)}{\ln 2} + 1$, then

$$\sum_{t=1}^{T} \alpha_{t-1} \|\mathbf{g}_t\|_2^2 = \sum_{t=1}^{T} \frac{\|\mathbf{g}_t\|_2^2}{\nu + \sqrt{V_{t-1}}} \leq K \frac{A}{\nu} + \frac{2}{\sqrt{(1-\beta_2) 2^{-1/\beta_2}}} \sqrt{K \sum_{t=1}^{T} \|\mathbf{g}_t\|_2^2}, \tag{34}$$

where $A$ is any almost-sure upper bound on $\|\mathbf{g}_t\|_2^2$. Under Assumptions 1 − 4, we take $A = G^2$. Taking expectation in (34) and using Jensen, $\mathbb{E}\left[\sqrt{\sum_{t=1}^{T} \|\mathbf{g}_t\|_2^2}\right] \leq \sqrt{\mathbb{E} \sum_{t=1}^{T} \|\mathbf{g}_t\|_2^2}$, gives

$$\mathbb{E} \sum_{t=1}^{T} \alpha_{t-1} \|\mathbf{g}_t\|_2^2 \leq K \frac{G^2}{\nu} + \frac{2}{\sqrt{(1-\beta_2) 2^{-1/\beta_2}}} \sqrt{KT} G,$$

where $K = \frac{T(1-\beta_2)}{\ln 2} + 1$. We let $T$ satisfy $\frac{T(1-\beta_2)}{\ln 2} \geq 1$, then $K \leq \frac{2T(1-\beta_2)}{\ln 2}$, therefore,

$$\mathbb{E} \sum_{t=1}^{T} \alpha_{t-1} \|\mathbf{g}_t\|_2^2 \leq \frac{2T(1-\beta_2)}{\ln 2} \cdot \frac{G^2}{\nu} + \frac{2}{\sqrt{(1-\beta_2) 2^{-1/\beta_2}}} \cdot \sqrt{\frac{2T^2(1-\beta_2)}{\ln 2}} G$$

$$= \frac{2T(1-\beta_2)}{\ln 2} \cdot \frac{G^2}{\nu} + \frac{2\sqrt{2/\ln 2}}{\sqrt{2^{-1/\beta_2}}} TG.$$

Since $\|\mathbf{u}_1\|_2 = D$ and $\alpha_0 = \frac{1}{\nu}$, plugging the above inequality into TERM-A gives

$$\mathbb{E}[\text{TERM-A}] \leq \frac{1-\beta_1}{2\beta_1} \gamma \left( \frac{2T(1-\beta_2)}{\ln 2} \cdot \frac{G^2}{\nu} + \frac{2\sqrt{2/\ln 2}}{\sqrt{2^{-1/\beta_2}}} TG \right) + \frac{\beta_1 \nu}{2\gamma(1-\beta_1)} D^2. \tag{35}$$

**TERM-B.** Since $\|\mathbf{u}_t\|_2 = D$ for all $t$,

$$\text{TERM-B} = \beta_1 \sum_{t=1}^{T-1} \left( \frac{\beta_1^t}{2\eta_t} D^2 - \frac{\beta_1^t}{2\eta_{t-1}} D^2 \right) = \frac{\beta_1 D^2}{2} \sum_{t=1}^{T-1} \beta_1^t \left( \frac{1}{\eta_t} - \frac{1}{\eta_{t-1}} \right).$$

By Lemma 12 with $\epsilon = \nu$,

$$\text{TERM-B} \leq \frac{\beta_1 D^2}{2} \left( \frac{\sqrt{V_{T-1}}}{\gamma(1-\beta_1)} + \frac{1}{\gamma} \sum_{t=1}^{T-2} \sqrt{V_t} + \frac{(T-1)\nu}{\gamma} \right). \tag{36}$$

Next we bound the EMA terms in expectation. First, by Jensen, $\mathbb{E}[\sqrt{V_t}] \leq \sqrt{\mathbb{E}[V_t]}$. Moreover,

$$\mathbb{E}[V_t] \leq (1-\beta_2) \sum_{s=1}^{t} \beta_2^{t-s} \mathbb{E} \|\mathbf{g}_s\|_2^2 \leq G^2,$$

so $\mathbb{E}[\sqrt{V_t}] \leq G$ for all $t$. Also, using Cauchy–Schwarz inequality and Lemma 16,

$$\mathbb{E} \sum_{t=1}^{T-2} \sqrt{V_t} \leq \sqrt{T} \cdot \sqrt{\mathbb{E} \sum_{t=1}^{T} V_t} = \sqrt{T} \cdot \sqrt{\mathbb{E} \sum_{s=1}^{T} (1 - \beta_2^{T+1-s}) \|\mathbf{g}_s\|_2^2} \leq \sqrt{T} \cdot \sqrt{\sum_{s=1}^{T} \mathbb{E} \|\mathbf{g}_s\|_2^2} \leq TG.$$

Taking expectation in (36) gives

$$\mathbb{E}[\text{TERM-B}] \leq \frac{\beta_1 D^2}{2} \left( \frac{G}{\gamma(1-\beta_1)} + \frac{TG}{\gamma} + \frac{(T-1)\nu}{\gamma} \right).$$

**TERM-C.** Using $\eta_t = \gamma(1 - \beta_1)\beta_1^t \alpha_t$ and $\|\Delta_{t+1}\|_2 \leq D$,

$$\text{TERM-C} = \sum_{t=1}^{T} \left( \frac{\beta_1^t}{2\eta_{t-1}} - \frac{\beta_1^t}{2\eta_t} \right) \|\Delta_{t+1}\|_2^2 = \frac{1}{2\gamma(1-\beta_1)} \sum_{t=1}^{T} \left( \frac{\beta_1}{\alpha_{t-1}} - \frac{1}{\alpha_t} \right) \|\Delta_{t+1}\|_2^2$$

$$\leq \frac{D^2}{2\gamma(1-\beta_1)} \sum_{t=1}^{T} \left[ \frac{\beta_1}{\alpha_{t-1}} - \frac{1}{\alpha_t} \right]_+.$$

By Lemma 11, $\left[ \frac{\beta_1}{\alpha_{t-1}} - \frac{1}{\alpha_t} \right]_+ \leq [\beta_1 - \sqrt{\beta_2}]_+ \sqrt{V_{t-1}}$. Hence

$$\text{TERM-C} \leq \frac{D^2}{2\gamma(1-\beta_1)} [\beta_1 - \sqrt{\beta_2}]_+ \sum_{t=1}^{T} \sqrt{V_{t-1}}.$$

Taking expectation and using $\mathbb{E} \sum_{t=1}^{T} \sqrt{V_{t-1}} \leq TG$ gives

$$\mathbb{E}[\text{TERM-C}] \leq \frac{D^2}{2\gamma(1-\beta_1)} [\beta_1 - \sqrt{\beta_2}]_+ TG.$$

Under the condition $\beta_2 \geq \beta_1^4$ we have $\sqrt{\beta_2} \geq \beta_1^2$ and thus

$$[\beta_1 - \sqrt{\beta_2}]_+ \leq \beta_1 - \beta_1^2 = \beta_1(1 - \beta_1).$$

Substituting into the above step for $\mathbb{E}[\text{TERM-C}]$ gives

$$\mathbb{E}[\text{TERM-C}] \leq \frac{\beta_1 D^2}{2\gamma} TG.$$

**Combine and Tune Parameters.** Combining (32) with (33), we obtain

$$\mathbb{E}\|\nabla F(\bar{\mathbf{x}})\|_c \leq \frac{F^*}{DT} + \frac{2G + \sigma}{(1 - \beta_1)T} + \sigma\sqrt{1 - \beta_1} + \frac{12cD^2}{(1 - \beta_1)^2}$$

$$+ \frac{1}{DT} \mathbb{E}[\text{TERM-A} + \text{TERM-B} + \text{TERM-C}]. \tag{37}$$

Now tune the parameters as in the theorem. Set

$$\beta_1 = 1 - \left( \frac{\epsilon}{16(G + \sigma)} \right)^2, \qquad D = \frac{(1 - \beta_1)\sqrt{\epsilon}}{\sqrt{48c}}, \qquad \gamma = \frac{\beta_1 D}{\sqrt{1 - \beta_1}}, \qquad \beta_2 \geq \max\left\{ 1 - \frac{\nu}{G + \sigma}, \beta_1^4 \right\}.$$

Then $\frac{12cD^2}{(1 - \beta_1)^2} = \epsilon/4$ and $\sigma\sqrt{1 - \beta_1} \leq (G + \sigma)\sqrt{1 - \beta_1} = \epsilon/16$. We next upper bound the FTRL contribution $\frac{1}{DT} \mathbb{E}[\text{TERM-A} + \text{TERM-B} + \text{TERM-C}]$. First, divide (35) by $DT$ and substitute $\gamma = \beta_1 D / \sqrt{1 - \beta_1}$:

$$\frac{1}{DT} \mathbb{E}[\text{TERM-A}] \leq \frac{\sqrt{1 - \beta_1}}{4} \cdot \left( \frac{2(1 - \beta_2)}{\ln 2} \cdot \frac{G^2}{\nu} + \frac{2\sqrt{2/\ln 2}}{\sqrt{2^{-1/\beta_2}}} G \right) + \frac{\nu}{2T\sqrt{1 - \beta_1}}.$$

Using $\beta_2 \geq 1 - \nu/(G + \sigma)$ gives $(1 - \beta_2)\frac{G^2}{\nu} \leq G$, hence

$$\frac{1}{DT} \mathbb{E}[\text{TERM-A}] \leq \frac{\sqrt{1 - \beta_1}}{4} G \left( \frac{2}{\ln 2} + \frac{2\sqrt{2/\ln 2}}{\sqrt{2^{-1/\beta_2}}} \right) + \frac{\nu}{2T\sqrt{1 - \beta_1}}. \tag{38}$$

Similarly, dividing $\mathbb{E}[\text{TERM-B}]$ by $DT$ and substituting $\gamma = \beta_1 D / \sqrt{1 - \beta_1}$ provides

$$\frac{1}{DT} \mathbb{E}[\text{TERM-B}] \leq \frac{G}{2T\sqrt{1 - \beta_1}} + \frac{G\sqrt{1 - \beta_1}}{2} + \frac{\nu\sqrt{1 - \beta_1}}{2}. \tag{39}$$

Dividing $\mathbb{E}[\text{TERM-C}]$ by $DT$ and substituting $\gamma = \beta_1 D/\sqrt{1-\beta_1}$ provides

$$\frac{1}{DT}\mathbb{E}[\text{TERM-C}] \leq \frac{G\sqrt{1-\beta_1}}{2}. \tag{40}$$

In addition, since we assume $\epsilon \leq G+\sigma$ (otherwise $\mathbb{E}\|\nabla F(\bar{\mathbf{x}})\|_c \leq G+\sigma \leq \epsilon$ is immediate), so $\beta_1 \geq 1-(1/16)^2 = 255/256$ and hence $\beta_2 \geq \beta_1^4 \geq (255/256)^4$. Therefore,

$$\frac{1}{\sqrt{2^{-1/\beta_2}}} = 2^{\frac{1}{2\beta_2}} \leq 2^{\frac{1}{2\beta_1^4}} < \frac{3}{2}.$$

Plugging this into (38) and using $\nu \leq G + \sigma$ gives

$$\frac{1}{DT}\mathbb{E}[\text{TERM-A}] \leq \frac{\epsilon}{64}\left(\frac{2}{\ln 2} + 3\sqrt{\frac{2}{\ln 2}}\right) + \frac{\nu}{2T\sqrt{1-\beta_1}} < \frac{\epsilon}{4} + \frac{\nu}{2T\sqrt{1-\beta_1}}.$$

Moreover, by (39)–(40) and $\nu \leq G + \sigma$,

$$\frac{1}{DT}\mathbb{E}[\text{TERM-B} + \text{TERM-C}] \leq \frac{G+\sigma}{2T\sqrt{1-\beta_1}} + \frac{\epsilon}{16} + \frac{\epsilon}{32}.$$

Combining the above two displays gives

$$\frac{1}{DT}\mathbb{E}[\text{TERM-A} + \text{TERM-B} + \text{TERM-C}] \leq \frac{11}{32}\epsilon + \frac{G+\sigma+\nu}{2T\sqrt{1-\beta_1}}. \tag{41}$$

Finally, choose $T$ to satisfy

$$T \geq \max\left\{(1-\beta_1)^{-1} \cdot \max\left\{\frac{16F^*\sqrt{48c}}{\epsilon^{3/2}}, \frac{16(G+\sigma)}{\epsilon}\right\}, \frac{\ln 2}{1-\beta_2}\right\}.$$

Then

$$\frac{F^*}{DT} \leq \frac{\epsilon}{16}, \qquad \frac{2G+\sigma}{(1-\beta_1)T} \leq \frac{(2G+\sigma)\epsilon}{16(G+\sigma)} \leq \frac{3\epsilon}{16},$$

and also, since $\sqrt{1-\beta_1} = \epsilon/(16(G+\sigma))$ and $\nu \leq G+\sigma$,

$$\frac{G+\sigma+\nu}{2T\sqrt{1-\beta_1}} \leq \frac{2(G+\sigma)}{2T} \cdot \frac{16(G+\sigma)}{\epsilon} = \frac{16(G+\sigma)^2}{T\epsilon} \leq \frac{\epsilon}{32}.$$

Putting everything back into (37) and using (41), we obtain $\mathbb{E}\|\nabla F(\bar{\mathbf{x}})\|_c \leq \epsilon$, finishing the proof. $\qquad\square$

### D.7. Proof of Theorem 5

*Proof.* Throughout the proof, we use $\beta_1$ for the first-moment discount factor in Adam and the discount factor in Algorithm 1, and $\beta_2$ for the second-moment terms.

**O2NC Conversion.** Apply Lemma 8 with $\beta = \beta_1$. With $\ell_t(\Delta) = \langle \mathbf{g}_t, \Delta \rangle$ and D-$\text{REG}_T(\mathbf{u}_{1:T}) = \sum_{t=1}^{T}\langle \mathbf{g}_t, \Delta_t - \mathbf{u}_t \rangle$, we obtain

$$\mathbb{E}\|\nabla F(\bar{\mathbf{x}})\|_c \leq \frac{F^*}{DT} + \frac{2G+\sigma}{(1-\beta_1)T} + \sigma\sqrt{1-\beta_1} + \frac{12cD^2}{(1-\beta_1)^2}$$
$$+ \frac{1}{DT}\mathbb{E}\left[\text{D-REG}_T(\mathbf{u}_{1:T}) + \beta_1\sum_{t=1}^{T-1}\sum_{s=1}^{t}\beta_1^{t-s}\big(\ell_s(\mathbf{u}_t) - \ell_s(\mathbf{u}_{t+1})\big)\right].$$

Recall that $\|\mathbf{u}_t\|_2 = D$ for all $t$ by the definition in Lemma 8.

**Use D2D Reduction.** We apply Lemma 14 to the discounted FTRL run on the clipped domain $\mathcal{D} = \{\Delta : \|\Delta\|_2 \leq D\}$, with decisions denoted by $\Delta_t$ and comparator denoted by $\mathbf{u}_t$. We choose Option II in Lemma 14, i.e.,

$$\Lambda_t = \eta_t \|\beta_1^{-t}\mathbf{g}_t\|_2^2 + \min\left\{\frac{\eta_{t-1}}{2}\|\beta_1^{-t}\mathbf{g}_t\|_2^2, \ \min\{2D, |\eta_{t-1} - \eta_t|\|\widetilde{\mathbf{g}}_{1:t-1}\|_2\} \cdot \|\beta_1^{-t}\mathbf{g}_t\|_2\right\},$$

where $\widetilde{\mathbf{g}}_{1:t} = \sum_{s=1}^{t} \beta_1^{-s}\mathbf{g}_s$. Using the same stepsizes $V_t, \alpha_t, \eta_t$ as in the proof of Theorem 4 in Appendix D.6, the path length term in O2NC conversion cancels $\beta_1 \sum_{t=1}^{T-1} \sum_{s=1}^{t} \beta_1^{t-s}\langle \mathbf{g}_s, \mathbf{u}_{t+1} - \mathbf{u}_t\rangle$ in Lemma 14. We obtain the decomposition

$$\text{D-REG}_T(\mathbf{u}_{1:T}) + \beta_1 \sum_{t=1}^{T-1} \sum_{s=1}^{t} \beta_1^{t-s}\big(\ell_s(\mathbf{u}_t) - \ell_s(\mathbf{u}_{t+1})\big)$$

$$\leq \underbrace{\sum_{t=1}^{T} \beta_1^t \eta_t \|\beta_1^{-t}\mathbf{g}_t\|_2^2 + \frac{\beta_1}{2\gamma(1-\beta_1)\alpha_0}\|\mathbf{u}_1\|_2^2}_{\text{TERM-A}}$$

$$+ \underbrace{\sum_{t=1}^{T} \beta_1^t \min\left\{\frac{\eta_{t-1}}{2}\|\beta_1^{-t}\mathbf{g}_t\|_2^2, \ \min\{D, |\eta_{t-1} - \eta_t|\|\widetilde{\mathbf{g}}_{1:t-1}\|_2\} \cdot \|\beta_1^{-t}\mathbf{g}_t\|_2\right\}}_{\text{TERM-B}}$$

$$+ \underbrace{\sum_{t=1}^{T} \left(\frac{\beta_1^t}{2\eta_{t-1}}\|\Delta_{t+1}\|_2^2 - \frac{\beta_1^t}{2\eta_t}\|\Delta_{t+1}\|_2^2\right)}_{\text{TERM-C}} + \underbrace{\beta_1 \sum_{t=1}^{T-1}\left(\frac{\beta_1^t}{2\eta_t}\|\mathbf{u}_{t+1}\|_2^2 - \frac{\beta_1^t}{2\eta_{t-1}}\|\mathbf{u}_t\|_2^2\right)}_{\text{TERM-D}}. \tag{42}$$

**TERM-A.** We have

$$\sum_{t=1}^{T} \beta_1^t \eta_t \|\beta_1^{-t}\mathbf{g}_t\|_2^2 = \gamma(1-\beta_1)\sum_{t=1}^{T} \alpha_t \|\mathbf{g}_t\|_2^2 = \gamma(1-\beta_1)\sum_{t=1}^{T} \frac{\|\mathbf{g}_t\|_2^2}{\nu + \sqrt{V_t}}.$$

Applying Lemma 10 with $a_t = \|\mathbf{g}_t\|_2^2$, $\beta = \beta_2$, and $\epsilon = \nu$ gives

$$\sum_{t=1}^{T} \alpha_t \|\mathbf{g}_t\|_2^2 \leq \frac{2}{\sqrt{(1-\beta_2)2^{-1/\beta_2}}}\sqrt{\left(\frac{T(1-\beta_2)}{\ln 2} + 1\right)\sum_{t=1}^{T}\|\mathbf{g}_t\|_2^2}.$$

Taking expectation and using Jensen's inequality together with $\mathbb{E}\sum_{t=1}^{T}\|\mathbf{g}_t\|_2^2 \leq TG^2$ gives

$$\mathbb{E}\left[\sum_{t=1}^{T}\beta_1^t \eta_t \|\beta_1^{-t}\mathbf{g}_t\|_2^2\right] \leq \gamma(1-\beta_1) \cdot \frac{2}{\sqrt{(1-\beta_2)2^{-1/\beta_2}}}\sqrt{\frac{T(1-\beta_2)}{\ln 2} + 1}\sqrt{T}G.$$

Assume that $T \geq (1-\beta_2)^{-1}\ln 2$, then $\frac{T(1-\beta_2)}{\ln 2} + 1 \leq \frac{2T(1-\beta_2)}{\ln 2}$, so after substituting $\gamma = \frac{\beta_1 D}{\sqrt{1-\beta_1}}$ and using $\beta_1 \leq 1$, the above inequality gives

$$\frac{1}{DT}\mathbb{E}\left[\sum_{t=1}^{T}\beta_1^t \eta_t \|\beta_1^{-t}\mathbf{g}_t\|_2^2\right] \leq \frac{2^{3/2+1/(2\beta_2)}}{\sqrt{\ln 2}}G\sqrt{1-\beta_1},$$

and by definition:

$$\frac{1}{DT}\mathbb{E}\left[\text{TERM-A}\right] \leq \frac{2^{3/2+1/(2\beta_2)}}{\sqrt{\ln 2}}G\sqrt{1-\beta_1} + \frac{\nu}{2T\sqrt{1-\beta_1}}.$$

**TERM-B.** We bound the deviation part using Lemma 13 and $\|\mathbf{g}_t\|_2 \leq G$:

$$\beta_1^t |\eta_{t-1} - \eta_t| \cdot \|\widetilde{\mathbf{g}}_{1:t-1}\|_2 \cdot \|\beta_1^{-t}\mathbf{g}_t\|_2 \ \leq \ \gamma(1-\beta_1) \cdot \left(1 - \frac{\beta_1 \alpha_t}{\alpha_{t-1}}\right) \cdot \sqrt{\frac{\beta_2}{(\beta_2 - \beta_1^2)(1-\beta_2)}} \cdot G.$$

Summing over $t$ and using the inequality $1 - x \leq \ln(1/x)$ for $x > 0$ with $x = \beta_1 \alpha_t / \alpha_{t-1}$ gives

$$\sum_{t=1}^{T} \left(1 - \frac{\beta_1 \alpha_t}{\alpha_{t-1}}\right) \leq \sum_{t=1}^{T} \ln \frac{\alpha_{t-1}}{\beta_1 \alpha_t} = \ln \frac{\alpha_0}{\alpha_T} - T \ln \beta_1.$$

Therefore,

$$\text{TERM-B} \leq \gamma(1 - \beta_1)G\left(\ln \frac{\alpha_0}{\alpha_T} - T \ln \beta_1\right) \cdot \sqrt{\frac{\beta_2}{(\beta_2 - \beta_1^2)(1 - \beta_2)}}. \tag{43}$$

Next we upper bound the $\beta_2$-dependent factor by the margin condition. Let $\beta_2^\star = \frac{1+\beta_1^2}{2}$ and recall $\beta_2 \in [\beta_1^2 + m, 1 - m]$ with $m = \frac{1-\rho}{2}(1 - \beta_1^2)$. Then $|\beta_2 - \beta_2^\star| \leq \frac{\rho}{2}(1 - \beta_1^2)$ and hence

$$(\beta_2 - \beta_1^2)(1 - \beta_2) = \frac{(1 - \beta_1^2)^2}{4} - (\beta_2 - \beta_2^\star)^2 \geq \frac{(1 - \beta_1^2)^2}{4}(1 - \rho^2).$$

Since $\beta_2 \leq 1$, we obtain

$$\sqrt{\frac{\beta_2}{(\beta_2 - \beta_1^2)(1 - \beta_2)}} \leq \frac{2}{1 - \beta_1^2} \cdot \frac{1}{\sqrt{1 - \rho^2}}. \tag{44}$$

Moreover, $\sqrt{V_t} \leq G$ implies $\alpha_T \geq \frac{1}{\nu + G}$ and $\alpha_0 = \frac{1}{\nu}$, so

$$\ln \frac{\alpha_0}{\alpha_T} \leq \ln \left(1 + \frac{G}{\nu}\right). \tag{45}$$

Also $-\ln \beta_1 \leq \frac{1-\beta_1}{\beta_1}$. Plugging (44)–(45) into (43) and using $\gamma = \frac{\beta_1 D}{\sqrt{1-\beta_1}}$ yields

$$\frac{1}{DT}\mathbb{E}[\text{TERM-B}] \leq \frac{2G}{T\sqrt{1-\beta_1}} \cdot \frac{1}{\sqrt{1-\rho^2}} \ln\left(1 + \frac{G}{\nu}\right) + \frac{2G}{\sqrt{1-\rho^2}}\sqrt{1-\beta_1}.$$

**TERM-C.** Since $\beta_2 > \beta_1^2$, Lemma 11 gives $\mathbb{E}[\text{TERM-C}] \leq 0$.

**TERM-D.** Since $\|\mathbf{u}_t\|_2 = D$ for all $t$,

$$\text{TERM-D} = \frac{\beta_1 D^2}{2} \sum_{t=1}^{T-1} \beta_1^t \left(\frac{1}{\eta_t} - \frac{1}{\eta_{t-1}}\right).$$

By Lemma 12 with $\epsilon = \nu$, taking expectation and using $\mathbb{E}[\sqrt{V_t}] \leq G$, $\mathbb{E}\sum_{t=1}^{T-2}\sqrt{V_t} \leq TG$, and $\gamma = \frac{\beta_1 D}{\sqrt{1-\beta_1}}$ gives

$$\frac{1}{DT}\mathbb{E}[\text{TERM-D}] \leq \frac{G}{2T\sqrt{1-\beta_1}} + \frac{G\sqrt{1-\beta_1}}{2} + \frac{\nu\sqrt{1-\beta_1}}{2}.$$

**Combine and Tune Parameters.** Combine the above bounds, we have:

$$
\begin{aligned}
\mathbb{E}\|\nabla F(\bar{\mathbf{x}})\|_c \leq\ & \frac{F^*}{DT} + \frac{2G + \sigma}{(1 - \beta_1)T} + \sigma\sqrt{1 - \beta_1} + \frac{12cD^2}{(1 - \beta_1)^2} \\
& + \frac{2^{3/2 + 1/(2\beta_2)}}{\sqrt{\ln 2}}G\sqrt{1 - \beta_1} + \frac{G + \nu}{2}\sqrt{1 - \beta_1} + \frac{2G}{\sqrt{1 - \rho^2}}\sqrt{1 - \beta_1} \\
& + \frac{G + \nu}{2T\sqrt{1 - \beta_1}} + \frac{2G}{T\sqrt{1 - \beta_1}} \cdot \frac{1}{\sqrt{1 - \rho^2}} \ln\left(1 + \frac{G}{\nu}\right).
\end{aligned}
\tag{46}
$$

Now set

$$D = \frac{(1-\beta_1)\sqrt{\epsilon}}{\sqrt{48c}}, \qquad \gamma = \frac{\beta_1 D}{\sqrt{1-\beta_1}}, \qquad T \geq \frac{\ln 2}{1-\beta_2}.$$

Then $\frac{12cD^2}{(1-\beta_1)^2} = 12c \cdot \frac{(1-\beta_1)^2 \epsilon}{48c} \cdot \frac{1}{(1-\beta_1)^2} = \frac{\epsilon}{4}$. Next, choose $\beta_1$ such that

$$\sqrt{1-\beta_1} \leq \frac{\epsilon\sqrt{1-\rho^2}}{64(G+\sigma)}.$$

Then

$$\sigma\sqrt{1-\beta_1} \leq (G+\sigma)\sqrt{1-\beta_1} \leq \frac{\epsilon}{64}, \qquad \frac{2(G+\sigma)}{\sqrt{1-\rho^2}}\sqrt{1-\beta_1} \leq \frac{\epsilon}{32}, \qquad (G+\sigma)\sqrt{1-\beta_1} \leq \frac{\epsilon}{64}.$$

Moreover, since $\beta_2 > \beta_1^2$ and $\beta_1 \geq 15/16$ (which follows whenever $\epsilon \leq 16(G+\sigma)/\sqrt{1-\rho^2}$), we have $2^{1/(2\beta_2)} \leq 3/2$ and hence $\frac{2^{3/2+1/(2\beta_2)}}{\sqrt{\ln 2}} < 6$, so

$$\frac{2^{3/2+1/(2\beta_2)}}{\sqrt{\ln 2}}(G+\sigma)\sqrt{1-\beta_1} \leq \frac{6\epsilon}{64} = \frac{3\epsilon}{32}.$$

Finally, choose $T$ such that

$$T \geq \frac{1}{1-\beta_1} \cdot \max\left\{\frac{32F^*\sqrt{c}}{\epsilon^{3/2}}, \frac{16(G+\sigma)}{\epsilon}\right\}, \qquad T \geq \frac{32G}{\epsilon\sqrt{1-\beta_1}\sqrt{1-\rho^2}} \cdot \ln\left(1+\frac{G}{\nu}\right).$$

Then we have

$$\frac{F^*}{DT} \leq \frac{\sqrt{48}}{32}\epsilon < \frac{\epsilon}{4}, \qquad \frac{2G+\sigma}{(1-\beta_1)T} \leq \frac{\epsilon}{8}, \qquad \frac{2(G+\sigma)}{T\sqrt{1-\beta_1}} \cdot \frac{1}{\sqrt{1-\rho^2}} \ln\left(1+\frac{G}{\nu}\right) \leq \frac{\epsilon}{16}.$$

Also using $\nu \leq G+\sigma$ and $T \geq \frac{16(G+\sigma)}{\epsilon(1-\beta_1)}$,

$$\frac{G+\nu}{2T\sqrt{1-\beta_1}} \leq \frac{G+\sigma}{T\sqrt{1-\beta_1}} \leq \frac{\epsilon\sqrt{1-\beta_1}}{16} \leq \frac{\epsilon}{16}.$$

Collecting all bounds in (46), gives $\mathbb{E}\|\nabla F(\bar{\mathbf{x}})\|_c \leq \epsilon$ and finishes the proof. $\qquad\square$

### D.8. Proof of Theorem 6

*Proof of Theorem 6.* Throughout the proof, we use $\beta_1$ for the first-moment discount factor in Adam and the discount factor in Algorithm 1, and $\beta_2$ for the second-moment terms.

**O2NC Conversion.** Apply Lemma 7 with $\beta = \beta_1$ and the comparator sequence $\{\mathbf{u}_t\}_{t=1}^T$ therein. With $\ell_t(\Delta) = \langle \mathbf{g}_t, \Delta \rangle + \frac{\mu}{2}\|\Delta\|_2^2$ and D-REG$_T(\mathbf{u}_{1:T}) = \sum_{t=1}^T \left(\ell_t(\Delta_t) - \ell_t(\mathbf{u}_t)\right)$, we obtain

$$\mathbb{E}\|\nabla F(\bar{\mathbf{x}})\|_c \leq \frac{F^*}{DT} + \frac{2G+\sigma}{(1-\beta_1)T} + \sigma\sqrt{1-\beta_1} + \frac{12cD^2}{(1-\beta_1)^2}$$

$$+ \frac{1}{DT}\mathbb{E}\left[\text{D-REG}_T(\mathbf{u}_{1:T}) + \beta_1\sum_{t=1}^{T-1}\sum_{s=1}^t \beta_1^{t-s}\left(\ell_s(\mathbf{u}_t) - \ell_s(\mathbf{u}_{t+1})\right)\right]. \tag{47}$$

**Use D2D Reduction.** Define

$$V_t = (1-\beta_2)\sum_{s=1}^t \beta_2^{t-s}\|\mathbf{g}_s\|_2^2, \qquad \eta_t = \gamma(1-\beta_1)\frac{\beta_1^t}{\nu+\sqrt{V_t}}.$$

We apply Lemma 15 (Option I) with stepsizes $\{\eta_t\}$, the iterate sequence $\{\Delta_t\}_{t=1}^T$, and comparator sequence $\{\mathbf{u}_t\}$ in Lemma 7. The effective stepsizes are $\eta_t' = \gamma(1-\beta_1)\beta_1^t\alpha_t'$ where $\alpha_t' = \frac{1}{\nu+\gamma\mu(1-\beta_1^t)+\sqrt{V_t}}$. Write $\psi_{t+1}(\Delta) = \frac{1}{2\eta_t}\|\Delta\|_2^2$. The path length term in Lemma 15 cancels the negative path length term in (47), and we obtain the decomposition

$$\text{D-Reg}_T(\mathbf{u}_{1:T}) + \beta_1\sum_{t=1}^{T-1}\sum_{s=1}^t \beta_1^{t-s}\big(\ell_s(\mathbf{u}_t) - \ell_s(\mathbf{u}_{t+1})\big)$$

$$\leq \underbrace{\frac{1-\beta_1}{2\beta_1}\gamma\sum_{t=1}^T \alpha_{t-1}'\|\mathbf{g}_t\|_2^2 + \frac{\beta_1}{2\gamma(1-\beta_1)\alpha_0'}\|\mathbf{u}_1\|_2^2}_{\text{Term-A}} + \underbrace{\beta_1\sum_{t=1}^{T-1}\Big(\frac{\beta_1^t}{2\eta_t}\|\mathbf{u}_{t+1}\|_2^2 - \frac{\beta_1^t}{2\eta_{t-1}}\|\mathbf{u}_t\|_2^2\Big)}_{\text{Term-B}}$$

$$+ \underbrace{\sum_{t=1}^T\Big(\frac{\beta_1^t}{2\eta_{t-1}}\|\Delta_{t+1}\|_2^2 - \frac{\beta_1^t}{2\eta_t}\|\Delta_{t+1}\|_2^2\Big)}_{\text{Term-C}}. \tag{48}$$

**Term-A.** Since $\gamma\mu(1-\beta_1^t) \geq 0$, we have $\alpha_{t-1}' \leq \alpha_{t-1}$. By the same argument as Term-A in the proof of Theorem 4 (Lemma 9 with $a_t = \|\mathbf{g}_t\|_2^2$, $\epsilon = \nu$, $\beta = \beta_2$, $A = G^2$), using $\alpha_0' = \alpha_0 = \frac{1}{\nu}$, $\|\mathbf{u}_1\|_2 = D$, and $K = \frac{T(1-\beta_2)}{\ln 2} + 1$, we obtain

$$\mathbb{E}[\text{Term-A}] \leq \frac{1-\beta_1}{2\beta_1}\gamma\left(K\frac{(G+\sigma)^2}{\nu} + \frac{2}{\sqrt{(1-\beta_2)2^{-1/\beta_2}}}\sqrt{KT}(G+\sigma)\right) + \frac{\beta_1\nu}{2\gamma(1-\beta_1)}D^2.$$

**Term-B.** Since $\|\mathbf{u}_t\|_2 = D$ for all $t$ and $\psi_{t+1}$ uses the base stepsize $\eta_t$,

$$\text{Term-B} = \frac{\beta_1 D^2}{2}\sum_{t=1}^{T-1}\beta_1^t\Big(\frac{1}{\eta_t} - \frac{1}{\eta_{t-1}}\Big).$$

Apply Lemma 12 with $\epsilon = \nu$ to obtain

$$\sum_{t=1}^{T-1}\beta_1^t\Big(\frac{1}{\eta_t} - \frac{1}{\eta_{t-1}}\Big) \leq \frac{\sqrt{V_{T-1}}}{\gamma(1-\beta_1)} + \frac{1}{\gamma}\sum_{t=1}^{T-2}\sqrt{V_t} + \frac{(T-1)\nu}{\gamma}.$$

Using $\|\mathbf{g}_t\|_2 \leq G$ implies $V_t \leq G^2$ and hence $\sqrt{V_t} \leq G$ for all $t$. Taking expectation in the above inequality gives

$$\mathbb{E}\Big[\sqrt{V_{T-1}} + \sum_{t=1}^{T-2}\sqrt{V_t}\Big] \leq TG.$$

Therefore $\mathbb{E}[\text{Term-B}] \leq \frac{\beta_1 D^2}{2}\left(\frac{TG}{\gamma(1-\beta_1)} + \frac{(T-1)\nu}{\gamma}\right)$.

**Term-C.** Since $\beta_2 \geq \beta_1^2$, Lemma 11 gives $\mathbb{E}[\text{Term-C}] \leq 0$.

**Combine and Tune Parameters.** Combining (47) with (48) and the above arguments gives

$$\mathbb{E}\|\nabla F(\bar{\mathbf{x}})\|_c \leq \frac{F^*}{DT} + \frac{2G+\sigma}{(1-\beta_1)T} + \sigma\sqrt{1-\beta_1} + \frac{12cD^2}{(1-\beta_1)^2} + \frac{1}{DT}\mathbb{E}[\text{Term-A} + \text{Term-B} + \text{Term-C}].$$

We now tune the parameters. Set $\mu = \frac{24cD}{(1-\beta_1)^2}$ as required by Lemma 7, and set

$$\beta_1 = 1 - \Big(\frac{\epsilon}{16(G+\sigma)}\Big)^2, \qquad D = \frac{(1-\beta_1)\sqrt{\epsilon}}{\sqrt{96c}}, \qquad \gamma = \frac{\beta_1 D}{\sqrt{1-\beta_1}}, \qquad \beta_2 \geq \max\Big\{1 - \frac{\nu}{G+\sigma}, \beta_1^2\Big\}.$$

With the above tuning, we have

$$\frac{12cD^2}{(1-\beta_1)^2} = \frac{\epsilon}{8}, \qquad \sigma\sqrt{1-\beta_1} \leq (G+\sigma)\sqrt{1-\beta_1} = \frac{\epsilon}{16}.$$

Substituting $\gamma = \beta_1 D/\sqrt{1-\beta_1}$, and noting $\alpha_0' = \alpha_0 = 1/\nu$, and assuming $T \geq \frac{\ln 2}{1-\beta_2}$, we obtain

$$\frac{1}{DT}\mathbb{E}[\text{TERM-A}] \leq G\sqrt{1-\beta_1}\left(\frac{1}{\ln 2} + \sqrt{\frac{2}{\ln 2}} \cdot \frac{1}{\sqrt{2^{-1/\beta_2}}}\right) + \frac{\nu}{2T\sqrt{1-\beta_1}}.$$

Since $\epsilon \leq G + \sigma$ implies $\beta_1 \geq 255/256$ and $\beta_2 \geq \beta_1^2 \geq (255/256)^2$, we have $\frac{1}{\sqrt{2^{-1/\beta_2}}} = 2^{\frac{1}{2\beta_2}} < \frac{3}{2}$. Therefore,

$$\frac{1}{DT}\mathbb{E}[\text{TERM-A}] \leq 4G\sqrt{1-\beta_1} + \frac{\nu}{2T\sqrt{1-\beta_1}} = \frac{\epsilon}{4} + \frac{\nu}{2T\sqrt{1-\beta_1}}.$$

Similarly, substituting $\gamma = \beta_1 D/\sqrt{1-\beta_1}$ gives

$$\frac{1}{DT}\mathbb{E}[\text{TERM-B}] \leq \frac{G}{2T\sqrt{1-\beta_1}} + \frac{G\sqrt{1-\beta_1}}{2} + \frac{\nu\sqrt{1-\beta_1}}{2} \leq \frac{G}{2T\sqrt{1-\beta_1}} + \frac{\epsilon}{16}.$$

Finally, choose $T$ to satisfy

$$T \geq \max\left\{\frac{1}{1-\beta_1}\max\left\{\frac{16F^*\sqrt{96c}}{\epsilon^{3/2}}, \frac{48(G+\sigma)}{\epsilon}\right\}, \frac{\ln 2}{1-\beta_2}\right\}. \tag{49}$$

Then $\frac{F^*}{DT} \leq \epsilon/16$ and $\frac{2G+\sigma}{(1-\beta_1)T} \leq \epsilon/16$. Moreover, by a standard assumption $\nu \leq G + \sigma$ and $\sqrt{1-\beta_1} = \epsilon/(16(G+\sigma))$, the condition (49) implies $\frac{G+\nu}{2T\sqrt{1-\beta_1}} \leq \frac{\epsilon}{32}$. Combining the above bounds and $\mathbb{E}[\text{TERM-C}] \leq 0$ gives $\mathbb{E}\|\nabla F(\bar{\mathbf{x}})\|_c \leq \epsilon$. $\square$

### D.9. Proof of Theorem 7

*Proof.* Throughout the proof, we use $\beta_1$ for the first-moment discount factor in Adam (and the discounting factor in Algorithm 1), and $\beta_2$ for the second-moment discount factor.

**O2NC Conversion.** Apply Lemma 7 with $\beta = \beta_1$. With $\ell_t(\Delta) = \langle \mathbf{g}_t, \Delta \rangle + \frac{\mu}{2}\|\Delta\|_2^2$ and $\text{D-REG}_T(\mathbf{u}_{1:T}) = \sum_{t=1}^{T}(\ell_t(\Delta_t) - \ell_t(\mathbf{u}_t))$, we obtain

$$\mathbb{E}\|\nabla F(\bar{\mathbf{x}})\|_c \leq \frac{F^*}{DT} + \frac{2G+\sigma}{(1-\beta_1)T} + \sigma\sqrt{1-\beta_1} + \frac{12cD^2}{(1-\beta_1)^2}$$
$$+ \frac{1}{DT}\mathbb{E}\left[\text{D-REG}_T(\mathbf{u}_{1:T}) + \beta_1\sum_{t=1}^{T-1}\sum_{s=1}^{t}\beta_1^{t-s}\big(\ell_s(\mathbf{u}_t) - \ell_s(\mathbf{u}_{t+1})\big)\right]. \tag{50}$$

Recall that $\|\mathbf{u}_t\|_2 = D$ for all $t$ by construction in Lemma 7.

**Use D2D Reduction.** Define

$$V_t = (1-\beta_2)\sum_{s=1}^{t}\beta_2^{t-s}\|\mathbf{g}_s\|_2^2, \qquad \eta_t = \gamma(1-\beta_1)\frac{\beta_1^t}{\nu + \sqrt{V_t}}.$$

We apply Lemma 15 (Option II) with stepsizes $\eta_t$, the iterate sequence $\{\Delta_t\}_{t=1}^{T}$, and comparator sequence $\{\mathbf{u}_t\}_{t=1}^{T}$ in Lemma 7. The effective stepsizes are $\eta_t' = \gamma(1-\beta_1)\beta_1^t\alpha_t'$ where $\alpha_t' = \frac{1}{\nu+\gamma\mu(1-\beta_1^t)+\sqrt{V_t}}$. Write $\psi_{t+1}(\Delta) = \frac{1}{2\eta_t}\|\Delta\|_2^2$. The path length term in Lemma 15 cancels the negative path length term in (50), and we obtain the decomposition

$$\text{D-REG}_T(\mathbf{u}_{1:T}) + \beta_1\sum_{t=1}^{T-1}\sum_{s=1}^{t}\beta_1^{t-s}\big(\ell_s(\mathbf{u}_t) - \ell_s(\mathbf{u}_{t+1})\big)$$

$$\leq \underbrace{\sum_{t=1}^{T}\beta_1^t\eta_t'\|\beta_1^{-t}\mathbf{g}_t\|_2^2 + \frac{\beta_1}{2\eta_0}\|\mathbf{u}_1\|_2^2}_{\text{TERM-A}} + \underbrace{\sum_{t=1}^{T}\beta_1^t\min\left\{\frac{\eta_{t-1}'}{2}\|\beta_1^{-t}\mathbf{g}_t\|_2^2, |\eta_{t-1}' - \eta_t'|\|\widetilde{\mathbf{g}}_{1:t-1}\|_2 \cdot \|\beta_1^{-t}\mathbf{g}_t\|_2\right\}}_{\text{TERM-B}}$$

$$+ \underbrace{\sum_{t=1}^{T} \beta_1^t \big(\psi_t(\Delta_{t+1}) - \psi_{t+1}(\Delta_{t+1})\big)}_{\text{TERM-C}} + \underbrace{\beta_1 \sum_{t=1}^{T-1} \beta_1^t \big(\psi_{t+1}(\mathbf{u}_{t+1}) - \psi_t(\mathbf{u}_t)\big)}_{\text{TERM-D}}, \tag{51}$$

where $\widetilde{\mathbf{g}}_{1:t} = \sum_{s=1}^{t} \beta_1^{-s} \mathbf{g}_s$.

**TERM-A.**  Since $\eta_t' = \gamma(1 - \beta_1)\beta_1^t \alpha_t'$, we have

$$\beta_1^t \eta_t' \|\beta_1^{-t} \mathbf{g}_t\|_2^2 = \gamma(1 - \beta_1)\alpha_t' \|\mathbf{g}_t\|_2^2.$$

Since $\alpha_t' \le \alpha_t = \frac{1}{\nu + \sqrt{V_t}}$, we have

$$\mathbb{E} \sum_{t=1}^{T} \alpha_t' \|\mathbf{g}_t\|_2^2 \le \mathbb{E} \sum_{t=1}^{T} \frac{\|\mathbf{g}_t\|_2^2}{\nu + \sqrt{V_t}} \le \frac{2^{3/2 + 1/(2\beta_2)}}{\sqrt{\ln 2}} TG.$$

Since $V_0 = 0$ and $\beta_1^0 = 1$, we have $\alpha_0' = \alpha_0 = 1/\nu$ and thus $\eta_0' = \eta_0 = \gamma(1 - \beta_1)/\nu$, so

$$\frac{1}{DT} \cdot \frac{\beta_1}{2\eta_0'} \|\mathbf{u}_1\|_2^2 = \frac{\nu}{2T\sqrt{1 - \beta_1}}. \tag{52}$$

**TERM-B.**  We upper bound the minimum by the deviation term:

$$\beta_1^t |\eta_{t-1}' - \eta_t'| \|\widetilde{\mathbf{g}}_{1:t-1}\|_2 \cdot \|\beta_1^{-t} \mathbf{g}_t\|_2 = |\eta_{t-1} - \eta_t| \Big\| \sum_{s=1}^{t-1} \beta_1^{t-1-s} \mathbf{g}_s \Big\|_2 \cdot \|\beta_1^{1-t} \mathbf{g}_t\|_2.$$

Apply Lemma 13 and use $\|\mathbf{g}_t\|_2 \le G$:

$$\beta_1^t |\eta_{t-1}' - \eta_t'| \|\widetilde{\mathbf{g}}_{1:t-1}\|_2 \cdot \|\beta_1^{-t} \mathbf{g}_t\|_2 \le \gamma(1 - \beta_1) \cdot \Big(1 - \frac{\beta_1 \alpha_t'}{\alpha_{t-1}'}\Big) \cdot \sqrt{\frac{\beta_2}{(\beta_2 - \beta_1^2)(1 - \beta_2)}} \cdot G.$$

Summing the above inequality over $t$ and using the fact $1 - x \le \ln(1/x)$ with $x = \beta_1 \alpha_t'/\alpha_{t-1}'$ gives $\sum_{t=1}^{T} (1 - \beta_1 \alpha_t'/\alpha_{t-1}') \le \sum_{t=1}^{T} \ln(\alpha_{t-1}'/(\beta_1 \alpha_t')) = \ln(\alpha_0'/(\beta_1^T \alpha_T'))$, which implies

$$\text{TERM-B} \le \gamma(1 - \beta_1)G\Big(\ln \frac{\alpha_0'}{\alpha_T'} - T \ln \beta_1\Big)\sqrt{\frac{\beta_2}{(\beta_2 - \beta_1^2)(1 - \beta_2)}}. \tag{53}$$

Next we bound the $\beta_2$-dependent factor using the margin condition. Let $\beta_2^\star = \frac{1 + \beta_1^2}{2}$. Since $\beta_2 \in [\beta_1^2 + m, \, 1 - m]$ with $m = \frac{1 - \rho}{2}(1 - \beta_1^2)$, we have $|\beta_2 - \beta_2^\star| \le \frac{\rho}{2}(1 - \beta_1^2)$ and hence

$$(\beta_2 - \beta_1^2)(1 - \beta_2) = \frac{(1 - \beta_1^2)^2}{4} - (\beta_2 - \beta_2^\star)^2 \ge \frac{(1 - \beta_1^2)^2}{4}(1 - \rho^2).$$

Therefore

$$\sqrt{\frac{\beta_2}{(\beta_2 - \beta_1^2)(1 - \beta_2)}} \le \frac{2}{1 - \beta_1^2} \cdot \frac{1}{\sqrt{1 - \rho^2}}. \tag{54}$$

Finally, since $\sqrt{V_t} \le G$ and $1 - \beta_1^t \le 1$, we have $\alpha_T' \ge \frac{1}{\nu + \gamma\mu + G}$ and $\alpha_0' = \frac{1}{\nu}$, so

$$\ln \frac{\alpha_0'}{\alpha_T'} \le \ln\Big(1 + \frac{\gamma\mu + G}{\nu}\Big). \tag{55}$$

Also $-\ln \beta_1 \le \frac{1 - \beta_1}{\beta_1}$. Plugging (54)–(55) into (53) and using $\gamma = \frac{\beta_1 D}{\sqrt{1 - \beta_1}}$ gives

$$\frac{1}{DT}\mathbb{E}[\text{TERM-B}] \le \frac{2G}{T\sqrt{1 - \beta_1}} \cdot \frac{1}{\sqrt{1 - \rho^2}} \ln\Big(1 + \frac{\gamma\mu + G}{\nu}\Big) + \frac{2G}{\sqrt{1 - \rho^2}}\sqrt{1 - \beta_1}. \tag{56}$$

**TERM-C.** Since $\psi_{t+1}(\Delta) = \frac{1}{2\eta_t}\|\Delta\|_2^2$ with $\eta_t = \gamma(1-\beta_1)\beta_1^t\alpha_t$, the sequence $\epsilon_t = \nu$ is constant. By Lemma 11,

$$\left[\frac{\beta_1}{\alpha_{t-1}} - \frac{1}{\alpha_t}\right]_+ \le [\beta_1 - \sqrt{\beta_2}]_+ \sqrt{V_{t-1}}.$$

Since $\beta_2 > \beta_1^2$, we have $\sqrt{\beta_2} > \beta_1$ and hence $[\beta_1 - \sqrt{\beta_2}]_+ = 0$. Therefore TERM-C $\le 0$.

**TERM-D.** Since $\|\mathbf{u}_t\|_2 = D$ for all $t$ and $\psi_{t+1}(\Delta) = \frac{1}{2\eta_t}\|\Delta\|_2^2$,

$$\text{TERM-D} = \frac{\beta_1 D^2}{2}\sum_{t=1}^{T-1}\beta_1^t\left(\frac{1}{\eta_t} - \frac{1}{\eta_{t-1}}\right).$$

Applying Lemma 12 with $\epsilon = \nu$ and $\sqrt{V_t} \le G$ gives

$$\frac{1}{DT}\mathbb{E}[\text{TERM-D}] \le \frac{G}{2T\sqrt{1-\beta_1}} + \frac{G\sqrt{1-\beta_1}}{2} + \frac{\nu\sqrt{1-\beta_1}}{2}.$$

**Combine and Tune Parameters.** Collecting the above inequalities, we obtain

$$\mathbb{E}\|\nabla F(\bar{\mathbf{x}})\|_c \le \frac{F^*}{DT} + \frac{2G+\sigma}{(1-\beta_1)T} + \frac{12cD^2}{(1-\beta_1)^2}$$
$$+ \frac{2^{3/2+1/(2\beta_2)}}{\sqrt{\ln 2}}G\sqrt{1-\beta_1} + \sigma\sqrt{1-\beta_1} + \frac{2G}{\sqrt{1-\rho^2}}\sqrt{1-\beta_1} + \frac{G+\nu}{2}\sqrt{1-\beta_1}$$
$$+ \frac{G+\nu}{2T\sqrt{1-\beta_1}} + \frac{2G}{T\sqrt{1-\beta_1}\sqrt{1-\rho^2}}\ln\left(1+\frac{\gamma\mu+G}{\nu}\right). \tag{57}$$

Now set $D = \frac{(1-\beta_1)\sqrt{\epsilon}}{\sqrt{96c}}$ and $\mu = \frac{24cD}{(1-\beta_1)^2}$. Then

$$\frac{12cD^2}{(1-\beta_1)^2} = \frac{\epsilon}{8}.$$

Next, by the choice of $\beta_1$, $\sqrt{1-\beta_1} \le \frac{\epsilon\sqrt{1-\rho^2}}{64(G+\sigma)}$, we have

$$\frac{2G}{\sqrt{1-\rho^2}}\sqrt{1-\beta_1} \le \frac{\epsilon}{32}, \quad \sigma\sqrt{1-\beta_1} \le (G+\sigma)\sqrt{1-\beta_1} \le \frac{\epsilon}{64}, \quad \frac{G+\nu}{2}\sqrt{1-\beta_1} \le (G+\sigma)\sqrt{1-\beta_1} \le \frac{\epsilon}{64}.$$

Moreover, under the mild condition $\epsilon \le \frac{16(G+\sigma)}{\sqrt{1-\rho^2}}$, the above choice implies $\beta_1 \ge 15/16$. Since $\beta_2 > \beta_1^2$, we get $2^{\frac{1}{2\beta_2}} < \frac{3}{2}$ and hence $\frac{2^{3/2+1/(2\beta_2)}}{\sqrt{\ln 2}} < 6$. Therefore,

$$\frac{2^{3/2+1/(2\beta_2)}}{\sqrt{\ln 2}}(G+\sigma)\sqrt{1-\beta_1} \le 6(G+\sigma)\sqrt{1-\beta_1} \le \frac{3\epsilon}{32}.$$

Finally, choose $T$ as in the theorem statement so that

$$\frac{F^*}{DT} \le \frac{\epsilon}{16}, \qquad \frac{2G+\sigma}{(1-\beta_1)T} \le \frac{\epsilon}{16}, \qquad \frac{2G}{T\sqrt{1-\beta_1}\sqrt{1-\rho^2}}\ln\left(1+\frac{\gamma\mu+G}{\nu}\right) \le \frac{\epsilon}{16},$$

and also $\frac{G+\nu}{2T\sqrt{1-\beta_1}} \le \frac{\epsilon}{16}$. Collecting all contributions in (57) yields $\mathbb{E}\|\nabla F(\bar{\mathbf{x}})\|_c \le \epsilon$, concluding the proof.

$\square$

# E. Supporting Lemmas

This part collects useful technical lemmas used in our analysis. For the sake of being self-contained, we state several key FTRL lemmas mainly from Orabona (2019).

### E.1. Technical Lemmas

**Lemma 16.** *Let $\beta \in (0,1]$ and $\{a_t\}_{t=1}^{T} \subseteq \mathbb{R}$. Define $V_t = (1-\beta)\sum_{s=1}^{t}\beta^{t-s}a_s, t \in [T]$. Then*

$$\sum_{t=1}^{T} V_t \;=\; \sum_{t=1}^{T}\big(1 - \beta^{T+1-t}\big)a_t.$$

*Proof.* By exchanging the order of summation,

$$\sum_{t=1}^{T} V_t = (1-\beta)\sum_{t=1}^{T}\sum_{s=1}^{t}\beta^{t-s}a_s = (1-\beta)\sum_{s=1}^{T}a_s\sum_{t=s}^{T}\beta^{t-s} = \sum_{s=1}^{T}a_s\big(1-\beta^{T+1-s}\big).$$

$\square$

**Lemma 17** (Lemma 5 in Jézéquel et al. (2020))**.** *Let $C > 0$ and $f: x \in \mathbb{R} \mapsto \ln(1 + \exp(-x))$. Then for all $a \in [-C, C]$ and $b \in \mathbb{R}$,*

$$f(a) \geq f(b) + f'(b)(a - b) + \frac{e^b}{2(1+C)}f'(b)^2(a-b)^2.$$

**Lemma 18** (Lemma 3.2 in Jacobsen & Cutkosky (2024))**.** *For any non-negative functions $f_0, \ldots, f_T$, $0 < \beta \leq \gamma < 1$, define $F_t^\gamma(\mathbf{u}) = \sum_{s=0}^{t}\gamma^{t-s}f_s(\mathbf{u})$, and*

$$P_T^\gamma = \sum_{t=1}^{T-1}\sum_{s=0}^{t}p_{t,s}^\gamma\left[f_s(\mathbf{u}_{t+1}) - f_s(\mathbf{u}_t)\right]_+, \;\text{ where } p_{t,s}^\gamma = \frac{\gamma^{t-s}}{\sum_{\tau=0}^{t}\gamma^{t-\tau}}.$$

*Then we have:*

$$\beta\sum_{t=1}^{T-1}\left(F_t^\beta(\mathbf{u}_{t+1}) - F_t^\beta(\mathbf{u}_t)\right) \leq \frac{\gamma}{1-\gamma}P_T^\gamma.$$

### E.2. FTRL Lemmas

For completeness, we present several key FTRL results that will be used in our analysis.

**Standard FTRL.** We consider FTRL in the following form: for a closed convex set $\mathcal{X} \subseteq \mathbb{R}^d$,

$$\mathbf{x}_{t+1} \;=\; \arg\min_{\mathbf{x}\in\mathcal{X}}\left\{\psi_{t+1}(\mathbf{x}) + \sum_{s=1}^{t}f_s(\mathbf{x})\right\}, \tag{58}$$

where $f_t : \mathcal{X} \to \mathbb{R}$ is convex and $\psi_t : \mathcal{X} \to \mathbb{R} \cup \{+\infty\}$ is a convex regularizer. For convenience, we define

$$F_t(\mathbf{x}) \;=\; \psi_t(\mathbf{x}) + \sum_{s=1}^{t-1}f_s(\mathbf{x}), \tag{59}$$

so that $\mathbf{x}_t \in \arg\min_{\mathbf{x}\in\mathcal{X}}F_t(\mathbf{x})$ and $\mathbf{x}_{t+1} \in \arg\min_{\mathbf{x}\in\mathcal{X}}\big(F_t(\mathbf{x}) + f_t(\mathbf{x}) + \psi_{t+1}(\mathbf{x}) - \psi_t(\mathbf{x})\big)$.

**Lemma 19** (Lemma 7.1 in Orabona (2019))**.** *FTRL with the update in Eq. (58) satisfies the following static regret decomposition: for any comparator $\mathbf{u} \in \mathcal{X}$,*

$$\sum_{t=1}^{T}f_t(\mathbf{x}_t) - f_t(\mathbf{u}) = \psi_T(\mathbf{u}) - \min_{\mathbf{x}\in\mathcal{X}}\psi_1(\mathbf{x}) + \sum_{t=1}^{T}\left(F_t(\mathbf{x}_t) - F_{t+1}(\mathbf{x}_{t+1}) + f_t(\mathbf{x}_t)\right) + F_{T+1}(\mathbf{x}_{T+1}) - F_{T+1}(\mathbf{u}),$$

*where $F_t(\cdot)$ is defined in (59).*

The next lemma is the one-step "stability" bound that is frequently used in FTRL analyses. It makes the role of strong convexity explicit, and the standard quadratic-regularizer statement becomes a direct corollary.

**Lemma 20** (Lemma 7.6 in Orabona (2019)). *Suppose $f_t$ is convex and differentiable on $\mathcal{X}$, and $\psi_t$ is $\lambda_t$-strongly convex with respect to a norm $\|\cdot\|$ (hence $F_t$ in (59) is also $\lambda_t$-strongly convex). Let $\mathbf{x}_t \in \arg\min_{\mathbf{x} \in \mathcal{X}} F_t(\mathbf{x})$ and $\mathbf{x}_{t+1} \in \arg\min_{\mathbf{x} \in \mathcal{X}} F_{t+1}(\mathbf{x})$, and let $\mathbf{g}_t \in \partial f_t(\mathbf{x}_t)$. Then*

$$F_t(\mathbf{x}_t) - F_{t+1}(\mathbf{x}_{t+1}) + f_t(\mathbf{x}_t) \le \frac{\|\mathbf{g}_t\|_*^2}{2\lambda_t} + \psi_t(\mathbf{x}_{t+1}) - \psi_{t+1}(\mathbf{x}_{t+1}), \tag{60}$$

*where $\|\cdot\|_*$ is the dual norm of $\|\cdot\|$.*

*In particular, if $\psi_t(\mathbf{x}) = \frac{1}{2\eta_{t-1}}\|\mathbf{x}\|_2^2$, then $\psi_t$ is $(1/\eta_{t-1})$-strongly convex w.r.t. $\|\cdot\|_2$, and (60) becomes*

$$F_t(\mathbf{x}_t) - F_{t+1}(\mathbf{x}_{t+1}) + f_t(\mathbf{x}_t) \le \frac{\eta_{t-1}}{2}\|\mathbf{g}_t\|_2^2 + \psi_t(\mathbf{x}_{t+1}) - \psi_{t+1}(\mathbf{x}_{t+1}).$$

We next state a scale-free, one-step bound adapted from Orabona & Pál (2018), used in the proof of Theorem 5.

**Lemma 21** (Adapted from Theorem 1 in Orabona & Pál (2018)). *Let $\mathcal{X} \subseteq \mathbb{R}^d$ be a closed convex set with diameter $D_\mathcal{X} = \max_{\mathbf{x},\mathbf{y} \in \mathcal{X}} \|\mathbf{x} - \mathbf{y}\|_2$. Assume the losses are linear:*

$$f_t(\mathbf{x}) = \langle \mathbf{g}_t, \mathbf{x} \rangle, \qquad \mathbf{g}_t \in \mathbb{R}^d.$$

*Consider FTRL in (58) with quadratic regularizers $\psi_t(\mathbf{x}) = \frac{1}{2\eta_{t-1}}\|\mathbf{x}\|_2^2$ for some step sizes $\eta_{t-1} > 0$. Let $F_t(\cdot)$ be defined in (59), and define the cumulative gradients $\mathbf{g}_{1:t} = \sum_{s=1}^t \mathbf{g}_s$. Then we have:*

$$F_t(\mathbf{x}_t) - F_{t+1}(\mathbf{x}_{t+1}) + f_t(\mathbf{x}_t)$$
$$\le \eta_t\|\mathbf{g}_t\|_2^2 + \min\left\{\frac{\eta_{t-1}}{2}\|\mathbf{g}_t\|_2^2, \min\{D_\mathcal{X}, |\eta_{t-1} - \eta_t|\,\|\mathbf{g}_{1:t-1}\|_2\} \cdot \|\mathbf{g}_t\|_2\right\} + \psi_t(\mathbf{x}_{t+1}) - \psi_{t+1}(\mathbf{x}_{t+1}).$$

*Proof.* Using the definition of $F_t$ and adding/subtracting $F_t(\mathbf{x}_{t+1})$ and $f_t(\mathbf{x}_{t+1})$, we obtain

$$F_t(\mathbf{x}_t) - F_{t+1}(\mathbf{x}_{t+1}) + f_t(\mathbf{x}_t) = \underbrace{\left(F_t(\mathbf{x}_t) - F_t(\mathbf{x}_{t+1})\right)}_{\text{TERM-A}} + \underbrace{\left(f_t(\mathbf{x}_t) - f_t(\mathbf{x}_{t+1})\right)}_{\text{TERM-B}} + \underbrace{\left(\psi_t(\mathbf{x}_{t+1}) - \psi_{t+1}(\mathbf{x}_{t+1})\right)}_{\text{TERM-C}}. \tag{61}$$

Since $\mathbf{x}_t \in \arg\min_{\mathbf{x}} F_t(\mathbf{x})$, we have TERM-A $\le 0$. Moreover $f_t$ is linear, so TERM-B $= \langle \mathbf{g}_t, \mathbf{x}_t - \mathbf{x}_{t+1}\rangle \le \|\mathbf{g}_t\|_2\|\mathbf{x}_t - \mathbf{x}_{t+1}\|_2$. Hence (61) implies

$$F_t(\mathbf{x}_t) - F_{t+1}(\mathbf{x}_{t+1}) + f_t(\mathbf{x}_t) \le \|\mathbf{g}_t\|_2\|\mathbf{x}_t - \mathbf{x}_{t+1}\|_2 + \psi_t(\mathbf{x}_{t+1}) - \psi_{t+1}(\mathbf{x}_{t+1}). \tag{62}$$

We upper bound $\|\mathbf{x}_t - \mathbf{x}_{t+1}\|_2$ in two ways.

First, since $f_s(\mathbf{x}) = \langle \mathbf{g}_s, \mathbf{x}\rangle$ and $\psi_t(\mathbf{x}) = \frac{1}{2\eta_{t-1}}\|\mathbf{x}\|_2^2$, the FTRL admits the projection form

$$\mathbf{x}_t = \Pi_\mathcal{X}\left[-\eta_{t-1}\mathbf{g}_{1:t-1}\right], \qquad \mathbf{x}_{t+1} = \Pi_\mathcal{X}\left[-\eta_t\mathbf{g}_{1:t}\right],$$

where $\Pi_\mathcal{X}$ is the Euclidean projection onto $\mathcal{X}$. By non-expansiveness of projection,

$$\begin{aligned}
\|\mathbf{x}_t - \mathbf{x}_{t+1}\|_2 &= \left\|\Pi_\mathcal{X}\left[-\eta_{t-1}\mathbf{g}_{1:t-1}\right] - \Pi_\mathcal{X}\left[-\eta_t\mathbf{g}_{1:t}\right]\right\|_2 \\
&\le \left\|-\eta_{t-1}\mathbf{g}_{1:t-1} + \eta_t\mathbf{g}_{1:t}\right\|_2 = \left\|\eta_t\mathbf{g}_t - (\eta_{t-1} - \eta_t)\mathbf{g}_{1:t-1}\right\|_2 \\
&\le \eta_t\|\mathbf{g}_t\|_2 + |\eta_{t-1} - \eta_t|\,\|\mathbf{g}_{1:t-1}\|_2.
\end{aligned} \tag{63}$$

Second, directly $\|\mathbf{x}_t - \mathbf{x}_{t+1}\|_2 \le D_\mathcal{X}$. Combining this fact and (63) yields

$$\|\mathbf{g}_t\|_2\|\mathbf{x}_t - \mathbf{x}_{t+1}\|_2 \le \min\left\{D_\mathcal{X}\|\mathbf{g}_t\|_2, \eta_t\|\mathbf{g}_t\|_2^2 + |\eta_{t-1} - \eta_t|\,\|\mathbf{g}_{1:t-1}\|_2\|\mathbf{g}_t\|_2\right\}$$

$$\leq \eta_t \|\mathbf{g}_t\|_2^2 + \min\left\{D_{\mathcal{X}}, |\eta_{t-1} - \eta_t| \|\mathbf{g}_{1:t-1}\|_2\right\} \cdot \|\mathbf{g}_t\|_2.$$

Substituting the above bound into (62) gives

$$F_t(\mathbf{x}_t) - F_{t+1}(\mathbf{x}_{t+1}) + f_t(\mathbf{x}_t) \leq$$
$$\eta_t \|\mathbf{g}_t\|_2^2 + \min\left\{D_{\mathcal{X}}, |\eta_{t-1} - \eta_t| \|\mathbf{g}_{1:t-1}\|_2\right\} \cdot \|\mathbf{g}_t\|_2 + \psi_t(\mathbf{x}_{t+1}) - \psi_{t+1}(\mathbf{x}_{t+1}). \tag{64}$$

On the other hand, applying Lemma 20 with $\lambda_t = 1/\eta_{t-1}$ gives

$$F_t(\mathbf{x}_t) - F_{t+1}(\mathbf{x}_{t+1}) + f_t(\mathbf{x}_t) \leq \frac{\eta_{t-1}}{2}\|\mathbf{g}_t\|_2^2 + \psi_t(\mathbf{x}_{t+1}) - \psi_{t+1}(\mathbf{x}_{t+1}). \tag{65}$$

Taking the minimum between (64) and (65), and using $\min\{a, c+b\} \leq c + \min\{a,b\}$ for $c \geq 0$, completes the proof. $\square$

**Optimistic FTRL.** We consider an optimistic variant of FTRL (Rakhlin & Sridharan, 2013; Orabona, 2019), where an optimistic function $h_t(\mathbf{x})$ is used when providing $\mathbf{x}_t$. This function can be realized as a "guess" of the incoming function.

$$\mathbf{x}_t = \arg\min_{\mathbf{x}\in\mathcal{X}}\left\{\psi_t(\mathbf{x}) + h_t(\mathbf{x}) + \sum_{s=1}^{t-1} f_s(\mathbf{x})\right\}. \tag{66}$$

**Lemma 22** (Adapted from Theorem 7 in Jézéquel et al. (2020)). *Define the decision without optimistic functions to be $\widehat{\mathbf{x}}_t$, such that,*

$$\widehat{\mathbf{x}}_t \in \arg\min_{\mathbf{x}\in\mathcal{X}} F_t(\mathbf{x}), \qquad F_t(\mathbf{x}) = \psi_t(\mathbf{x}) + \sum_{s=1}^{t-1} f_s(\mathbf{x}).$$

*For the optimistic FTRL in Eq. (66), for any comparator $\mathbf{u} \in \mathcal{X}$, we have*

$$\sum_{t=1}^T f_t(\mathbf{x}_t) - f_t(\mathbf{u}) = \psi_{T+1}(\mathbf{u}) - \min_{\mathbf{x}\in\mathcal{X}}\psi_1(\mathbf{x}) + \sum_{t=1}^T \left(f_t(\mathbf{x}_t) + F_t(\widehat{\mathbf{x}}_t) - F_{t+1}(\widehat{\mathbf{x}}_{t+1})\right) + F_{T+1}(\widehat{\mathbf{x}}_{T+1}) - F_{T+1}(\mathbf{u}).$$

*In particular, since $F_{T+1}(\widehat{\mathbf{x}}_{T+1}) \leq F_{T+1}(\mathbf{u})$, we have the upper bound*

$$\sum_{t=1}^T \left(f_t(\mathbf{x}_t) - f_t(\mathbf{u})\right) \leq \psi_{T+1}(\mathbf{u}) - \min_{\mathbf{x}\in\mathcal{X}}\psi_1(\mathbf{x}) + \sum_{t=1}^T \left(f_t(\mathbf{x}_t) + F_t(\widehat{\mathbf{x}}_t) - F_{t+1}(\widehat{\mathbf{x}}_{t+1})\right).$$

### E.3. Self-Confident Tuning Lemmas

**Lemma 23** (Lemma 5 in Xie et al. (2024)). *Let $a_0 > 0$ and $a_t \in [0,B]$ be real numbers for all $t \in [T]$ and let $f : (0, +\infty) \to [0, +\infty)$ be a nonincreasing function. Then*

$$\sum_{t=1}^T a_t f\left(\sum_{s=0}^{t-1} a_s\right) \leq B \cdot f(a_0) + \int_{a_0}^{\sum_{t=0}^T a_t} f(u)\mathrm{d}u.$$

**Lemma 24** (Lemma 3.5 in Auer et al. (2002)). *Let $a_1, \ldots, a_T$ and $\delta$ be non-negative real numbers. Then*

$$\sum_{t=1}^T \frac{a_t}{\sqrt{\delta + \sum_{s=1}^t a_s}} \leq 2\left(\sqrt{\delta + \sum_{t=1}^T a_t} - \sqrt{\delta}\right).$$

**Lemma 25** (Lemma G.2 in Jacobsen & Cutkosky (2024)). *Let $\beta \in (0,1]$, $\lambda > 0$, $\mathbf{z}_t \in \mathbb{R}^d$, and define $A_0 = \lambda I$ and*

$$A_t = \mathbf{z}_t\mathbf{z}_t^\top + \beta A_{t-1} \qquad \text{for each } t > 0.$$

*Then for any sequence $c_1, c_2, \ldots \in \mathbb{R}$,*

$$\sum_{t=1}^T c_t^2 \mathbf{z}_t^\top A_t^{-1} \mathbf{z}_t \leq d\ln\left(\frac{1}{\beta}\right)\sum_{t=1}^T c_t^2 + \left\{\max_{t\in[T]} c_t^2\right\} \cdot d\ln\left(1 + \frac{\sum_{t=1}^T \beta^{T-t}\|\mathbf{z}_t\|_2^2}{\lambda d}\right).$$

