# OpenReview forum: "Dynamic Regret via Discounted-to-Dynamic Reduction with Applications to Curved Losses and Adam Optimizer"
_ICML.cc/2026/Conference — ICML 2026 regular_

### Official Review · Reviewer_oXr4 · 2026-02-22

**Soundness:** 2
**Presentation:** 1
**Significance:** 3
**Originality:** 3
**Overall Recommendation:** 4
**Confidence:** 2

**Summary:**

The paper introduces the *Modular D2D reduction*, a general reduction from discounted regret minimization, where losses are exponentially discounted, to dynamic regret minimization. The key idea is that any algorithm with a discounted regret guarantee can be converted, in a modular way, into an algorithm with dynamic regret guarantees.

The paper demonstrates several applications of this reduction. First, for online linear regression (OLR), the authors apply the reduction to an existing discounted-regret algorithm and obtain dynamic regret guarantees.

Second, for online logistic regression, they introduce a discounted variant of AIOLI and use the Modular D2D reduction to derive dynamic regret bounds.

Finally, the paper applies the framework to analyze Adam. The authors build on the O2NC framework, which reduces stochastic non-convex and non-smooth optimization to an online learning problem. They interpret Adam as an instance of discounted FTRL within this framework. Since bounding dynamic regret suffices to control the O2NC objective, the Modular D2D reduction yields dynamic regret guarantees for both clipped and non-clipped Adam, leading to convergence guarantees to a $(c,\varepsilon)$-stationary point.

**Compliance With Llm Reviewing Policy:**

Affirmed.

**Final Justification:**

The main strength of the paper is that it adds to the theoretical understanding of the widely used Adam algorithm. I do not find the contributions in the other parts of the paper to be particularly significant, but the Adam-related contribution is, in my view, sufficient.

The main weakness, in my opinion, is the presentation. The main reduction involves parameters that are not fully explained, at least in the main text. In addition, the Adam section does not clearly state the connection between O2NC+FTRL and Adam, which makes it harder to appreciate the contribution. The rebuttal was important for me to fully understand this point.

Nevertheless, I believe these issues are mainly related to clarity and can be addressed in a revised version. I therefore lean towards acceptance.

**Key Questions For Authors:**

1. The paper states that OMD generally reacts more quickly to changes than FTRL. Could you elaborate on this claim? My understanding is that responsiveness is primarily governed by the learning rate and the curvature (second derivative) of the regularizer, rather than by whether the algorithm is instantiated as FTRL or OMD. Is there a formal or intuitive reason supporting this distinction?

2. It would be very helpful to provide a clear and explicit mapping from the parameters of O2NC + discounted FTRL to the hyperparameters of Adam. Since both formulations involve multiple parameters, an organized proposition summarizing the correspondence would significantly improve clarity and help verify the correctness of the reduction.

3. Could you provide a clearer comparison to the state-of-the-art results for each application (OLR, logistic regression, and Adam)?

4. In Line 244, $h_t(x)$ is described as being ``realized as a guess of the incoming loss function.'' Could you clarify this interpretation? In particular, should the empirical probability of observing $1/-1$ not evolve over time rather than remain fixed?

5. Typos - Line 212: what does ``fst'' denote? Should this be $f_s$? Algorithm 1, Line 6: should this be $\Delta_t$?

**Limitations:**

yes

**Strengths And Weaknesses:**

**Soundness:**
The overall structure of the arguments appears mathematically coherent, and I did not identify a concrete technical error. However, some key components of the Modular D2D theorem are insufficiently specified, which makes it difficult to fully verify correctness.
In particular, the role of the quantity $\Lambda$ (described as a ``stability term'') is not clearly formalized. It is unclear what precise properties $\Lambda$ must satisfy and whether it is uniquely defined or can take multiple forms. Because the assumptions of the reduction are not stated in a fully explicit manner, it is difficult to rigorously check that the later applications (OLR, logistic regression, and Adam) satisfy all required conditions.
Thus, while the high-level reasoning appears sound, the lack of precise specification limits my ability to fully assess the technical correctness.

**Presentation:**
I found the presentation challenging, particularly in the exposition of the Modular D2D theorem. As the central contribution of the paper, the theorem would benefit from a more self-contained and structured statement, with clearly enumerated assumptions and a precise description of all terms involved.
Because of this, when the reduction is applied in later sections, it is difficult to clearly see how the preceding analysis fits into the theorem’s conditions. While the connections appear mathematically elegant, the logical alignment between assumptions and conclusions is not sufficiently transparent.
The Adam section is especially hard to follow. Both Adam and discounted FTRL + O2NC involve multiple parameters, but the paper does not provide a clear proposition mapping the parameters of discounted FTRL + O2NC to the corresponding Adam hyperparameters. An explicit parameter correspondence would substantially improve clarity.
Finally, the paper lacks a structured literature comparison, which makes it difficult to contextualize the strength of the results relative to prior work.

**Originality:**
The connection between discounted regret and dynamic regret is not entirely new, and prior work has already explored relationships between these notions. The Modular D2D reduction refines and systematizes this connection into a more modular and reusable framework. While this does not introduce a fundamentally new paradigm, it provides a cleaner and more unified perspective.
The applications demonstrate the breadth of the framework. The OLR result matches known optimal rates, and the Adam analysis improves upon prior work by relaxing the commonly used assumption $\beta_2 \ge \beta_1^2$ to the weaker condition $\beta_2 \ge \beta_1^4$. This relaxation is a meaningful refinement in the theoretical understanding of Adam’s convergence behavior.
Overall, the contribution lies primarily in unification and refinement, together with this improved analysis of Adam, rather than in introducing fundamentally new algorithmic ideas.

**Significance:**
The Modular D2D reduction provides a clean and potentially reusable analytical tool, and the idea of modularly converting discounted regret guarantees into dynamic regret guarantees is appealing.
However, the impact of the applications varies. While the OLR result matches optimal rates (possibly with a simplified proof), the logistic regression result does not improve upon the state of the art. In the Adam application, although the conclusion claims optimal rates, the paper does not clearly position the result relative to existing convergence guarantees, making it difficult to assess its significance.
Overall, the contribution appears more as a unifying and refining framework than as a breakthrough in rates or algorithmic performance.

---

> ### Author Rebuttal · Authors · 2026-03-30
>
> Thank you for the review. We address your concerns below.
>
> ---
>
> **Q1.** "OMD reacts more quickly to changes than FTRL?"
>
> **A.** The key distinction is structural. OGD (a special case of OMD): $x_{t+1} = \Pi[x_t - \eta g_t]$ moves from the *current* iterate using only the latest gradient, naturally forgetting the past. FTRL: $x_{t+1} = \Pi[-\eta \sum_{s=1}^t g_s]$, aggregating all historical gradients with equal weight, making it difficult to adapt.
>
> Formally, [3, Theorem 2] show that vanilla FTRL can suffer $\Omega(T)$ dynamic regret even when $P_T = O(1)$, whereas OMD achieves $O(\sqrt{T})$. This motivates the discounting mechanism in our paper.
>
> ---
>
> **Q2.** "Explicit parameter mapping from O2NC + discounted FTRL to Adam."
>
> **A.** We provide the mapping here and will add details in the revision.
>
> | Discounted FTRL + O2NC | Adam |
> |--|--|
> | Discount factor $\beta$ | $\beta_1$ |
> | Domain $\mathcal{D} = \{\Delta: \|\Delta\| \le D\}$ | Clipping radius $D$ (clip-free: $\mathbb{R}^d$) |
> | Loss $\ell_t(\Delta) = \langle g_t, \Delta \rangle$ | Stochastic gradient $g_t$ |
>
> The discounted FTRL update in Eq. (6) recovers the clipped Adam update in Eq. (8) and clip-free Adam in Eq. (10).
>
> ---
>
> **Q3.** "Clearer comparison to state-of-the-art results."
>
> **A.** These comparisons are in Section 3.2, Appendix A (logistic regression) and Section 4 (Adam).
>
> **Logistic regression.** Our result has linear dependence on $B$, whereas exp-concave guarantees incur exponential dependence. The only result on unbounded domains is [1], achieving $\tilde{O}(T^{1/3}P_T^{2/3})$ but requiring Monte Carlo sampling. Our result offers a tradeoff between efficiency and rate.
>
> **Adam.** The only prior result is [2] with $\beta_2 = \beta_1^2$. We provide: (i) flexible $(\beta_1, \beta_2)$ conditions; (ii) first clip-free variant; (iii) optimal rates matching lower bounds. By choosing $c$ appropriately (Section 4.5), we also recover optimal rates for smooth and second-order smooth settings.
>
> ---
>
> **Q4.** "$h_t(x)$ as a guess. Should probability of $+1/-1$ evolve?"
>
> **A.** The term "guess" is intuitive; formally $h_t$ is an optimism term in optimistic FTRL. The equal weighting $h_t(x) = \ell(x^\top z_t, +1) + \ell(x^\top z_t, -1)$ is not a probability estimate — our setting is adversarial, so estimating label frequencies provides no guarantees. The equal-weight sum is a worst-case hedge ensuring the gap between $h_t$ and $f_t$ is controlled. This follows the AIOLI algorithm [4].
>
> ---
>
> **Q5.** "Typos."
>
> **A.** Thanks for pointing them out. "fst" in Line 212 should read $\sum_{s=1}^t f_s(u)$. For Algorithm 1 Line 6: the intent is to provide the loss *function* $\ell_t(\cdot)$, not the scalar $\ell_t(\Delta_t)$. We will fix the first point.
>
> ---
>
> **W1.** "Role of $\Lambda$ not clearly formalized."
>
> **A.** $\Lambda_s \ge 0$ in Theorem 1 is determined by the base algorithm — it captures per-round stability cost. The paper discusses this below Theorem 1, and each application specifies its $\Lambda_s$: Lemma 1 (OLR), Lemma 2 (logistic), Section 4.2 (Adam). We will add clearer cross-references.
>
> ---
>
> **W2.** "D2D theorem needs more self-contained presentation."
>
> **A.** Section 2 presents the reduction independently of any application. The three applications are self-contained instantiations. We will improve cross-references.
>
> ---
>
> **W3.** "Adam section hard to follow."
>
> **A.** (See Q2 above.)
>
> ---
>
> **W4.** "Lacks structured literature comparison."
>
> **A.** Appendix A provides a detailed comparison covering non-stationary online learning, discounted learning, Adam convergence, and O2NC. We will improve positioning in the revision.
>
> ---
>
> **W5.** "Contribution lies primarily in unification and refinement."
>
> **A.** We acknowledge that the D2D connection builds on prior work. Beyond modularization, the Adam application requires new technical tools (self-confident tuning lemma, scale-free analysis) to derive convergence under flexible $(\beta_1, \beta_2)$, including the first clip-free Adam analysis and optimal rates to $(c, \epsilon)$-stationary points. We will clarify these contributions in the revision.
>
> ---
>
> **W6.** "Logistic regression/Adam not clearly positioned."
>
> **A.** (See Q3 above.)
>
> ---
>
> We note that several concerns (role of $\Lambda_s$, literature comparison, positioning) are addressed in the manuscript. We will improve presentation to make them more prominent. If our responses have addressed your concerns, please consider updating your score. Thanks!
>
> ---
>
> **References**
>
> [1] Zhang et al., Non-stationary Online Learning for Curved Losses: Improved Dynamic Regret via Mixability.
>
> [2] Ahn and Cutkosky, Adam with Model Exponential Moving Average is Effective for Nonconvex Optimization.
>
> [3] Jacobsen and Cutkosky, Parameter-free Mirror Descent.
>
> [4] Jézéquel et al., Efficient Improper Learning for Online Logistic Regression.

---

> > ### Author Rebuttal · Reviewer_oXr4 · 2026-04-02
> >
> > **Connection between O2NC+FTRL and Adam.**
> >
> > The point I initially missed is that equations (8) and (10) effectively define Adam. I believe the paper would benefit from making this connection more explicit. In particular, it would help to first formally define Adam (both clipped and unclipped variants), and then state a clear proposition showing that O2NC+FTRL yields updates equivalent (or closely related) to Adam. This would make the contribution more transparent. I hope the authors will incorporate such a clarification in the final version.
> >
> > **Significance of the paper.**
> >
> > Given that a $\sqrt{TP}$ bound is achievable, I do not see the $T^{2/3}P^{1/3}$ bound as a significant contribution on its own, even if the algorithm is simpler or more efficient, unless the prior approaches have substantially worse computational complexity (e.g., exponential runtime), in which case this should be stated clearly. That said, I now better appreciate the contribution in the Adam section, which in my view is sufficient to justify the paper.
> >
> > Small note - OGD can also be interpreted as FTRL.
> >
> > Therefore, I increase my score to 4

---

> > > ### Author Response · Authors · 2026-04-03
> > >
> > > Thank you for the response and for re-evaluating our work. Based on your comments, we provide the following supplementary clarifications.
> > >
> > > - **Adam:** We agree with your suggestion. We will add a clear proposition showing that the O2NC + discounted FTRL update recovers the Adam-like update, along with a comparison table summarizing our results versus prior work. We believe this is feasible given the additional page allowed in the camera-ready version.
> > >
> > > - **Online logistic regression:** We agree that this result should be presented more clearly. The key issue is the tradeoff between convergence rate and computational efficiency:
> > >
> > > | | Dynamic Regret | Per-round Cost |
> > > |--|--|--|
> > > | Baby and Wang, COLT'21 | $\tilde{O}(d^{3.5}e^B T^{1/3}P_T^{2/3})$ | $O(d^3\log T)$ |
> > > | Zhang et al., ICML'25 | $\tilde{O}(dT^{1/3}P_T^{2/3})$ | $O(d^{12}T^{11}\log T + T^{23}\log T)$ |
> > > | Ours (Theorem 8) | $O(dB\log(BT) + \sqrt{dBTP_T^{\beta_\star}})$ | $O(d^2\log T + \log^2 T)$ |
> > >
> > > The analysis in Baby and Wang (COLT'21) relies on comparator-loss-specific structure via KKT conditions, which makes it unclear how to further improve the exponential dependence on $B$. Zhang et al. (ICML'25) leverage the mixability property to obtain the tightest guarantees, but their approach requires MCMC sampling, resulting in substantially higher computational costs. Our method achieves a tradeoff between rate and efficiency while admitting a transparent, template-level analysis.
> > >
> > > - **OGD and FTRL:** Thank you for this observation. As discussed in Orabona (2019, Section 7.3.1), OMD and FTRL are equivalent under specific configurations (barrier regularizer, fixed learning rate, linearized losses). Indeed, Jacobsen and Cutkosky (ICML'24) on online linear regression rewrites FTRL as OMD to analyze dynamic regret. However, for more general configurations and for understanding Adam as discounted FTRL, the FTRL formulation offers structural advantages. Our work explores building modular analyses for discounted FTRL in this context.
> > >
> > > We will incorporate these discussions in the revision to improve clarity. Thank you again for your feedback.

---

### Official Review · Reviewer_2y4z · 2026-03-12

**Soundness:** 2
**Presentation:** 4
**Significance:** 3
**Originality:** 3
**Overall Recommendation:** 4
**Confidence:** 1

**Summary:**

This article examines the minimization of dynamic regret in non-stationary online learning, with a primary focus on Follow-the-Regularized-Leader (FTRL) methods. The authors apply modular reduction to two representative regression tasks. Specifically, for online linear regression, modular reduction simplifies the analysis by circumventing algorithm-specific Bregman divergence arguments, while still achieving the corresponding optimal bound. For online logistic regression, it provides a dynamic regret guarantee for a discounted variant of AIOLI. Furthermore, the authors demonstrate that D2D can be applied to analyze the convergence rate of ADAM in stochastic non-convex and non-smooth settings via the online-to-nonconvex (O2NC) conversion.

**Compliance With Llm Reviewing Policy:**

Affirmed.

**Final Justification:**

I thank the authors for their detailed answers and for the additional experiments they provided. Many of my concerns were resolved, with the exception of the assumption $G < \infty$, which I consider restrictive. Nevertheless, I am inclined to leave a positive assessment of this work.

**Key Questions For Authors:**

Could the authors please clarify what they mean by the optimality of the convergence rate of ADAM (line 028)?

**Strengths And Weaknesses:**

First of all, I would like to note that I am not an expert in the field of online optimization. Nevertheless, I would like to share my perspective on the strengths and weaknesses of the paper below. If any of my comments are based on misunderstandings, I would be very happy to discuss them with the authors.

**Soundness:** The paper appears to be technically sound, and the theoretical claims are supported by the analysis provided in the appendix. However, the work does not include an experimental section, which makes the paper purely theoretical. In my opinion, this is more of a limitation than a strength, especially since the authors consider algorithms that are widely used in practical applications, such as ADAM. The proofs seem correct overall; however, some of the assumptions appear to be rather strong. For example, Assumption 1 seems to implicitly require that $G < \infty$, although this is not explicitly stated. If this is indeed the intended assumption, it would be helpful to clarify whether there exist meaningful classes of functions that satisfy it. On the other hand, if the case  $G=\infty$ is allowed, then in Theorem 4 we obtain $T = \infty$, which would imply that no convergence guarantees are provided. In addition, the assumption that $||g_t || \leq G$ may also be quite restrictive and could benefit from further discussion or justification.

**Presentation:** The paper is clearly written, and the presentation is well structured. The exposition is generally easy to follow, even for readers who are not experts in this specific area. In my opinion, the work positions itself appropriately within the context of related literature.

**Significance** The paper addresses an important problem in optimization. It certainly contributes to a better theoretical understanding of optimization methods relevant to machine learning. However, as mentioned above, the work remains purely theoretical and does not support its claims with numerical experiments. Nevertheless, the results may provide a useful foundation for future research in this direction.

**Originality:** The paper deepens the understanding of important properties of existing optimization methods. It presents perspectives that could contribute to further progress in the field of online optimization. The novelty of the work is clearly articulated and distinguishable from prior related research.

---

> ### Author Rebuttal · Authors · 2026-03-30
>
> Thank you for your review. We appreciate the positive assessment on presentation and originality. Below we address your concerns.
>
> ---
>
> **Clarification: Paper Motivation and Roadmap.**
>
> We would like to first elaborate the roadmap of this paper to facilitate understanding. FTRL is effective for exploiting problem structure (curved losses, adaptive optimizers like Adam), but its reliance on the full history limits adaptivity in non-stationary settings: vanilla FTRL can suffer linear dynamic regret even under mild non-stationarity. Our paper proposes a modular reduction for discounted FTRL to obtain dynamic regret guarantees, with direct applications to online linear/logistic regression. Furthermore, via the O2NC conversion [1, 2], analyzing Adam in stochastic, non-convex, and non-smooth settings can be reduced to an online learning problem, allowing us to leverage the tools developed in modular reduction and online self-confident tuning to derive convergence guarantees for Adam.
>
> ---
>
> **Q1.** "Could the authors please clarify what they mean by the optimality of the convergence rate of ADAM (line 028)?"
>
> **A.** Thank you for this question. By "optimal convergence rate" we mean that our convergence guarantees for Adam match the known *lower bound* for finding a $(c, \epsilon)$-stationary point in stochastic, non-convex, and non-smooth optimization.
>
> Specifically, [2] establish a lower bound of $\Omega(F^* G^2 c^{1/2} \epsilon^{-7/2})$ for finding a $(c, \epsilon)$-stationary point. As stated in our Theorem 4 (and the discussion below it), the leading term of our iteration complexity is $\mathcal{O}(F^* G^2 c^{1/2} \epsilon^{-7/2})$, which matches this lower bound. For the margin-style convergence condition (Theorem 5 \& 7), the theorems suggest a choice of $\beta_2 = (1 + \beta_1^2)/2$ (corresponding to $\rho = 0$). To support a broader range of $(\beta_1, \beta_2)$, we allow $\rho \in [0, 1)$, which introduces an additional $(1-\rho)^{-1}$ factor in the convergence rate. Treating this factor as a constant and under the mild condition $G/\nu \le \mathcal{O}(\exp(G/\epsilon))$, the margin-style results also match the lower bound. Moreover, when the objective is smooth, by setting $c$ with the smoothness constants accordingly (details in Appendix D.4), our results also imply optimal rates in the non-convex smooth and second-order smooth settings as special cases.
>
> We will provide a more detailed discussion on the optimality of convergence rates in the revised version.
>
> ---
>
> **W1.** "Assumption 1 seems to implicitly require that $G < \infty$"
>
> **A.** Yes, Assumption 1 indeed requires $G < \infty$, i.e., $F$ is Lipschitz with a finite constant $G$. This is a standard assumption in the O2NC framework and the non-convex, non-smooth optimization literature [1, 2]. We will make this explicit in the revised version for clarity.
>
> ---
>
> **W2.** "The assumption $g_t \le G$ may be quite restrictive"
>
> **A.** This is a technical trade-off. While not a general assumption in non-convex optimization, bounded stochastic gradients have been adopted in related analyses [3, 4, 5]. In our analysis, this condition is used to facilitate the telescoping argument where $\mathbb{E}[\max_t \|{g}_t\|_2]$ appears, enabling more fine-grained convergence conditions on $(\beta_1, \beta_2)$. The only prior result for Adam under non-smooth settings, [6], does not require this assumption but is restricted to $\beta_2 = \beta_1^2$. As noted below Assumption 4, this can be relaxed to tail conditions at the price of extra logarithmic factors.
>
> ---
>
> **W3.** On the lack of experiments.
>
> **A.** During the rebuttal period, we conducted experiments on CIFAR-10/ResNet-18 with L1 regularization ($\lambda\|x\|_1$), creating a non-smooth objective. Using the same lr $=0.001$ and $(\beta_1,\beta_2)=(0.9, 0.999)$ as default Adam, both Clipped Adam and Clip-Free Adam consistently improve over Adam, with the advantage growing as $\lambda$ increases. Full results: [https://osf.io/8dqsj/](https://osf.io/8dqsj/files/mp56d?view_only=69d6ff6561684f1db8de57259cdfc590).
>
> ---
>
> We hope our responses provide a clearer picture of the paper's contributions and technical choices. We warmly invite you to re-evaluate our work and consider updating your score and confidence. Thank you!
>
> ---
>
> **References**
>
> [1] Cutkosky et al., Optimal Stochastic Non-smooth Non-convex Optimization through Online-to-Non-convex Conversion.
>
> [2] Zhang and Cutkosky, Random Scaling and Momentum for Non-smooth Non-convex Optimization.
>
> [3] Cutkosky and Orabona, Momentum-based Variance Reduction in Non-convex SGD.
>
> [4] Guo et al., A Novel Convergence Analysis for Algorithms of the Adam Family.
>
> [5] Défossez et al., A Simple Convergence Proof of Adam and Adagrad.
>
> [6] Ahn and Cutkosky, Adam with Model Exponential Moving Average is Effective for Nonconvex Optimization.

---

> > ### Author Rebuttal · Reviewer_2y4z · 2026-04-03
> >
> > I thank the authors for their detailed answers and for the additional experiments they provided. Many of my concerns were resolved, with the exception of the assumption $G < \infty$, which I consider restrictive. Nevertheless, I am inclined to leave a positive assessment of this work.

---

> > > ### Author Response · Authors · 2026-04-04
> > >
> > > Thank you for the acknowledgement and the positive assessment. We would like to further address your remaining concern about the $G < \infty$ assumption.
> > >
> > > At a high level, in the absence of smoothness, gradients can vary arbitrarily, making it difficult to establish meaningful convergence guarantees. The $G$-Lipschitz condition ($|F(x) - F(y)| \le G\|x - y\|$, $G < \infty$) serves as a critical regularity condition in this setting. Zhang et al. [1] adopted this condition to define the function class for studying finite-time complexity of non-smooth non-convex optimization, and it has since been used throughout the subsequent line of research [2–5].
> > >
> > > Our work builds on the exponential O2NC framework [6], which fits naturally with Adam's update structure and also enables clip-free algorithms to achieve optimal rates. Both [6] and the prior Adam analysis under non-smooth settings [7] adopt the $G$-Lipschitz assumption. Our analysis follows the same standard setup.
> > >
> > > Currently, our results require this assumption, and we will make its role in non-smooth non-convex optimization more explicit in the revision. We hope this clarification addresses your concern and increases your confidence in our work.
> > >
> > > [1] Zhang et al., Complexity of Finding Stationary Points of Nonsmooth Nonconvex Functions, 2020.
> > >
> > > [2] Davis et al., A Gradient Sampling Method with Complexity Guarantees for Lipschitz Functions, 2022.
> > >
> > > [3] Kornowski and Shamir, On the Complexity of Finding Small Subgradients in Nonsmooth Optimization, 2022.
> > >
> > > [4] Jordan et al., On the Complexity of Deterministic Nonsmooth and Nonconvex Optimization, 2022.
> > >
> > > [5] Tian et al., On the Finite-Time Complexity and Practical Computation of Approximate Stationarity Concepts of Lipschitz Functions, 2022.
> > >
> > > [6] Zhang and Cutkosky, Random Scaling and Momentum for Non-smooth Non-convex Optimization, 2024.
> > >
> > > [7] Ahn and Cutkosky, Adam with Model EMA is Effective for Nonconvex Optimization, 2024.

---

### Official Review · Reviewer_vRCu · 2026-03-13

**Soundness:** 3
**Presentation:** 3
**Significance:** 2
**Originality:** 2
**Overall Recommendation:** 4
**Confidence:** 3

**Summary:**

This paper studies dynamic regret for FTRL in non-stationary online learning through a modular discounted-to-dynamic (D2D) reduction. The main technical claim is a template-level reduction that converts discounted regret bounds on rescaled losses into dynamic regret bounds while keeping key stability and comparator-variation terms explicit. The paper then applies this method to two settings:
curved-loss online learning, where it recovers the optimal dynamic regret for discounted VAW in unconstrained online linear regression and gives a new dynamic regret guarantee for discounted AIOLI in online logistic regression; and (2) Adam-type methods, where it combines the reduction with O2NC to derive convergence guarantees for clipped and clip-free Adam under more flexible ( $\beta_1, \beta_2$ ) choices than prior work, including conditions such as $\beta_2 \geq \beta_1^4$ for clipped Adam.

**Compliance With Llm Reviewing Policy:**

Affirmed.

**Final Justification:**

Rebuttal addressed your main concerns. I decided to increase score.

**Key Questions For Authors:**

1. The related-work section notes that it is unclear whether the approach can recover optimal dynamic regret for general convex losses. Can the authors say more about whether this is likely a limitation of the reduction itself or just of the current instantiations?

2. For online logistic regression, can the modular reduction be pushed further to recover the sharper $\widetilde{O}\left(T^{1 / 3} P_T^{2 / 3}\right)$-type rates known for exp-concave settings, or is there a real barrier in the template level D2D approach that prevents this? The paper currently presents its result as a complementary efficiency/transparency trade-off.

**Strengths And Weaknesses:**

Strengths:
The paper has a clear unifying idea, and the modular D2D theorem is interesting. The linear-regression instantiation is clean and appears to recover the optimal rate while simplifying the route taken by prior work. To me, the logistic-regression result is one of the most interesting parts of the paper. In particular, the paper argues that the discounted AIOLI variant avoids the usual exponential dependence on the comparator norm $B$, and obtains a dynamic-regret guarantee of the form
$
O(d B \log T + \sqrt{d T P_T^\beta})
$
via a two-layer tuning scheme. Even if this result is not claimed to be optimal in the bounded-comparator setting, it still seems to offer a meaningful tradeoff between statistical efficiency and proof transparency.

Weaknesses:
1. A substantial part of the novelty lies in modularization and proof streamlining, rather than in establishing stronger end results. For linear regression, the paper mainly recovers a known optimal rate using a cleaner derivation. For logistic regression, the paper explicitly positions its result as a complementary tradeoff, rather than an improvement over the strongest dynamic-regret guarantees known for exp-concave losses. More broadly, I am not yet fully convinced that the ``modular'' theorem, by itself, represents a sufficiently large conceptual advance over the closest prior D2D-based work. The paper argues that making the key terms explicit enables more flexible tuning and broader applicability, but the evidence for the superiority of this viewpoint over more direct, problem-specific reductions still feels somewhat limited. In practice, for linear regression the main contribution is a streamlined proof of a known optimal rate, while for logistic regression and Adam the most substantial gains seem to come from the technical details of the instantiations rather than from the reduction alone.

2. The paper makes several tradeoffs that deserve stronger discussion. In the logistic-regression setting, the paper itself acknowledges that sharper $\tilde{O} (T^{1/3} P_T^{2/3})$ dynamic-regret guarantees are already known for exp-concave losses, whereas this work provides a $\tilde{O}(\sqrt{T P_T})$-type guarantee together with better efficiency and modularity. This is a reasonable tradeoff, but the paper does not yet make a fully convincing case for why this tradeoff is especially important. Similarly, in the Adam section, the more flexible parameter conditions come at the cost of bounded-gradient assumptions and modified updates that omit bias correction and, in the clip-free case, introduce an additional damping term. These choices weaken the practical relevance of the claims about Adam.

3. I did not find any empirical validation. This is not fatal for a theory submission, but for the optimizer-related claims it weakens the practical case, especially because the analyzed algorithms differ in important ways from the standard variants used in practice.

---

> ### Author Rebuttal · Authors · 2026-03-30
>
> Thank you for the review. Below we address your concerns.
>
> ---
>
> **Q1.** "Optimal dynamic regret for general convex losses?"
>
> **A.** Currently we are not able to recover the optimal rate for general convex losses. The core difficulty lies in the comparator-dependent term $\varphi_t(u)$ in Theorem 1. For our curved-loss applications, curvature makes $\varphi_t$ time-independent, so the last term in Theorem 1 vanishes; for Adam, $\|u_t\|=D$ for all $t$ also keeps this term manageable.
>
> For general convex losses, consider discounted FTRL: $x_{t+1} = \arg\min_x \{ \sum_{s=1}^t \beta^{t-s}\langle g_s, x\rangle + \frac{1}{2\eta_t}\|x\|^2 \}$. Apply Theorem 1:
>
> $$\text{D-Reg} \lesssim \sum_t \eta_{t-1}\beta^{-1}\|g_t\|^2 + \beta\sum_t \sum_s \beta^{t-s}\langle g_s, u_{t+1}-u_t\rangle + \beta\sum_t \beta^t(\frac{1}{2\eta_t\beta^t}\|u_{t+1}\|^2 - \frac{1}{2\eta_{t-1}\beta^{t-1}}\|u_t\|^2),$$
>
> where $\varphi_t(u) = \frac{1}{2\eta_{t-1}\beta^{t-1}}\|u\|^2$ (in Thm. 1) is time-dependent. This creates the following tension:
>
> - Constant $\eta$: the first term $= O(\eta G^2 T)$, but the last term becomes $\frac{\beta}{2\eta}\sum_t(\frac{1}{\beta}\|u_{t+1}\|^2 - \|u_t\|^2)$; the $1/\beta$ vs $1$ mismatch prevents forming a path-length quantity, accumulating as $O(\frac{1-\beta}{\beta\eta}\sum_t \|u_t\|^2)$.
>
> - Decaying $\eta_t = \gamma\beta^{-t}$: the last term forms a path length, but the first term $= O(\gamma G^2 \beta^{-T})$ grows exponentially. Tuning $\gamma$ small does not help since there is a factor $\frac{1}{\gamma}$ in the last term that explodes.
>
> Neither choice simultaneously controls all terms to achieve the optimal $O(\sqrt{T(1+P_T)})$ rate. We believe this could be improved through more careful tuning of $\beta$ (e.g., time-varying discount factors), but it remains an open problem.
>
> ---
>
> **Q2.** "Recover $\tilde{O}(T^{1/3}P_T^{2/3})$ for exp-concave, or is there a barrier?"
>
> **A.** Existing $\tilde{O}(T^{1/3}P_T^{2/3})$ results are achieved by aggregating guarantees over intervals [2, 3]. These approaches require interval-level regret analysis and comparator-loss-specific arguments, e.g., exploiting KKT conditions relating $u_t$ and $\ell_t$ on bounded domains, which are not directly available in our template-level reduction.
>
> Our result offers a complementary tradeoff: a simple analysis yielding computationally efficient algorithms with meaningful guarantees, including the ability to handle unbounded domains that existing approaches do not easily accommodate. We believe that incorporating interval-level analysis into the modular framework could generalize the capability toward sharper rates, and we will discuss this direction in the revision.
>
> ---
>
> **W1.** "Novelty lies in modularization and proof streamlining."
>
> **A.** We respectfully believe that our contributions extend beyond modularization. The reduction enables: (i) the first Adam-like convergence guarantees with flexible $(\beta_1, \beta_2)$ in the non-smooth setting, including the first clip-free variant; (ii) the first dynamic regret bound for logistic regression avoiding exponential dependence on $B$ with efficient implementations; (iii) new technical tools (self-confident tuning lemma, analysis for margin conditions).
>
> ---
>
> **W2.** "Tradeoffs deserve stronger discussion."
>
> **A.**  **Logistic regression:** Since logistic loss is exp-concave, one can apply exp-concave methods:
>
> | | Dynamic Regret | Per-round Cost |
> |--|--|--|
> | [2] | $\tilde{O}(d^{3.5}e^B T^{1/3}P_T^{2/3})$ | $O(d^3\log T)$ |
> | [3] | $\tilde{O}(dT^{1/3}P_T^{2/3})$ | $O(d^{12}T^{11} + T^{23}\log T)$ |
> | Ours | $O(dB\log T + \sqrt{dBTP_T^\beta})$ | $O(d^2\log T + \log^2 T)$ |
>
> Our method avoids exponential dependence on $B$ and is efficient.
>
> **Adam:** The only prior result is [1], which requires $\beta_2 = \beta_1^2$ and clipping. Our tradeoff: bounded gradient assumption for sharper $(\beta_1, \beta_2)$ conditions. [1] also requires clipping — we *relax* this with a clip-free variant. Rebuttal experiments validate the clipping/damping designs and confirm acceptable impact of omitting bias correction.
>
> We will discuss these trade-offs more clearly in the revision.
>
> ---
>
> **W3.** "No empirical validation."
>
> **A.** During the rebuttal, we conducted experiments on CIFAR-10/ResNet-18 with L1 regularization (non-smooth). Using default Adam hyperparameters, the analyzed methods consistently improve over Adam. Results: [https://osf.io/8dqsj/](https://osf.io/8dqsj/files/mp56d?view_only=69d6ff6561684f1db8de57259cdfc590).
>
> ---
>
> We hope our responses have clarified the contributions and tradeoffs, and we would appreciate your reconsideration in light of these clarifications. Thank you.
>
> ---
>
> **References**
>
> [1] Ahn and Cutkosky, Adam with Model Exponential Moving Average is Effective for Nonconvex Optimization.
>
> [2] Baby and Wang, Optimal Dynamic Regret in Exp-Concave Online Learning.
>
> [3] Zhang et al., Non-stationary Online Learning for Curved Losses: Improved Dynamic Regret via Mixability.

---

> > ### Author Rebuttal · Reviewer_vRCu · 2026-04-05
> >
> > I thank the authors for their detailed answers and for the additional experiments they provided. Therefore, I increase my score to 4. However, I still don't agree with the novelty of modularization .

---

> > > ### Author Response · Authors · 2026-04-06
> > >
> > > We thank the reviewer for re-evaluating our work and the positive assessment.
> > >
> > > Regarding the novelty of the modular analysis, we would like to offer a further clarification. As acknowledged in the paper, our analysis builds on the D2D connection proposed by Ahn et al. (ICML'24). We agree that each application involves problem-specific technical work. However, we believe the value of this modular analysis lies in providing a *unified framework* that brings online linear regression, online logistic regression, and Adam convergence under a single template-level analysis. Analyzing these problems separately may require their own specialized analysis; our modular analysis unifies them by casting them all as instances of discounted FTRL with explicit stability and comparator-variation terms.
> > >
> > > This unified view is both natural and effective. FTRL is well suited for exploiting curved loss structure, and the template-level analysis directly enables new dynamic regret guarantees for online logistic regression. Moreover, since FTRL naturally fits Adam's update formula, the same reduction transforms the Adam analysis into a self-confident tuning problem in online learning, yielding new convergence results. We believe this ability to derive new results across different domains from a single modular analysis constitutes a meaningful contribution.
> > >
> > > We will incorporate the feedback from reviewers to more precisely position the modular analysis in the revision. Thank you again for the discussion and for the positive re-evaluation of our work.

---

### Official Review · Reviewer_NySE · 2026-03-13

**Soundness:** 3
**Presentation:** 3
**Significance:** 3
**Originality:** 3
**Overall Recommendation:** 5
**Confidence:** 2

**Summary:**

The paper proposes a modular discounted-to-dynamic reduction for FTRL-type methods and applies it in three places: discounted VAW for unconstrained online linear regression, discounted AIOLI for online logistic regression, and clipped/clip-free Adam variants under the O2NC framework. The resulting guarantees recover optimal dynamic regret for the linear-regression setting, give a logistic-regret bound without exponential-in-B dependence, and derive optimizer convergence results with refined conditions on $(\beta_1, \beta_2)$.

**Compliance With Llm Reviewing Policy:**

Affirmed.

**Final Justification:**

The rebuttal addressed my main concerns. I changed my evaluation from 4 to 5.

As mentioned in the initial review, I am unfamiliar with some pieces of related work. After careful consideration, the initial confidence score was a inaccurate choice which didn't match my confidence, and I have corrected it to 2.

**Key Questions For Authors:**

1. If feasible, can the paper add at least minimal experiments showing that the studied update rule and the newly justified parameter settings are practically meaningful?
2. I may be missing something, but the tuned bound stated in main-text Theorem 3 does not obviously match Appendix Theorem 8; in particular, the dependence on $B$ and the logarithmic factors looks different. Could the paper clarify the relationship between these statements and explain which bound should be taken as the final tuned guarantee?
3. For the clipped-Adam result, how tight is the sufficient condition $\beta_2 \ge \max(1 - \nu/(G+\sigma), \beta_1^4)$? In particular, is the $\beta_1^4$ term essential, or mainly an artifact of the proof?
4. The introduction discusses removing dependence on $\nu$ and bounded-gradient assumptions, but Theorems 4-7 still use both. Could the paper clarify the intended distinction?

I am not fully familiar with this line of work, and I would be open to updating my assessment after reading the authors' clarification.

**Limitations:**

The paper could summarize its limitations more explicitly in one place, especially that the optimizer results concern modified Adam variants and rely on assumptions such as bounded stochastic gradients and $\nu$-dependent conditions. It also lacks empirical evidence on whether the justified parameter regimes are practically meaningful.

**Strengths And Weaknesses:**

Strengths

- The modular discounted-to-dynamic reduction is clearly presented and appears to streamline prior dynamic-regret analyses.
- The theoretical development is technically substantial, and the optimizer extension is technically ambitious.
- The logistic-regret application is a substantive contribution: it gives a new dynamic-regret guarantee for a discounted AIOLI variant and avoids exponential-in-B dependence.
- The D2D theorem keeps comparator variation, stability, and comparator-dependent potentials explicit, which may make the reduction reusable in other settings.
- The optimizer analysis goes beyond the $\beta_2 = \beta_1^2$ coupling, including a clipped-Adam sufficient condition involving $\max(1 - \nu/(G+\sigma), \beta_1^4)$ and a margin-style dependence on $\beta_2 - \beta_1^2$.
- For curved losses, the linear regression result recovers known optimal rates and the logistic result avoids exponential-in-B dependence.
- The theorem statement and the mapping from discounted FTRL to the analyzed optimizer are transparent and potentially reusable.

Weaknesses

- Some claims about optimality and new $(\beta_1, \beta_2)$ conditions could use more careful wording.
- Given the optimizer-facing motivation, the paper would be stronger with at least some empirical evidence on the studied update rules and parameter settings, although I do not view experiments as essential to the theoretical contribution.
- The paper studies an Adam-like variant with a shared scalar adaptive denominator rather than standard coordinate-wise Adam. The paper already notes that the bias-correction terms are dropped, but this distinction would benefit from being stated more explicitly when broad "Adam" terminology is used.
- The novelty seems strongest in the template-level formulation and the applications, rather than in an entirely new discounted-to-dynamic identity.
- Some introduction-level wording could be aligned more closely with the formal theorem statements, which still rely on bounded-gradient and $\nu$-dependent assumptions.
- The relationship between the tuned logistic bound in Theorem 3 and the appendix statement in Theorem 8 could be clarified more explicitly.

---

> ### Author Rebuttal · Authors · 2026-03-30
>
> Thank you for the detailed review and the constructive feedback. We appreciate your recognition of the modular reduction, the logistic regression application, and the refined $(\beta_1, \beta_2)$ analysis. Below we address your questions and concerns.
>
> ---
>
> **Q1.** On the experiments.
>
> **A.** During the rebuttal period, we conducted preliminary experiments on CIFAR-10/ResNet-18 with L1 regularization (non-smooth objective). Using the same lr$=0.001$ and $(\beta_1,\beta_2)=(0.9, 0.999)$ as default Adam, both Clipped Adam and Clip-Free Adam consistently improve over Adam, with the advantage growing as the non-smoothness increases. These results validate that the algorithmic designs motivated by theory are meaningful in non-smooth settings. Full results: [https://osf.io/8dqsj/](https://osf.io/8dqsj/files/mp56d?view_only=69d6ff6561684f1db8de57259cdfc590).
>
> ---
>
> **Q2.** "Theorem 3 does not obviously match Appendix Theorem 8."
>
> **A.** Sorry for the confusion. The formal version of Theorem 3 is in Appendix C.2.2. The main-text version applies an upper bound on $\sum_{t=1}^T (F_t^{\beta, \varphi}(u_{t+1}) - F_t^{\beta, \varphi}(u_{t}))$ to make the tension between path length and stability more transparent. We will add cross-references to clarify it in the revision.
>
> ---
>
> **Q3.** "Is the $\beta_1^4$ term essential, or mainly an artifact?"
>
> **A.** When $1 - \nu/(G+\sigma)$ is close to $1$ (common in practice), the numerical distinction between $\beta_2 \ge \beta_1^4$ and $\beta_2 \ge \beta_1^2$ is small. Throughout the paper, we view $\beta_2 \ge \beta_1^2$ as the essential condition, especially for clip-free Adam. The $\beta_1^4$ relaxation is better understood as a theoretical signal that clipping can weaken the coupling between $\beta_1$ and $\beta_2$ by bounding $\|\Delta_t\|_2 \le D$. We will clarify this point more clearly in the revision.
>
> ---
>
> **Q4.** "Theorems 4–7 still use $\nu$ and bounded-gradient assumptions."
>
> **A.** Thank you for pointing this out. We agree the introduction could be more precise, and we address the two aspects separately.
>
> - **Bounded-gradient assumption:** All our results (Theorems 4–7) rely on Assumption 4 (bounded stochastic gradients). This is a trade-off we make to obtain more fine-grained $(\beta_1, \beta_2)$ conditions. The only prior result in this setting (Ahn and Cutkosky, NeurIPS'24) does not require this assumption but is restricted to $\beta_2 = \beta_1^2$. We will revise the introduction to accurately reflect this trade-off.
>
> - **Dependence on $\nu$:** Theorem 4 involves $\nu$ in the $\beta_2$ condition ($\beta_2 \ge \max \{ \beta_1^4, 1-\nu/(G+\sigma)\}$. The margin-style conditions are specifically designed to address this: the condition on $\beta_2$ becomes $\beta_2 \in [\beta_1^2 + m, 1-m]$, which does not involve $\nu$. While $\nu$ still appears as a $\ln(G/\nu)$ factor in the final rate, this term is non-dominant under typical parameter regimes (e.g., it is dominated when $G/\nu \le \mathcal{O}(\exp(G/\epsilon))$, a mild condition).
>
> ---
>
> **W1.** "Optimality and $(\beta_1, \beta_2)$ conditions could use more careful wording."
>
> **A.** We will revise the wording to more precisely reflect the formal statements. We will state explicitly that our optimality holds under mild conditions on $\nu$ with an additional bounded gradient assumption, and that $\beta_2 \ge \beta_1^4$ (Theorem 4) involves a $\nu$-dependent constraint while the margin-style condition suggests a preferred choice $\beta_2^\star = (1 + \beta_1^2)/2$ with $\rho$ quantifying the trade-off over a wider range of $\beta_2$.
>
> ---
>
> **W2.** "Empirical evidence."
>
> **A.** (See Q1.)
>
> ---
>
> **W3.** "Scalar adaptive denominator rather than coordinate-wise Adam," "bias-correction terms are dropped", "distinction would benefit from being stated more explicitly."
>
> **A.** As noted in Section 4.5, coordinate-wise guarantees can be recovered by running a separate algorithm for each coordinate; thus, the results naturally extend to the coordinate-wise version. In our experiments, we also compare results with and without the bias-correction term, and observe that their performance is similar. We agree that the wording of Adam can be improved, and we will revise it accordingly.
>
> ---
>
> **W4.** "Novelty strongest in template-level formulation, rather than a new D2D identity."
>
> **A.** We agree with this evaluation of our work. Our contribution is *not* a new D2D identity, but rather a modular way to *apply* it. The same template can be applied across different online regression problems, and new Adam convergence conditions through a unified framework.
>
> ---
>
> **W5.** "Introduction wording could be aligned with formal statements."
>
> **A.** (Also see W1.) We will align the wording in the Introduction with the formal statements.
>
> ---
>
> **W6.** "Theorem 3 vs Theorem 8."
>
> **A.** (See Q2.)
>
> ---
>
> We hope our responses have addressed your concerns. We warmly invite you to re-evaluate our work and welcome further discussions. Thank you!

---

> > ### Author Rebuttal · Reviewer_NySE · 2026-04-01
> >
> > I have read the rebuttal. Thank you for the detailed response. The added preliminary experiments are helpful, and the clarifications regarding the $\beta_1^4$ condition, the $\nu$/bounded-gradient assumptions, and the intended wording of the Adam claims address several of my concerns. The main point that remains unclear to me is the relationship between the main-text Theorem 3 and Appendix Theorem 8; the response helps with cross-referencing, but it does not fully resolve the discrepancy I was concerned about. Overall, the rebuttal is helpful, but it does not change my assessment to a clearly higher score. I will therefore keep my overall recommendation.

---

> > > ### Author Response · Authors · 2026-04-01
> > >
> > > Thank you for raising this again — we are sorry that our earlier response was not detailed enough to fully resolve your concern. To clarify:
> > >
> > > - **Theorem 3** (main text) is an *untuned* bound: $\beta\lambda\|u_1\|^2 + d(1+BR)\log(1 + \frac{R^2\sum_t \beta^{T-t}}{d\lambda(1+BR)}) + \frac{\beta}{1-\beta}P_T^\beta + \frac{1-\beta}{\beta}d(1+BR)T$, with explicit $\beta, \lambda$, designed to show the path-length/stability tension. We briefly mentioned that this can be tuned to $O(dB\log T + \sqrt{TP_T^\beta})$ to highlight that we also designed an ensemble algorithm capable of aggregating to this rate; this statement was kept informal due to page limitations.
> > >
> > > - **Theorem 8** (Appendix C.3) is the *final tuned* result: setting $\lambda = 1/B^2$ and tuning $\beta$ via a two-layer ensemble yields $O(dB\log(BT) + \sqrt{dBTP_T^{\beta_\star}})$. This should be taken as the final rate.
> > >
> > > We would also like to emphasize that the online logistic regression result is a *new* contribution. Compared to existing approaches for exp-concave losses:
> > >
> > > | | Dynamic Regret | Per-round Cost |
> > > |--|--|--|
> > > | Baby and Wang, COLT'21 | $\tilde{O}(d^{3.5}e^B T^{1/3}P_T^{2/3})$ | $O(d^3\log T)$ |
> > > | Zhang et al., ICML'25 | $\tilde{O}(dT^{1/3}P_T^{2/3})$ | $O(d^{12}T^{11}\log T + T^{23}\log T)$ |
> > > | Ours (Theorem 8) | $O(dB\log(BT) + \sqrt{dBTP_T^{\beta_\star}})$ | $O(d^2\log T + \log^2 T)$ |
> > >
> > > Our result provides a meaningful tradeoff between effectiveness and efficiency — avoiding exponential dependence on $B$, accommodating unbounded domains, and being computationally efficient — demonstrating the reusability of the modular analysis. We hope this clarification addresses the remaining concern.

---

### Decision · Program_Chairs · 2026-04-30

**Decision:**

Accept (regular)

**Comment:**

This paper develops a modular discounted-to-dynamic reduction for FTRL-based methods, enabling streamlined dynamic regret analysis for curved losses (online linear and logistic regression) and convergence guarantees for Adam optimizers under flexible parameter conditions.

The scores are overall quite positive: 5 (Accept), 4 (Weak accept), 4 (Weak accept), 4 (Weak accept) — overall positive.

All reviewers acknowledged the authors’ rebuttal; concerns about presentation clarity, assumptions, and novelty were partially resolved, with three reviewers increasing their scores (one from 4 to 5, two from 3 to 4) and the fourth maintaining a positive score after the authors provided additional experiments and clarifications.

Given the strong theoretical contributions and the constructive resolution of reviewer concerns, we recommend accepting the paper.